

# Seasonal variations of high time-resolved chemical compositions, sources and evolution for atmospheric submicron aerosols in the megacity Beijing

Wei Hu, Min Hu[*], Wei-Wei Hu, Jing Zheng, Chen Chen, Yusheng Wu, Song Guo

State Key Joint Laboratory of Environmental Simulation and Pollution Control, College of Environmental
Sciences and Engineering, Peking University, Beijing 100871, China
[*]Corresponding author: minhu@pku.edu.cn

## Abstract

Severe regional haze problem in the megacity Beijing and surrounding areas, caused by fast formation and
growth of fine particles, has attracted much attention in recent years. In order to investigate the secondary formation
and aging process of urban aerosols, four intensive campaigns were conducted in four seasons between March 2012
and March 2013 at an urban site in Beijing (116.31°E, 37.99°N). An Aerodyne high resolution time-of-flight
aerosol mass spectrometry (HR-ToF-AMS) and other relevant instrumentations for gaseous and particulate
pollutants were deployed. The average mass concentrations of submicron particulate matter ($PM_1$) were 45.1±45.8,
37.5±31.0, 41.3±42.7, and 81.7±72.4 μg m$^{-3}$ in spring, summer, autumn and winter, respectively. Organic aerosol
(OA) was the most abundant component in $PM_1$, accounting for 31, 33, 44 and 36% seasonally, and secondary
inorganic aerosol (SNA, sum of sulfate, nitrate and ammonium) accounted for 59, 57, 43, and 55% of $PM_1$
correspondingly. Based on the application of positive matrix factorization (PMF), the sources of OA were obtained,
including the primary ones of hydrocarbon-like (HOA), cooking (COA), biomass burning OA (BBOA) and coal
combustion OA (CCOA), and secondary component oxygenated OA (OOA). OOA, usually composed of more-
oxidized (MO-OOA) and less-oxidized OOA (LO-OOA), accounted for 63, 69, 47 and 50% in four seasons,
respectively. Totally, the fraction of secondary components (OOA+SNA) contributed about 60−80% to $PM_1$,
suggesting that secondary formation played an important role in the PM pollution in Beijing, and primary sources
were also non-negligible. The evolution process of OA in different seasons was investigated with multiple metrics
and tools. The average carbon oxidation states and other metrics show that the oxidation state of OA was the highest
in summer, probably due to both strong photochemical and aqueous-phase oxidations. BBOA and CCOA were
only resolved in autumn and winter, respectively, consistent with the agricultural activities (e.g., straw burning



after the harvest in suburban areas) in autumn and domestic heating in winter, signifying that the comprehensive management for the emissions from biomass burning and coal combustion are needed. High concentrations of chemical components in $PM_1$ in Beijing, especially in winter or in adverse meteorological conditions, suggest that further strengthening the regional emission control of primary particulate and precursors of secondary species is expected.

**Key words:** $PM_1$, Secondary organic aerosol, AMS, PMF, Seasonal variations, Urban aerosols

# 1. Introduction

With rapid economic development and urbanization, air pollution stands in an increasingly serious situation in China. Severe regional air pollution, mostly characterized by high concentration of fine particulate matter ($PM_{2.5}$), happens frequently. Severe fine particle pollution can lead to visibility deterioration, damage ecosystems and human health, and affect climate change substantially, which has attracted widespread attention. Submicron particulate matter ($PM_1$) accounts for a large proportion of both mass and number concentrations in $PM_{2.5}$. Its physical and chemical properties can be greatly modified by dynamic and chemical conversion processes in the atmosphere (Buseck and Adachi, 2008). A full examination on the properties of chemical compositions, sources, and formation mechanisms of $PM_1$ will improve our abilities to understand, predict, and control its impacts.

Multiple observations have been conducted to investigate the concentrations and chemical compositions of non-refractory submicron particle (NR-$PM_1$) at diverse sites around the world (Jimenez et al., 2009 and references therein), and also in China (Hu et al., 2016a; Huang et al., 2014; Xu et al., 2014; Sun et al., 2010 and references therein). The $PM_1$ concentration in the megacity Beijing was 2−10 times higher than those in western cities, highlighting the severe situation of particulate pollution. Anthropogenic emissions due to a dense population, as well as the adverse meteorological and geographical conditions, can result in high PM pollution in Beijing (Hu et al., 2016a).

In urban atmospheres, aerosols can be directly emitted from complicated anthropogenic sources, such as coal and biomass burning, cooking, traffic-related and industrial emissions, or formed through photochemical and heterogeneous/aqueous-phase oxidations (Lee and Allen, 2012). Secondary inorganic aerosols (SNA, sum of sulfate, nitrate and ammonium) are important in $PM_1$. Organic aerosol (OA) is the most complicated component in





fine particles, with a high mass fraction of 20−90% (Turpin et al., 2000; Carlton et al., 2009). OA is constituted by hundreds of species, but only a small fraction (~10%) can be quantified by chemical analysis (Hallquist et al., 2009). A better understanding on OA properties is crucial for identifying the sources of OA.

Aerodyne aerosol mass spectrometry (AMS) is currently one of the most popular technologies used to characterize the main species in NR-PM$_1$, and has been applied to many field and laboratory studies (Hallquist et al., 2009). By combining the time series of OA mass spectra (MS) from AMS with source receptor models, different OA sources can be effectively distinguished. Analyzed with positive matrix factorization (PMF) model, several OA factors have been resolved (Hu et al., 2013; Jimenez et al., 2009; Ng et al., 2011). For instance, hydrocarbon-like (HOA), cooking (COA), biomass burning (BBOA) and coal combustion OA (CCOA), etc., are classified into primary organic aerosols (POA). Oxygenated OA (OOA) usually consists of low-volatility (LV-OOA) and semivolatile OOA (SV-OOA) based on their correlations with sulfate and nitrate, respectively, and inferred volatilities. OOA is also identified as more oxidized (MO-OOA) and less oxidized OOA (LO-OOA) because they show different O/C ratios but insignificantly different volatility (Hu et al., 2016a; Setyan et al., 2012). Based on the AMS measurement, the aging process of OA can also be characterized with some metrics, e.g., C/H/O atomic ratios, OA/OC, carbon oxidation state (OS$_C$), and the abundance of characteristic fragment ions ($f_{44}$ and $f_{43}$). These results can help to quantify the contributions of primary emissions and secondary formations, and probe into secondary formation mechanisms and the aging process of OA (Ulbrich et al., 2009).

The characteristics and evolution of aerosol pollution are multifactorial, e.g., influenced by meteorological conditions, regional transport and local sources. Generally, these factors have quite distinct patterns in different seasons, thus the formation, transformation and removal of pollutants are affected. Some studies on the characteristics of submicron aerosol pollution in Beijing using AMS were mainly carried out in summer and winter. The characterization of PM$_1$ in summer (e.g., Huang et al., 2010; Sun et al., 2010), and the comparisons of PM$_1$ characteristics between summer and winter (e.g., Hu et al., 2016a; Sun et al., 2012, 2013a), between the low and high humidity atmosphere (e.g., Sun et al., 2013b), and between the polluted and unpolluted days (e.g., Zhang et al., 2014), were conducted in Beijing. Yet the study comprehensively characterized the seasonal variations of PM$_1$ under different meteorological conditions and pollutant sources in Beijing is lacking, especially based on high mass-resolution measurements. Zhang et al. (2013) and Sun et al. (2015) investigated the seasonal variations of PM$_1$ pollution in Beijing based on unit mass resolution (UMR), and there was no elemental information in these studies. High mass-resolution AMS can obtain elemental information, which can be used to more easily determine




the oxidation state of OA and characterize the evolution of SOA. Therefore, to deeply explore the formation and evolution of SOA in urban atmosphere, the researches on $PM_1$ seasonality in Beijing based on field observations applying high mass-resolution AMS remain very necessary.

In this study, a high resolution time-of-flight AMS (HR-ToF-AMS) and other real-time online measurement instruments for gaseous and particulate pollutants were deployed at an urban site in Beijing from 2012 to 2013 to investigate the seasonal characteristics of $PM_1$. At first, the seasonal variations of chemical compositions in $PM_1$ were analyzed. Then the seasonal sources of OA were fully explored. Finally, we compared and discussed the evolution process of OA in different seasons. This study is of great significance to further understand complex air pollution, and to provide scientific support for model simulations of atmospheric aerosols, and also a theoretical basis for fine particulate pollution control.

## 2. Experiments

### 2.1 Sampling site and measurements

During the periods from March 2012 to March 2013, four intensive campaigns were carried out at the PeKing University Urban Atmosphere Environment MonitoRing Station (PKUERS, 39.99°N, 116.31°E), which is located on the roof of a building (approximately 20 m above ground level) on the campus (Fig. S1 in the supplement). The details about the observation site were described in several published papers (Wu et al., 2007; Huang et al., 2010). A HR-ToF-AMS (Aerodyne Research Inc., USA) was deployed to measure the mass concentrations and size distributions of submicron non-refractory species, including OA and inorganic aerosols (sulfate, nitrate, ammonium and chloride). In addition to HR-ToF-AMS, multiple high time-resolution instruments for gaseous and particulate pollutant measurements (as listed in Table S1) were also available in these campaigns.

### 2.2 HR-ToF-MS operation and data analysis

Ambient air was first introduced through a cutoff cyclone of $PM_{2.5}$ (1.5 m above the roof) and a copper tube (3/8 inch) at 10 L min$^{-1}$, then was sampled into the AMS at 0.09 L min$^{-1}$. A Nafion drier was set in front of the inlet of AMS to keep the relative humidity (RH) of ambient air sampled into the AMS below 30%. The time resolutions applied in all campaigns were 4 min, with 2 min in V-mode for concentrations and size distributions, and 2 min W mode to obtain the HR-MS data. Ionization efficiency (IE) calibrations were done every few days by sampling





monodispersed 400 nm dried pure $NH_4NO_3$ particles into the AMS. Those particles were aerosolized by an aerosol atomizer (3076, TSI Inc., USA) and selected with a differential mobility analyzer (DMA, model 3081, TSI Inc., USA). IE values measured by the Brute-Force Single-Particle method (BFSP) were used. The ratio of IE to airbeam signal (=IE/AB, AB refers to $N_2^+$ detected by the AMS) of each calibration was applied for converting the instrument signals to actual mass concentrations. The IE/AB calibration values (Fig. S2) within the interval of two calibrations were obtained by linear interpreting IE/AB values before and after. Unless there was an instrument failure (e.g., turbo pump down and filament exchange), last well calibrated IE/AB value was applied till the instrument failure time. The size distribution of main species in NR-PM$_1$ was done in the beginning and ending of all campaigns by selecting pure $NH_4NO_3$ particles (stokes diameter 50−550 nm) with vacuum aerodynamic diameters ($d_{va}$) of 100−900 nm. The detected limits (V-mode) of organics, sulfate, nitrate, ammonium, and chloride during all campaigns were shown in Table S3. For the detailed operation and calibration procedures of HR-ToF-AMS, refer to Hu et al. (2016a).

The data analysis procedures are similar to the method reported in Hu et al. (2016a). Middlebrook et al. (2012) created an algorithm estimating AMS collection efficiencies (CE) for field data based on the aerosol chemical compositions and sampling line RH. The chemical composition-based CEs (around 0.5) were applied to calculate the mass concentration of chemical compositions in PM$_1$. Good correlations (shown in Fig. S3−S5) of the results between the AMS and other instruments also prove the reliability of CE used here. Pieber et al. (2016) found that the $CO_2^+$ interference related to $NH_4NO_3$ sampling on the vaporizer showed a median of +3.4% relative to nitrate, and highly varied between instruments and with operation history (percentiles $P_{10-90}$ = +0.4 to +10.2%). The effect of $NH_4NO_3$ on $CO_2^+$ ion signal was quite limited (about 1−2%) and not considered here. In this study, the "improved-ambient" correction (Canagaratna et al., 2015) was performed to calculate the elemental ratios of OA. The HR-MS data of OA in four seasons ($m/z$ range 12−283 in winter campaign and 12−196 in other campaigns) were analyzed with PMF respectively, which follows the procedures described in Ulbrich et al. (2009).

## 3. Results and discussion

### 3.1 Dynamic variations of PM$_1$ pollution

#### 3.1.1 Seasonality of chemical compositions in PM$_1$

The average concentrations of main chemical components in PM$_1$ and gaseous pollutants, and meteorological



conditions are listed in Table 1. The time series of meteorological parameters, including temperature, relative humidity (RH), atmospheric pressure, and wind speed and direction, can be found in Fig. S6-S9.

The average $PM_1$ concentrations (sum of non-refractory species measured by AMS and black carbon (BC) by aethalometer or MAAP) showed little difference in spring, summer, and autumn, which were 45.1±45.8, 37.5±31.0, and 41.3±42.7 µg m$^{-3}$, respectively. The average $PM_1$ concentration of 81.7±72.4 µg m$^{-3}$ in winter was the highest among the four seasons. OA was the most important component (31−44%) in $PM_1$, similar to previous studies (Fig. 1). In autumn, the contribution of the carbonaceous components (OA+BC) to $PM_1$ was over 50%. In spring, summer and winter, SNA accounted for about 60% in $PM_1$ due to the secondary aerosol formation through strong photochemical and aqueous-phase reactions. Wang et al. (2016) and Cheng et al. (2016) found that high levels of sulfate and fine PM can be explained by reactive aqueous oxidation of $SO_2$ by $NO_2$ under certain atmospheric conditions. Lower temperatures in winter, spring and autumn favored the partitioning to particulate nitrate and were partially related to higher concentrations of nitrate in $PM_1$ (Table 1).

Compared with the results obtained over the world, $PM_1$ and the concentrations of major chemical species in Beijing, especially in winter, were 2−10 times higher than those at American and European sites (Hu et al., 2016a). In China, $PM_1$ in urban Beijing in winter was much higher than those results (27−48 µg m$^{-3}$) obtained in the Yangtze River Delta (YRD) and Pearl River Delta (PRD) regions (Gong et al., 2012; He et al., 2011; Huang et al., 2011, 2012, 2013). While, $PM_1$ in other seasons was approximate to those obtained at the sites at Changdao Island, and in the YRD and PRD regions in the same seasons (Hu et al., 2013, 2016a and references therein). These results suggest that Beijing suffered from severe particulate pollution, especially in winter, which should be seriously taken into consideration.

Aerosol pollution in Beijing exhibited distinctive characteristics in four seasons, because of the significantly different meteorological conditions and emission sources of pollutants. Compared with the previous results in Beijing (Fig. 1), $PM_1$ in summer was lower than before, which likely resulted from the more effective rainout (Fig. S7) and lower concentrations of gaseous precursors (Table 1). In summer, high temperature, strong solar radiation and high oxidant concentrations generally enhance the secondary aerosol formation from gaseous precursors (such as $SO_2$, $NO_x$ and VOCs). As illustrated in Fig. 2, $PM_1$ in summer was mainly in the range of less than 50 µg m$^{-3}$ (~80%), and skewed-normally distributed, with the highest frequency (~45%) in 15−35 µg m$^{-3}$. The probability distribution of $PM_1$ was similar to the previous results in Beijing in summer (Huang et al., 2010; Sun et al., 2010),





but the range of $PM_1$ was narrower. In contrast, $PM_1$ in winter was higher than before, and equivalent to those in close periods (Zhang et al., 2014; Sun et al., 2014). The heaviest particulate pollution in winter was the co-effect of the large amount of particles emitted from primary sources, prolonged control of weak weather system (i.e., low mixed layer and static air), as well as the rapid generation and accumulation of secondary particles from gaseous

precursors (Hu et al., 2016a; Sun et al., 2013a).

In spring, autumn and winter, the passage of strong cold air parcels transported from Siberia and Mongolia through Beijing was usually accompanied by strong winds, thus atmospheric relative humidity decreased, pressure increased, which was conductive to the dispersal of pollutants (Fig. S6, S8 and S9). The heavy pollution processes were usually ended by the passages, demonstrating periodic cycles of aerosol pollution (Guo et al., 2014; Wu et al.,

2009). In spring and autumn, the biomass burning emissions in North China occurred often due to agricultural activities and traditional Tomb-sweeping Day. Dense fire points were observed by satellites (https://firms.modaps.eosdis.nasa.gov/firemap) in Beijing and surrounding areas, e.g., during 7−8, 26−28 Apr. 2012 (Fig. S6), and 18−20, 23−27 Oct. and 6−11 Nov. 2012 (Fig. S8). During these periods, the concentrations of OA and other pollutants increased substantially. Higher nitrate concentrations in spring and autumn, likely caused

by the secondary conversion of gaseous nitrogen oxides emitted from biomass burning (Fabian et al., 2005). The probability distributions of $PM_1$ in spring, autumn and winter were quite different from that in summer (Fig. 2), because the scavenging effect of strong wind was apparent due to the intrusion of long-range transported air masses. $PM_1$ mainly concentrated in the range of low levels (<20 $\mu g\ m^{-3}$), with frequencies of 40−60%. While, $PM_1$ ranged much more broadly, with the highest concentrations of over 200 or 300 $\mu g\ m^{-3}$, resulting from accumulated

pollutants under extremely unfavorable meteorological conditions or strong primary emissions.

**3.1.2 Contributions of chemical compositions to the increase of $PM_1$**

The contributions of chemical components in atmospheric aerosols at different concentration levels can help better understand the origins of chemical components. The proportions of chemical compositions as a function of $PM_1$ are shown in Fig. 2.

In all seasons, at low $PM_1$, the contribution of carbonaceous components (OA+BC) was dominant, accounting for 50−80% of $PM_1$. With the increase of $PM_1$, the proportions of SNA in $PM_1$ increased gradually, indicating that the enhancement of SNA primarily contributed to the increase of $PM_1$, consistent with previous studies (Huang et al., 2010; Sun et al., 2010; Sun et al., 2012). The proportions of nitrate increased more significantly, and the nitrate





concentration increased rapidly under higher RH (Fig. 2; Pearson correlation coefficients r=0.71, 0.34, 0.49 and 0.79, $p<0.01$). These results indicate that the aqueous reactions could contribute to nitrate remarkably in highly humid and static air. It is worth noting that when $PM_1$ higher than 150 μg m$^{-3}$, carbonaceous aerosols dominated $PM_1$ again in autumn, mainly resulting from large amounts of primary emissions. The proportions of chemical components in $PM_1$ varied less significantly in winter, reflecting the characteristics of regional pollution over the North China under the control of static weather system.

### 3.1.3 Diurnal patterns of chemical compositions in $PM_1$

Affected by different meteorological conditions, e.g., solar radiation, temperature, RH, boundary layer, and mountain-valley breeze in summer (Fig. S7), as well as different emission sources, the chemical compositions in $PM_1$ showed distinct diurnal patterns in four seasons. The diurnal patterns of gaseous and particulate pollutants during the seasonal observations are shown in Fig. 3.

Diurnal patterns of OA were similar in spring, summer and autumn. OA showed two obvious peaks around noon and in the evening. Similar peaks were also observed in winter, but were not as strong as in the other seasons. These two peaks corresponding to meal time were mainly caused by cooking emissions, which will be discussed later in Sect. 3.2.1. The peak concentration of OA in the evening in autumn was about two times higher than in spring and summer, consistent with the results in Oct.−Nov. 2011 (Sun et al., 2015), possibly because of the more intense cooking activities. In winter study, OA increased at night due to extra primary emissions, e.g., coal combustion and biomass burning.

Flatter diurnal cycles of sulfate were observed in four seasons, identifying the regional characteristics of sulfate formation (Sun et al., 2015). In the summer of Beijing, the formation of sulfate is mainly attributed to in-cloud/aqueous-phase reactions (70−80%), and also arises from photochemical oxidation of $SO_2$ (Guo et al., 2010). In this study, sulfate enhanced gradually from morning to late afternoon in summer, indicating that the photochemical production of sulfate might be significant. The aqueous-phase formation of sulfate in summer should also be more intense than in spring and autumn due to high temperature and high humidity (Fig. 3k–l). In winter, sulfate showed two peaks in the morning (6:00−10:00) and evening (around 20:00), which was likely influenced by the secondary formation of primarily emitted $SO_2$ (Fig. 3f), and the lower sulfate concentration in the daytime was quite related to the dilution effect of the planetary boundary layer (PBL). The PBL height decreases at night, and the atmospheric stability increases, thus the pollutants are difficult to spread (Hu et al., 2016a).



Nitrate showed distinct diurnal patterns in four seasons. In-cloud/aqueous-phase reactions and gas-to-particle condensation processes are two main pathways to form fine mode nitrate (Guo et al., 2010). Nitrate exhibited an obvious diurnal pattern in summer, with the concentrations gradually decreasing in the daytime, indicating that $NH_4NO_3$ evaporated due to high temperatures (Fig. 3k) and overcame the photochemical production of nitrate. The elevated PBL also plays a role in the low concentrations of nitrate in the daytime (Sun et al., 2015). In winter, nitrate increased gradually during 9:00−20:00, which did not appear in other seasons, suggesting that in addition to photochemical processes in the daytime, the partitioning of $NH_4NO_3$ into particulate nitrate was more significant due to low temperatures (<5°C; Fig. 3k). Similar diurnal patterns in summer and winter have been observed in previous studies (Sun et al., 2015). The diurnal variation of nitrate in spring and autumn was insignificant. Because ammonium mainly existed in the forms of $(NH_4)_2SO_4$ and $NH_4NO_3$, its diurnal patterns exhibited a combined effect of sulfate and nitrate formation mechanisms.

Chloride showed higher concentrations at night, with a peak in the morning, and then deceased in the daytime. Such a diurnal pattern was possibly caused by high primary emissions of chloride at night. It was also driven by the diurnal patterns of both temperatures and the PBL height, and controlled by the temperature-dependent gas-particle partitioning (Hu et al., 2016a; Sun et al., 2015). Chloride usually presents at semi-volatile state in the form of ammonium chloride in urban areas (Zhang et al., 2005), therefore the diurnal cycles of temperature and chloride trend oppositely. In winter, chloride was higher at night because of coal combustion and biomass burning emissions. Hu et al. (2016a) estimated that in winter over 50% of chloride in $PM_1$ existed as $NH_4Cl$, and part of chloride existed as KCl and NaCl. BC was mainly affected by the diurnal variation of the PBL height. BC showed a marked rise at night (around 21:00-0:00) and a small morning peak (7:00-8:00). Nocturnal heavy-duty vehicular exhausts (Lin et al., 2009; Hu et al., 2012), coal combustion and biomass burning emissions in the urban atmosphere also contributed to the high concentration of BC at night. The morning peak of BC was consistent with that of $NO_x$, probably due to vehicular emissions during rush hour.

### 3.1.4 Size distributions of chemical compositions in $PM_1$

The mass size distributions of chemical compositions in $PM_1$ during seasonal campaigns are shown in Fig. 4. The size distributions of SNA concentrated in accumulation mode, peaking at about 600−800 nm ($d_{va}$), indicating the internal mixed states of submicron aerosols. The mode diameters of SNA in Beijing were higher than those (500−600 nm) in Hong Kong in four seasons (Li et al., 2015). The mode size (695 nm) of sulfate was smaller than that (790 nm) of nitrate in autumn, while the mode size of sulfate (760 nm) was larger than that (700 nm) of nitrate


in winter. These results suggest the different particle growth rates were likely related to the different formation mechanisms of sulfate and nitrate. Sulfate was formed regionally, while nitrate formation was locally dominant (Guo et al., 2010). The size distribution of ammonium was more consistent with that of nitrate in autumn and winter. Nitrate likely mainly existed in the form of $NH_4NO_3$, resulted from the condensation of gaseous $HNO_3$ and $NH_3$ on

the surfaces of atmospheric particles (Liu et al., 2008).

During the winter, the peak size of OA was close to those of SNA; while in other seasons it was smaller than those of SNA, indicating a more aged state of OA in winter. The mass size distributions of chemical components in winter were also similar to those in the heavy-polluted episodes in other seasons, consistent with the severity of aerosol pollution in winter. In spring, summer and autumn, the size distributions of OA were wider than those of

SNA. The OA concentrations were much higher than those of SNA in the range of small size (100−500 nm), similar to the results in other urban areas (Weimer et al., 2006; Aiken et al., 2009), probably caused by the contribution of POA (Huang et al., 2010; Hu et al., 2013). Especially, influenced by primary emissions, the concentration of OA at peak size was much higher than those of other species in autumn. During the spring, summer and autumn, the proportions of OA decreased, while those of SNA increased rapidly with the increase of particle size (>200 nm),

indicating that SNA species were the main components contributed to the enhancement of $PM_1$, in accordance with that mentioned in Sect. 3.1.2. Different OA sources played different roles with the increase of particle size in winter, and the proportion of OA (about 40%) varied slightly. However, the proportions of sulfate rose with the increase of size, implying the significant contribution of regional transported sulfate.

## 3.2 Investigating OA sources with PMF

The sources of OA in four seasons were resolved by combining high-resolution OA mass spectra and PMF model respectively. The resolved fractions of OA during seasonal observations are shown in Fig. 5. The mass spectra and time series of OA components and some external tracers in each season are shown in Fig. S10-S13 and Fig. S14-S17, respectively.

In spring and summer, four OA components, i.e., MO-OOA, LO-OOA, COA and HOA, were identified. OOA

(MO-OOA+LO-OOA) dominated OA (63−69%), signifying the predominant contribution of SOA formation. Consistent with previous results, OA was dominated by SOA (about 60−70%) in Beijing in spring and summer (Fig. 5, and Guo et al., 2012, 2013). In recent years, the SOA contribution to OA (about two third) is quite stable in summer in Beijing (Fig. 5). OA was constituted by OOA, COA, BBOA and HOA in autumn and MO-OOA, LO-



OOA, COA, HOA and CCOA in winter. In autumn, POA (sum of HOA, COA and BBOA) was dominant in OA because primary emissions, e.g., biomass burning, strongly influenced Beijing and surrounding areas. The fraction of POA was close to that in the autumn of 2008 (Fig. 5b). In winter, secondary formations and primary emissions (e.g., biomass burning and fossil fuel combustion) contributed to OA equivalently. The fraction of SOA in OA in

winter (50%) was much higher than those (20−30%) in previous studies, and comparable to the results in the same winter in Beijing (Fig. 5b and Table S5), which was related to effective SOA formation under the stable meteorological conditions during long-lasting severe haze episodes. LO-OOA dominated OA in summer (44%) due to the freshly secondary formation from strong photochemical oxidations; whereas, MO-OOA was dominant in OA in winter (33%), maybe because the air masses were more aged on heavy-polluted days.

The comparison of seasonal variations of OA in $PM_1$ between this study and studies carried out at other sites is shown in Fig. 5c. The fractions of SOA during all seasons in Beijing were higher than or comparable to those at an elevated site (1534 m a.s.l) in Mt. Tai, North China (Zhang et al., 2014), and in both Beijing and Mt. Tai the fractions of SOA were higher in summer than in other seasons. The contribution of SOA to OA in Beijing was much lower than those in less-polluted atmospheres in Hong Kong (Li et al., 2015) and southeastern USA

(Budisulistiorini et al., 2016). The fraction of SOA in OA was as high as 80−86% in Hong Kong in four seasons. At a downtown site in Atlanta, SOA accounted for over 80% in spring and summer, and 65 and 56% in autumn and winter. At a rural/forested site Look Rock, Great Smoky Mountains, even no POA sources were resolved except for BBOA in winter. Different from the two sites in southeastern USA, there were no isoprene-epoxydiols-derived SOA and a biogenic influenced factor characterized by distinct $m/z$ 91 resolved in urban atmosphere in Beijing, the

same as addressed in Hu et al. (2015).

        Overall, the proportions of OOA in OA in this study were comparable to the average (58%) obtained at other urban sites around the world (Zhang et al., 2011). In total, secondary species (SNA+SOA) accounted for 80, 80, 64 and 73% of the total $PM_1$, respectively. The serious secondary pollution stresses the importance of control measures targeting the emission reduction of gaseous precursors ($NO_x$, $SO_2$ and VOCs).

**3.2.1 Primary OA sources**

**COA**

        COA refers to OA emitted from cooking activities (Allan et al., 2010). The knowledge on the transformation of COA in the atmosphere is limited (Dall′Osto et al., 2015). The reductive alkyl fragment ions are abundant in the



MS of COA (Fig. S10-S13). The abundance of fragments $m/z$ 55 and $m/z$ 57 ($f_{55}$ and $f_{57}$, mass fractions of $m/z$ 55 and $m/z$ 57) in the MS of COA were about 7−9% and 3−4%, respectively. Higher $f_{55}$ and $f_{57}$ are the most remarkable characteristic in the MS of identified COA factors, which are crucial to identify COA and HOA (Mohr et al., 2012). In this study, $f_{55}$ in COA factors was significantly higher than in corresponding HOA factors, and $f_{57}$ in COA factors

were comparable to $f_{57}$ in corresponding HOA factors (Fig. S10-S13). The oxidation state of COA was low, with low O/C (0.13−0.23) and OA/OC (1.33−1.46) ratios in this study.

As the edible oil is rich in oleic and linoleic acids (Dyer et al., 2008), the fragments $m/z$ 55 and $m/z$ 57 are not only contributed by the reductive alkyl fragments $C_4H_7^+$ and $C_4H_9^+$, but also by oxygenated fragments $C_3H_3O^+$ and $C_3H_5O^+$. The mass fractions of the later ones were about 1/3 of those of the reductive alkyl fragments with the same

$m/z$. The ratios of oxygenated and reductive fragments $m/z$ 55 and $m/z$ 57 are used to identify the sources of OA. The ratio of $C_3H_3O^+$ to $C_3H_5O^+$ in COA is about 2, while in HOA about 1; the ratio of $C_4H_7^+$ and $C_4H_9^+$ in COA is about 2.5, in HOA approximate 1 (Mohr et al., 2012). In this study, the ratios of $C_3H_3O^+$ to $C_3H_5O^+$ in COA were 2.8, 3.4, 3.8 and 2.4, and in HOA were 0.0, 0.6, 0.0 and 1.3; the ratios of $C_4H_7^+$ and $C_4H_9^+$ in COA and HOA were 1.8, 1.8, 2.3 and 2.4, and 1.1, 1.6, 1.1 and 1.1 during four seasons, respectively.

The rationality of resolved COA factors was investigated through their concentrations, diurnal patterns, correlations of COA with external tracers, and the uncentered correlations (UC) of MS. The COA concentrations could be simply estimated with the formula provided by Mohr et al. (2012) based on the fragments $m/z$ 55, $m/z$ 57 and $m/z$ 44. The estimated COA concentrations were 4.7, 4.4, 6.5 and 5.3 μg m$^{-3}$ in four seasons, accounting for 33, 35, 35 and 18% of OA, respectively. Compared with the concentrations (3.7, 2.5, 5.2 and 4.3 μg m$^{-3}$) and

proportions (27, 20, 28 and 15%) based on AMS-PMF analysis (Fig. 5), the results were overestimated, especially in summer, which is likely related to the applicability of experience parameters in different regions and seasons. The COA factors showed apparent diurnal patterns during seasonal observations, which is a key feature in identifying COA (see Sect. 3.2.2).

COA correlated well with some gaseous and other particulate pollutants, e.g., $NO_x$ and BC (Table S6-S9).

Dall′Osto et al. (2015) considered that the COA factors at a rural site contained more sources other than cooking emissions, based on the good correlations between COA with HOA, BC, $NO_x$, nitrate, etc. The UC coefficients between the MS of resolved COA factors in this study and the average MS of COA in previous studies (Lanz et al., 2008; Mohr et al., 2009; Huang et al., 2010; He et al., 2010) were calculated as Ulbrich et al. (2009). The UC



coefficients were in the range of 0.835−0.996 (Table S10), confirming the good similarity and rationality of the resolved COA factors.

In previous studies, it was reported that COA accounted for 14−25% (~20% on average) of the total OA in summer and winter in Beijing, with an average concentration (~6 µg m$^{-3}$), which is a relatively stable component of OA (Hu et al., 2016a; Wang et al., 2009). In this study, the concentrations and proportions of COA in OA during seasonal observations were in the range of 2.5−5.2 µg m$^{-3}$ and 15−28%, respectively, comparable to previous results. The highest concentration and proportion of COA in autumn were likely caused by strong emissions from barbecue activities, which are popular in autumn in both urban and suburban Beijing. Fewer cooking activities during and around the Chinese New Year holiday (7−19 Feb.; Fig. S17), as well as the lower evaporation rate of oil, led to the lower concentration and proportion of COA in winter. Overall, COA is an important non-fossil POA sources in all seasons, which should be taken into consideration, especially in autumn.

**BBOA**

The BBOA factor that was identified in autumn accounted for 11% of OA. However, the contributions of biomass burning to OA in other seasons might be relatively low (probably<5%) and cannot be resolved in our dataset by PMF. Levoglucosan is a significant tracer of biomass burning emissions, the fragment of which, $C_2H_4O_2^+$ (*m/z* 60) is regarded as a tracer of BBOA (Alfarra et al., 2007; Cubison et al., 2011). The abundance of *m/z* 60 (~1.3%) in the MS of BBOA was much higher than the background abundance (0.3%) in the urban without biomass burning emissions, which is a primary feature in identifying BBOA factor. The O/C ratio (0.24) of BBOA factor was higher than that (0.07) of HOA, and comparable to those (0.2−0.4) in previous studies (Mohr et al., 2009; He et al., 2010, 2011). Autumn is the harvest season for corn and other crops in North China. In autumn, burning crop straw randomly happened quite often, resulting in serious atmospheric particulate pollution in Beijing and surrounding areas (Duan et al., 2004; Zheng et al., 2005).

BBOA tracked well (Fig. S16, Table S8) with chloride, BC, $C_2H_4O_2^+$ and acetonitrile, similar to the results of previous studies (DeCarlo et al., 2010; He et al., 2011; Gong et al., 2012). Particles emitted from biomass burning could contains a high proportion of chlorides (Silva et al., 1999). The diurnal variations of BBOA were similar with those of HOA, with higher concentrations at night and in the morning and lower ones in daytime (see Sect. 3.2.2). They could be from similar emission processes, such as residential burning activities.

**CCOA**


Coal accounts for two third of the total primary energy consumption and coal combustion is an important source of air pollution in China (You and Xu, 2010; Huang et al., 2014). CCOA was only resolved in winter, consistent with the domestic heating period, suggesting that this CCOA factor was dominated by residential burning with higher OA emission factors (Hu et al., 2016a). In addition to typical ion fragments from fossil fuel combustion,

alkyl fragments ($C_nH_{2n+1}^+$ and $C_nH_{2n-1}^+$), the MS of CCOA also showed pronounced signals of polycyclic aromatic hydrocarbons (PAHs) ion fragments, e.g., $C_{10}H_8^+$ (*m/z* 128) from naphthalene and $C_{14}H_{10}^+$ (*m/z* 178) from anthracene (Fig. S13). The CCOA spectrum in winter is also similar to those in our previous studies at Changdao Island and in the winter of Beijing (Hu et al., 2013, 2016a). Compared to the average H/C ratios (1.76–1.96) in other POA factors (Canagaratna et al., 2015), H/C in CCOA (1.45) is lower. The O/C and OA/OC ratios in CCOA

were 0.14 and 1.32, respectively. CCOA accounted for 17% of total OA on average, within the range of 10–33% reported in other studies in the winter of Beijing (Figure 5; Hu et al., 2016a).

**HOA**

HOA is the sum of unresolved reductive and primary OA except specific OA (e.g., COA, BBOA and CCOA), which is generally considered to be related to the emissions of fossil fuel combustion. The mass spectra (MS)

features of HOA in this study are similar to those in previous studies (Huang et al., 2010; Hu et al., 2013 and references therein). Alkyl fragments ($C_nH_{2n+1}^+$ and $C_nH_{2n-1}^+$) are abundant in the spectra (Fig. S10-S13). During four observations, O/C and OA/OC ratios of HOA factors were 0.11, 0.19, 0.07 and 0.36, and 1.32, 1.41, 1.27 and 1.63 during four seasons, respectively. Compared with the corresponding ratios (O/C: 0.04−0.26; OA/OC: 1.21−1.50) of HOA in previous studies (Canagaratna et al., 2015), the higher values in winter were influenced by

the contribution of BBOA. Absolutely independent sources cannot be identified by factor analysis (e.g. PMF model). Therefore, the typical MS features of OA from other primary sources can be found in the MS of HOA. In this study, the abundant fragment ions in the MS of POA factors, e.g., CCOA (*m/z* 67, 69, 91, etc.) and COA (*m/z* 55, 57, etc.) were also presented in the MS of HOA (He et al., 2011; Hu et al., 2013, 2016a). This can also be confirmed through the correlations between HOA and external tracers (See Table S6-S9). HOA tracked well

(r=0.6–0.9) with traces of primary emissions (e.g., coal combustion and vehicular exhausts), such as chloride, $NO_x$, CO, BC and $C_2H_4O_2^+$, consistent with the primary sources of HOA.

**3.2.2 Oxygenated OA sources**

OOA, considered as a good alternative of SOA in most cases, has been extensively investigated in a large




number of previous studies (Jimenez et al., 2009; Zhang et al., 2005; Ulbrich et al., 2009; Ng et al., 2011). During the seasonal observations, OOA was an important component of OA. In spring, summer and winter, OOA was separated into MO-OOA and LO-OOA. Oxygenated fragments ($C_xH_yO_z^+$) are prominent in the MS of OOA. The abundance of $C_xH_yO_z^+$ in MO-OOA was higher than that in LO-OOA. For instance, the fragment of carboxylic acid $CO_2^+$ (*m/z* 44) accounted for 17−21% in MO-OOA, and 7−16% in LO-OOA. The average O/C ratios for SV-OOA, LV-OOA and OOA reported in some studies around the world were 0.53, 0.84 and 0.67, respectively (Canagaratna et al., 2015). In spring, summer and winter, the O/C ratios in resolved MO-OOA factors were about 0.90, 0.91 and 0.84, higher than those of corresponding LO-OOA (0.72, 0.67 and 0.77); in autumn, the O/C ratio of OOA was 0.88. In all four seasons, O/C ratios of OOA were higher than the average values mentioned above, indicating that OA was much more oxygenated and secondary formation contributed significantly to OA in urban Beijing.

MO-OOA, as extremely aged secondary species, exhibited good correlations with SNA (r=0.78−0.97), and the correlation coefficients between LO-OOA and SNA were slightly lower (Fig. S14−S17 and Table S6−S9). The total oxidant $O_x$ ($O_x = O_3 + NO_2$) showed good correlations (r = 0.5−0.8) with LO-OOA and OOA (autumn) factors, suggesting that gas-phase oxidation made an important contribution to LO-OOA formation. In spring and summer, LO-OOA and nitrate trended well during most days; and LO-OOA trended well with nitrate during the whole winter campaign. Overall, OOA (MO-OOA+LO-OOA) correlated well (r=0.89−0.96) with secondary inorganics ($SO_4^{2-}$+$NO_3^-$). Note that gaseous pollutants and main chemical compositions in $PM_1$ displayed good correlations with each other in winter, which may be caused by regional pollution characteristics under weak weather system in heavy pollution days.

### 3.2.3 Diurnal variations of OA factors

The diurnal patterns of OA components during seasonal observations are shown in Fig. 6. As mentioned above, the COA factors showed obvious diurnal patterns during all four campaigns, with two peaks at noon (about 13:00) and in the evening (about 20:00), in accordance with living habits of residents. The concentrations of COA at the noon peak (about 12:00−14:00) were 3.9, 2.5, 4.0 and 4.4 μg m$^{-3}$ on average; the daily highest concentrations of COA factors appeared at the evening peak (18:00−21:00), as 7.6, 6.2, 14.0 and 7.5 μg m$^{-3}$ on average, respectively. HOA was lower in the daytime and higher at night. The nocturnal activities, such as heavy-duty diesel vehicles only permitted at night (Lin et al., 2009), biomass burning and coal combustion, could make more significant



contributions to POA. In autumn, the OA concentrations began to raise in the evening, peaked at about 19:00, and kept higher at night (Fig. 2), reaching twice that in the daytime. As Fig. 6c shown, the anthropogenic emissions, e.g., biomass burning, coal combustion and especially cooking emissions, contributed significantly to OA. In winter, CCOA showed clear diurnal variations with high concentrations at night, consistent with the residential heating periods. The high concentration and (10 μg m$^{-3}$) and fraction (~30%) of CCOA in total OA points to strong coal combustion emissions at night in Beijing.

The peaks of OOA (or LO-OOA) coincided with the peaks of primary emitted COA (spring, summer and autumn) and HOA (winter) in diurnal patterns, probably because strong primary emissions favored the partitioning of oxidized gas precursors to particulate phase. In summer, OOA showed obvious diurnal variations: MO-OOA peaked in the morning and afternoon; LO-OOA showed two pronounced peaks at noon and at night, which was likely influenced by the photochemical oxidations and aqueous-phase formation from POA. The concentration and proportion of LO-OOA increased significantly in the afternoon (12:00−16:00), up to 7 μg m$^{-3}$ and 50%, respectively (Fig. 6b), suggesting that LO-OOA was a strong local/regional photochemical product despite the much higher PBL in the daytime (Hu et al., 2016a). In winter, despite the expanded PBL, MO-OOA increased gradually from noon, and peaked in the evening, implying that MO-OOA could form through photochemical oxidations in the daytime.

### 3.2.4 Source contributions to the OA increase

The probability distributions of OA and the fractions of OA components as a function of OA concentrations are illustrated in Fig. 7. In spring, autumn and winter, similar to the probability distributions of PM$_1$, the OA concentrations were frequently (about 50−80%) lower than 10 μg m$^{-3}$, due to the scavenging effect of strong wind accompanying long-range transported air parcels. In autumn and winter, primary emissions (e.g., cooking emission and coal combustion) influenced strongly, resulting in the wide ranges of OA concentrations, up to 120 μg m$^{-3}$ and 140 μg m$^{-3}$, respectively. In contrast, OA concentrations presented a skewed-normal distribution, mainly in the range of 5−20 μg m$^{-3}$ in summer, which is similar to previous results (Huang et al., 2010; Hu et al., 2016a).

During the spring observation (Fig. 7a), as the OA concentration was less than 10 μg m$^{-3}$, LO-OOA dominated OA (up to 70%), which was likely associated with the new particle formation in clean atmosphere (Wu et al., 2007). When the OA concentration was higher than 10 μg m$^{-3}$, OOA primarily contributed to the increase of OA, and the fraction of MO-OOA kept as about 40−50% as the OA concentration was over 30 μg m$^{-3}$. In summer, as the OA



concentration was lower than 30 μg m$^{-3}$, OOA accounted for about 60−70% of OA (Fig. 7b), which may have resulted from efficient LO-OOA formation due to both the strong photochemical and aqueous-phase oxidations under high RH and high temperature conditions (Xu et al., 2017). With the increase of OA, the fraction of MO-OOA enhanced gradually, implying that the aged MO-OOA contributed importantly to the OA increase. In autumn,

with the increase of OA, the proportion of OOA decreased gradually (Fig. 7c). In winter, when the OA concentration was lower than 70 μg m$^{-3}$, OOA and POA contributed equally to OA, while the fraction of LO-OOA increased gradually (Fig. 7d). In both autumn and winter, the fractions of OOA slightly increased around 100 μg m$^{-3}$, implying that POA probably transformed to SOA more effectively within this range.

    In spring, summer, and autumn, as the OA concentration was higher than 30 μg m$^{-3}$, the fraction of COA in

OA increased to different extent, reaching 30−60%. In autumn, the contributions of BBOA and HOA to OA were relatively stable, while that of COA dramatically enhanced with OA increase, probably because that the static air during some intervals (e.g., 23-25 Oct.) was not conducive to the dispersal of heavy emissions from cooking activities (Fig. S8 and Fig. S15). Furthermore, in autumn charcoal-grilling or barbecuing has become one of the most popular outdoor recreational activities in urban Beijing and surrounding areas, which could be an important

source of COA. These results indicate that strong cooking emissions were responsible for high OA concentrations partially in these seasons. In winter, when the OA concentration reached 100 μg m$^{-3}$, the fraction of CCOA dramatically increased, from 10% to about 40%, indicating that the strong emissions from coal combustion contributed predominately to high OA concentrations (Hu et al., 2016a). The O/C ratios of OA at different OA concentrations were dependent on the source contributions, showing lower values when the POA were dominant

in all seasons (Fig. 7).

## 3.3 The aging process of OA

### 3.3.1 Elemental ratios, van Krevelen diagram and carbon oxidation state

    Some important metrics or tools, such as elemental ratios, the van Krevelen (VK) diagram and carbon oxidation state (OS$_C$) of OA can be used to investigate dynamic evolution and oxidation mechanisms for bulk

organic aerosols.

    The elemental compositions of OA are closely related to their properties, e.g., density, hygroscopicity and vapor pressure. The element ratios and OA/OC ratios in OA obtained from seasonal observations in Beijing and at





other urban and rural/suburban sites are listed in Table 2. During the campaigns, the average O/C and H/C ratios (in atomic number), and OA/OC ratios were in the range of 0.47–0.53, 1.52–1.61, and 1.77–1.88, respectively. In spring, the average OA/OC ratio in Beijing was lower than those determined in Mexico City, Bologna, and at Changdao Island (Table 2). In the summers of 2011–2012, the OA/OC ratios in Beijing were higher than those

measured at urban sites in Shenzhen and Riverside, and rural/suburban sites in Jiaxing and Melpitz, indicating a high oxidation state of OA due to strong SOA formations via photochemical reactions in Beijing. However, in summers of 2008 in Beijing and 2010 in Shanghai, the OA/OC ratios were quite lower, maybe because the reduction of pollutant emissions reduced SOA formation during the Beijing Olympic Games and Shanghai World Expo, respectively. During autumn, the OA/OC ratio in Beijing was slightly higher than that in Shenzhen, but far lower

than those obtained at rural/suburban sites, e.g., Kaiping, Heshan and Melpitz. The observed OA/OC ratio in Beijing in winter 2012 was higher than those in Fresno and in Beijing 2010, but comparable to or lower than those at rural/suburban sites such as Ziyang, Jiaxing and Melpitz. Overall, the OA/OC ratios in Beijing were higher among urban sites except for in spring, and lower among rural/suburban sites except for in summer.

The van Krevelen (VK) Diagram is an important tool to investigate the evolution and functional group

alteration of OA (Heald et al., 2010). As shown in Table 3 and Fig. S18–S21, H/C and O/C ratios of OA exhibited good negative correlation (coefficient of determination $r^2$ = 0.70–0.79), with the slopes -0.57, -0.62 and -0.67, and intercepts 1.91, 1.94, 1.90 in the VK diagram in spring, summer and autumn, respectively. The slopes were shallower than those (-1.0–-0.7) obtained across the world (Chen et al., 2015), but steeper than that determined in Ziyang (Table 3). The intercepts were slightly lower than those (2.1–2.2) at urban and downwind sites (Chen et al.,

2015). However, in winter, the correlation was not good ($r^2$ = 0.02), with a fitted slope -0.08, exhibiting a "broader" range in the VK Diagram than in other seasons (Fig. 8 and Fig. S21).

Heald et al. (2010) concluded that OA trends to evolve along with a slope of -1 in the VK Diagram. Ng et al. (2011) derived that the evolution from SV-OOA to LV-OOA is mainly along with a slope of approximately -0.5, and is associated with the replacement of carboxyl functional group (OH-(C=O)-). The identified reactions related

to the alteration of functional groups are shown in Fig. 8. In spring, summer and autumn, the slopes fell between -1 (the addition of carboxyl functional groups without fragmentation or carbonyl and hydroxyl in different carbons) and -0.5 (carboxyl functionalization with fragmentation). In winter, the scatterplot of H/C vs. O/C ratios in the VK Diagram showed "broader" slopes, hinting the more complex sources and evolution processes of OA. The scatterplot indicated that OA in winter mainly evolved between the hydroxylation or peroxidation reactions (slope

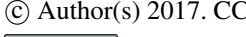



= 0) and carboxyl groups addition with fragmentation (slope = -0.5).

The average carbon oxidation state ($\overline{OS_C}$), approximated by 2×O/C–H/C, can also reflect the oxidation degree of OA (Kroll et al., 2011). The estimated $\overline{OS_C}$ of OA during seasonal observations are listed in Table 2 and the integrated Tri-VK-$\overline{OS_C}$ diagram is shown as Fig. 8. In summer and winter, the average carbon oxidation states

were higher, with the average of -0.54±0.27 (in the range of -1.42–0.23) and -0.58±0.25 (-1.32–0.11), respectively. In spring and autumn, the average carbon oxidation states were -0.64±0.37 (-1.96–0.55) and -0.66±0.39 (-1.66–0.64). Compared with previous studies, in spring, the $\overline{OS_C}$ was significantly lower than those at Changdao Island and Bologna sites, but slightly higher than that in Mexico City. In summer, the $\overline{OS_C}$ was comparable to those in Beijing in 2011 and Melpitz, and much higher than those in Beijing in 2008, Shenzhen, Shanghai, Riverside,

and Jiaxing. In autumn, it was higher than that in Shenzhen, lower than those in Kaiping and Melpitz, and close to that at Heshan site. The $\overline{OS_C}$ in winter was significantly higher than those in Beijing 2010, Fresno and Jiaxing, but lower than those in Ziyang and Melpitz. In summary, in urban Beijing, bulk organic aerosols were in a higher oxidation state in summer and in a medium oxidation state in other seasons, generally with lower oxidation state than rural/suburban sites. However, compared with the oxidation states of SV-OOA, OOA and LV-OOA

summarized by Canagaratna et al. (2015), the oxidation states of OOA in Beijing were generally higher than in other areas, especially for LO-OOA (Fig. 8). The oxidation states of MO-OOA in Beijing were only lower than those in very aged air masses in Ziyang in the basin (Hu et al., 2016b), over Mexico City (DeCarlo et al., 2010) and in Barcelona (Mohr et al., 2012). The oxidation states of LO-OOA were only slightly lower than those of MO-OOA in Beijing, and were comparable to those of MO-OOA in other urban areas (Fig. 8).

**3.3.2 Evolution of OA along with the aging of air masses**

The aging process of the air mass indicates that the physical and chemical changes occur and that secondary organics or inorganics form continuously, and the physicochemical properties of the air mass are modified during the transport of the air mass from source areas. In previous literatures, the ratio or standardized ratio of $NO_x$ to $NO_y$ concentrations, is used as a criterion to characterize the aging degree of air masses (Liang et al., 1998). Here the

metric $-\log(NO_x/NO_y)$ will be used to investigate the relationship between OA oxidation and the aging of air masses. The larger the metric is, the more aged the air mass is (Kleinman et al., 2008; Decarlo et al., 2008, 2010).

When $-\log(NO_x/NO_y) <0.1$, it is considered to be fresh air plume (Liang et al., 1998). The maxima of the parameter $-\log(NO_x/NO_y)$ were about 0.4, 0.4, 0.2, and 0.2 during seasonal observations, in accordance with the



photochemical reactions which were more active in spring and summer due to strong solar radiation, while in autumn and winter, the plumes of primary emissions had greater impacts. The urban Beijing was under the control of the relatively aged air masses during some periods (e.g., 23−24 Apr., 17−18 Aug., and 26−27 Oct. 2012). However, compared with the aerial results in Mexico City, where the highest -log ($NO_x/NO_y$) exceeded 1.4 (DeCarlo et al., 2010), the aging degree of air masses in urban Beijing were much lower.

The scatterplot of H/C vs. O/C ratios in Tri-VK-$\overline{OS_C}$ diagram (Fig. 8) was colored with -log ($NO_x/NO_y$). There was a trend that the oxidation state of OA was higher (i.e., higher O/C ratio and $\overline{OS_C}$) as the air mass aged more (from the upper left to the lower right). The OA factors resolved by AMS-PMF analysis during seasonal campaigns are also marked in Fig. 8. In the order from POA (HOA, COA, CCOA and BBOA) to SOA (LO-OOA, OOA and MO-OOA), the factors evolved along with the direction to a higher oxidation state, which is consistent with the oxidation characteristics of the factors.

The OA/ΔCO ratio is used to evaluate the contribution of SOA formation, where ΔCO is CO subtracted the regional background concentration (0.1 ppmv) to exclude the influences of emitted and transported OA (de Gouw and Jimenez, 2009). With the SOA formation, the OA/ΔCO ratio increases. The average of OA/ΔCO ratios (in the range of first and ninety-ninth percentile) were 37.0±30.7, 25.0±16.4, 20.3±16.5 and 15.1±8.1 μg m$^{-3}$ ppmv$^{-1}$, respectively. The ratios were comparable to or higher than the urban ΔPOA/ΔCO ratios (<15 μg m$^{-3}$ ppmv$^{-1}$), but were much lower than the OA/ΔCO ratios (70±20 μg m$^{-3}$ ppmv$^{-1}$) in aged urban air (de Gouw and Jimenez, 2009), implying that the contribution of POA was considerable in Beijing during seasonal campaigns, especially in autumn and winter.

Investigating the variations of OA/ΔCO ratios with the aging process of air masses can help reveal the formation rate of SOA and parameterize it in the model simulation (Dzepina et al., 2011). The scatterplots of OA/ΔCO ratios over O/C ratios during the seasonal observations in Beijing are shown in Fig. 9, colored with the metric -log ($NO_x/NO_y$). In laboratory and field studies on OA aging under strong oxidizing conditions, it was found that the OA/ΔCO ratios remain relatively stable at high O/C ratios with the increase of O/C ratios, because in the oxidation processes, organics obtain oxygen atoms but loss carbon atoms (DeCarlo et al., 2008, 2010). During seasonal campaigns in Beijing, the OA/ΔCO ratios showed different trends with the increase of O/C ratios as well as aging degrees of air masses (Fig. 9). Due to the intricate sources of OA in urban Beijing, the O/C ratios cannot get higher and the OA/ΔCO ratios cannot reach an obviously stable level. In autumn, the OA/ΔCO ratios decreased



with the increase of O/C ratios in more aged air masses, suggesting that reductive POA contributed substantially to high OA concentrations. An important reason for the decrease of OA/ΔCO ratios in aged air in autumn is the mixing of urban and biomass burning plumes with high CO content (DeCarlo et al., 2010). In summer and winter, along with the increase of O/C ratios and the aging degree of air masses, the OA/ΔCO ratios trended to increase,

indicating the material contribution of SOA formation. Guo et al. (2014) concluded that photochemical oxidations of VOCs from urban traffic emissions are primarily responsible for the secondary formation during severe aerosol pollution events. Many studies reported the rapid increase of SOA significantly exceeding the initial POA emission ratios in urban atmosphere in a day in the absence of biomass burning (DeCarlo et al., 2010).

In mass spectra measured by AMS, some relatively abundant fragment ions have certain representativeness.

As mentioned above, $m/z$ 43 and $m/z$ 44, mainly as $C_2H_3O^+$ and $CO_2^+$, respectively, are predominant in the MS of OOA. While in the MS of HOA $m/z$ 43 is mainly composed of alkyl fragment $C_3H_7^+$ (Ng et al., 2010). The difference between the relative contents of $m/z$ 43 and $m/z$ 44 in OA can reflect the oxidation degree too, e.g., the abundance of $m/z$ 44 in MO-OOA is higher than that in LO-OOA. The fragment $m/z$ 60 (mainly $C_2H_4O_2^+$) is often used as a tracer of biomass burning (Alfarra et al., 2007; Cubison et al., 2011).

The scatterplots of $f_{44}$ against $f_{43}$, and $f_{44}$ against $f_{60}$ during seasonal campaigns are shown in Fig. 10, colored with -log (NO$_x$/NO$_y$). The data points in the scatterplots of $f_{44}$ vs. $f_{43}$ substantially fell into the triangle derived by Ng et al. (2010). As the aging degree of air masses increased, OA showed the evolution trends moving from the lower right to the upper left generally in the triangle. The POA factors (HOA, COA, CCOA and BBOA) are concentrated in the bottom of the triangle, LO-OOA with a higher oxidation state is in an intermediate location,

and MO-OOA with the highest oxidation state is at the top of the triangle. Ng et al. (2010) found that with the enhancement of OA oxidation, data points gradually move upward from the lower half of the triangle through summarizing the results of chamber experiments and field observations.

In the scatterplots of $f_{44}$ vs. $f_{60}$, the majority of the data points concentrated on the left side and presented in a band-shaped region (Fig. 10e-h). Cubison et al. (2011) found that in the $f_{44}$ against $f_{60}$ space, data from biomass

burning appear in the lower right part, while data with negligible biomass burning influence are concentrated on the left side as a band shape. There are a part of data points appearing in the conceptual space for BBOA (Cubison et al., 2011), more obviously in spring, autumn and winter (Fig. 10e, g, h). In spring and autumn, a large number of data points are of higher $f_{60}$ than in other seasons, and the aging degrees of air masses were low, which may be



subject to biomass burning emissions as mentioned above. HOA, COA, CCOA, LO-OOA, MO-OOA and OOA are located on the left side, while BBOA appears in the right. With the increase of the aging degree of air masses, $f_{60}$ did not trend to change dramatically, suggesting that the contribution of biomass burning to OA has no dependence on the aging degree of air masses.

## 4. Conclusions

We investigated the compositions of submicron aerosols during four seasons in urban Beijing with highly time-resolved HR-ToF-AMS measurements. These conclusions can be drawn:

(1) The submicron aerosol pollution in Beijing was serious. The average $PM_1$ was highest (82 µg m$^{-3}$) in winter, resulting from intense emissions due to adverse meteorological conditions and secondary formation, and lowest (38 µg m$^{-3}$) in summer due to frequent rainout. Chemical compositions in $PM_1$ also showed seasonal dependence. Carbonaceous fraction (OA+BC) constituted more than 50% of $PM_1$ in autumn due to primary emissions, while SNA contributed 60% of $PM_1$ in other seasons.

(2) OA was identified as secondary fractions, MO-OOA and LO-OOA (or total OOA), and primary fractions COA, CCOA, BBOA, and HOA. In spring and summer, OOA dominated OA (63 and 69%) due to the crucial contribution of SOA formation; while POA was predominant (53%) in autumn and equivalent (50%) to SOA in winter. SOA contributed to OA much more significantly in winter (50%) than in previous studies (20−30%), which may be associated with stronger SOA formation under long-lasting static and heavy-polluted days.

(3) Secondary species (SNA+SOA) represented about 60−80% of $PM_1$ in four seasons, stressing the importance of measures targeting the reduction of gaseous precursor ($NO_x$, $SO_2$ and VOCs) emissions.

(4) OA was in a higher oxidation state in urban Beijing according to OA/OC and O/C ratios, and $\overline{OS_C}$, especially in summer and winter. The $\overline{OS_C}$ were -0.64, -0.54, -0.66 and -0.58 in four seasons, respectively. The evolution of OA was, along with the increase of oxidation state, from POA (HOA, COA, CCOA, and BBOA) to SOA (LO-OOA and MO-OOA). Meanwhile, higher oxidation states of OA were observed in more aged air masses.

In this study, the chemical compositions of $PM_1$ varied obviously in four seasons. Meteorological conditions and gaseous and particulate emissions determined the severity of atmospheric aerosol pollution. To prevent regional aerosol pollution effectively, further strengthening the control of primary particulate emissions is expected. In addition, the emissions of secondary species′ precursors must be reduced, especially in adverse meteorological



conditions.

## 5. Data availability

The data presented in this article are available from the authors upon request (minhu@pku.edu.cn).

**Acknowledgements**

5          This work was supported by the National Natural Science Foundation of China (91544214, 41421064), the National Basic Research Program of China (2013CB228503) and National Key Research and Development Program of China (2016YFC0202003). The authors appreciate Prof. M. Shao and Dr. Y. Yang in Peking University for data of VOCs and Prof. J. Morrow in the Prefectural University of Kumamoto for his assistance in the word and grammar editing.



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





Table 1 Seasonal variations of main chemical components in $PM_1$ and gaseous pollutants, and meteorological conditions (temperature, T; relative humidity, RH; wind speed, WS; pressure, P). The data are listed in the form of "average±standard deviation". Units of particulate and gaseous pollutants are ppbv and $\mu g\ m^{-3}$, respectively.

|  | Spring Mar. 30–May 7 2012 | Summer Jul. 29–Aug. 29 2012 | Autumn Oct. 13–Nov. 13 2012 | Winter Jan. 23–Mar. 2 2013 |
|---|---|---|---|---|
| $PM_1$ | 45.1±45.8 | 37.5±31.0 | 41.3±42.7 | 81.7±72.4 |
| OA | 14.0±11.3 | 12.5±7.8 | 18.2±17.5 | 29.7±25.7 |
| MO-OOA | 4.3±6.3 | 3.3±2.9 | 8.6±8.4 | 9.8±8.5 |
| LO-OOA | 4.6±3.6 | 5.3±3.6 |  | 5.0±6.9 |
| COA | 3.7±3.8 | 2.5±2.1 | 5.2±7.3 | 4.3±4.4 |
| HOA | 1.4±1.6 | 1.4±1.1 | 2.5±2.9 | 5.5±5.7 |
| BBOA |  |  | 2.0±2.7 |  |
| CCOA |  |  |  | 5.0±7.2 |
| $SO_4^{2-}$ | 9.3±11.5 | 9.7±9.9 | 5.5±7.3 | 17.4±17.9 |
| $NO_3^-$ | 10.2±13.6 | 6.4±8.2 | 7.9±10.3 | 16.2±15.4 |
| $NH_4^+$ | 7.3±9.0 | 5.4±5.8 | 4.5±5.7 | 11.7±10.8 |
| $Cl^-$ | 1.2±1.5 | 0.4±0.6 | 2.0±2.7 | 2.8±3.1 |
| BC | 3.1±2.3 | 3.2±2.0 | 3.2±2.8 | 3.9±3.6 |
| $SO_2$ | 7.4±7.0 | 3.1±3.3 | 9.2±8.8 | 32.0±21.2 |
| CO | 636±568 | 671±317 | 1229±1139 | 2224±1844 |
| NO | 10.5±21.0 | 3.5±6.6 | 37.5±44.7 | 24.4±30.9 |
| $NO_2$ | 29.3±17.7 | 23.2±10.4 | 33.8±19.4 | 36.8±19.3 |
| $O_3$ | 28.3±20.6 | 41.6±34.7 | 11.6±11.5 | 13.1±13.0 |
| T (°C) | 16.8±4.8 | 26.8±4.0 | 11.2±4.8 | -0.1±4.0 |
| RH (%) | 33.7±23.1 | 61.4±16.9 | 43.5±19.9 | 37.5±20.3 |
| WS ($m\ s^{-1}$) | 2.3±1.5 | 1.8±0.9 | 1.0±0.9 | 2.0±1.5 |
| P (hPa) | 1004.2±5.6 | 1001.6±3.4 | 1012.6±3.8 | 1019.2±5.3 |





Table 2 Elemental ratios and OA/OC ratios in OA obtained from field observations at urban and rural/suburban sites. The ratios are corrected by the "improved-ambient" method (Canagaratna et al., 2015), except for Hu et al. (2013) and Gong et al. (2013).

| Sites | Site types | Periods | O/C | H/C | $\overline{OS_C}$ | OA/OC | References |
|---|---|---|---|---|---|---|---|
| Beijing (China) | Urban | Mar.–May 2012 (sp) | 0.49 | 1.63 | -0.64 | 1.81 | This study |
| | | Jul.–Aug. 2012 (su) | 0.53 | 1.61 | -0.54 | 1.88 | |
| | | Oct.–Nov. 2012 (a) | 0.46 | 1.58 | -0.66 | 1.77 | |
| | | Jan.–Mar. 2013 (w) | 0.47 | 1.52 | -0.58 | 1.79 | |
| | | Aug.–Sept. 2011 (su) | 0.56 | 1.61 | -0.49 | 1.91 | Hu et al., 2016a |
| | | Nov.–Dec. 2010 (w) | 0.32 | 1.65 | -1.01 | 1.58 | |
| | | Jul.–Sept. 2008 (su) | 0.41 | 1.63 | -0.80 | 1.69 | Huang et al., 2010 |
| Shenzhen (China) | | Oct.–Dec. 2009 (a) | 0.39 | 1.83 | -1.04 | 1.71 | He *et al.*, 2011 |
| | | Aug.–Sept. 2011 (su) | 0.45[a] | 1.74[a] | -0.84 | 1.81[a] | Gong et al., 2013 |
| Shanghai (China) | | May–Jun. 2010 (su) | 0.40 | 1.92 | -1.12 | 1.69 | Huang *et al.*, 2012 |
| Mexico City (Mexico) | | Mar. 2006 (sp) | 0.53 | 1.82 | -0.77 | 1.86 | Aiken *et al.*, 2009 |
| Bologna (Italy) | | Mar.–Apr. 2008 (sp) | 0.59 | 1.64 | -0.46 | 1.92 | Saarikoski *et al.*, 2012 |
| Fresno, CA (US) | | Jan. 2010 (w) | 0.35 | 1.75 | -1.05 | 1.63 | Ge et al., 2012 |
| Riverside, CA (US) | | Jul.–Aug. 2005 (su) | 0.44 | 1.71 | -0.82 | 1.73 | Docherty et al., 2011 |
| Kaiping (China) | Rural/Suburban | Oct.–Nov. 2008 (a) | 0.60 | 1.64 | -0.44 | 1.94 | Huang et al., 2011 |
| Heshan (China) | | Nov.–Dec. 2010 (a) | 0.50 | 1.63 | -0.63 | 1.87 | Gong et al., 2012 |
| Changdao Island (China) | | Mar.–Apr. 2011 (sp) | 0.75[a] | 1.48[a] | 0.02 | 2.08[a] | Hu et al., 2013 |
| Ziyang (China) | | Dec. 2012–Jan. 2013 (w) | 0.65 | 1.56 | -0.26 | 2.02 | Hu et al., 2016b |
| Jiaxing (China) | | Jun.–Jul. 2010 (su) | 0.36 | 1.94 | -1.22 | 1.67 | Huang et al., 2013 |
| | | Dec. 2010 (w) | 0.43 | 1.73 | -0.87 | 1.75 | |
| Melpitz (Germany) | | May–Jul. 2008 (su) | 0.52 | 1.51 | -0.47 | 1.83 | Poulain et al., 2011 |
| | | Sept.–Nov. 2008 (a) | 0.54 | 1.48 | -0.40 | 1.84 | |
| | | Feb.–Mar 2009 (w) | 0.53 | 1.48 | -0.41 | 1.83 | |

[a] The O/C, H/C and OA/OC ratios were scaled up by 27, 11 and 7% (the average corrections for ambient OA; Canagaratna et al., 2015), respectively.





Table 3 Slopes in the van Krevelen diagram based on field observations at urban and background/suburban sites.

| Sites | Site types | Periods | Slope | Intercept | $r^2$ | O/C ranges | References |
|---|---|---|---|---|---|---|---|
| Beijing (China) | Urban | Mar.–May 2012 | -0.57 | 1.91 | 0.74 | 0.12−0.95 | This study |
|  |  | Jul.–Aug. 2012 | -0.62 | 1.94 | 0.79 | 0.20−0.83 |  |
|  |  | Oct.–Nov. 2012 | -0.67 | 1.90 | 0.70 | 0.15−0.92 |  |
|  |  | Jan.–Mar. 2013 | -0.08 | 1.56 | 0.02 | 0.18−0.77 |  |
| Mexico City (Mexico) |  | Mar. 2006 | -0.69 | 2.19 | 0.99 |  | Aiken et al., 2009 |
| Fresno (US) |  | Jan. 2010 | -0.95 | 2.08 | 0.77 |  | Ge et al., 2012 |
| Riverside (US) |  | Jul.–Aug. 2005 | -0.96 | 2.13 | 0.81 |  | Docherty et al., 2011 |
| Lake Hongze (China) | Background /suburban | Mar.–Apr. 2011 | -0.72 | 2.00 | 0.56 |  | Zhu et al., 2016 |
| Mount Wuzhi (China) |  | Mar.–Apr. 2015. | -0.69 | 1.99 | 0.54 |  |  |
| Ziyang (China) |  | Dec. 2012–Jan. 2013 | -0.44 | 1.84 | 0.70 | 0.12−0.70 | Hu et al., 2016b |
| Melpitz (Germany) |  | May–Jul. 2008 | -0.69 | 1.83 | 0.91 |  | Poulain et al., 2011 |

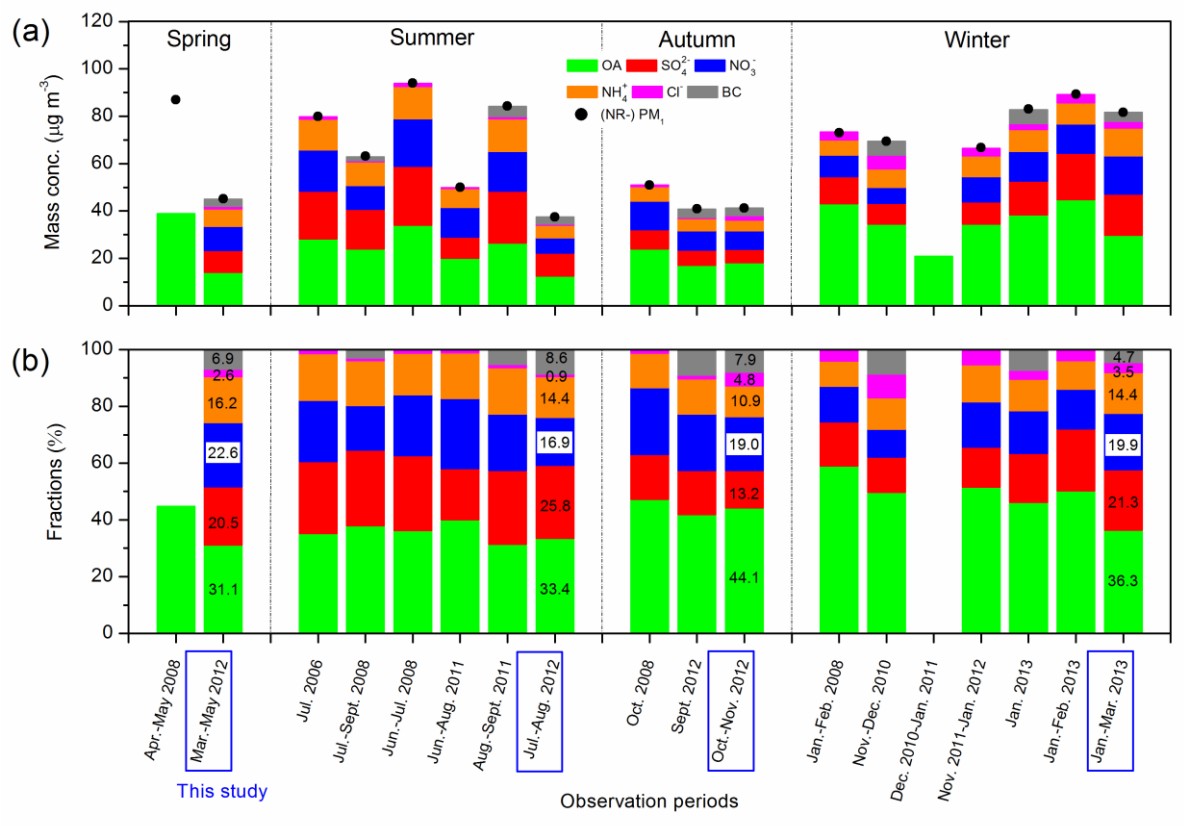

Figure 1. Concentrations (a) and fractions (b) of main chemical components in PM$_1$ during seasonal observations in Beijing in recent years. The data and references are available in Table S4 in the supplement.



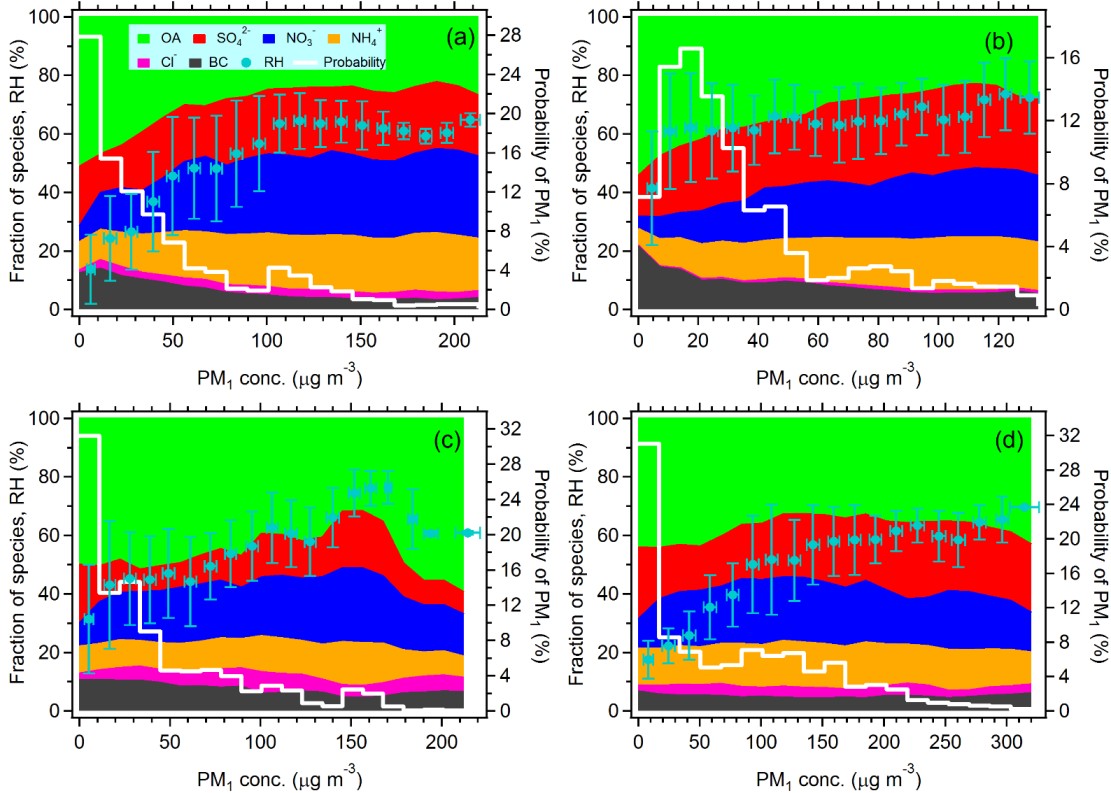

Figure 2. Fractions of main chemical compositions in submicron aerosols at different $PM_1$ levels, and the probability density of $PM_1$ (white curves) during the spring (a), summer (b), autumn (c) and winter (d) observations. The average RH values in each segment are illustrated and the error bars in the x- and y-axis were the standard deviations of $PM_1$ and RH, respectively.



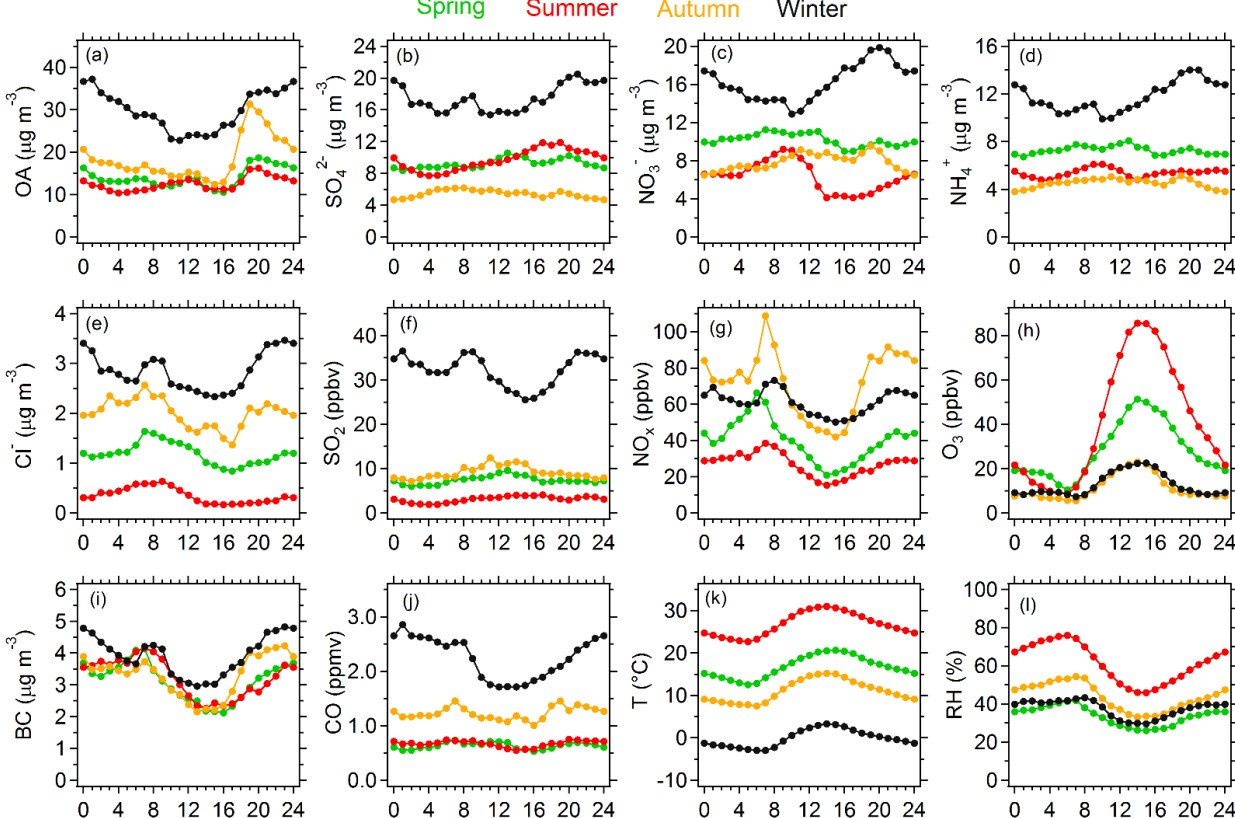

Figure 3. Diurnal patterns of chemical species of PM$_1$, gaseous pollutants, temperature (T) and relative humidity (RH) during seasonal observations.





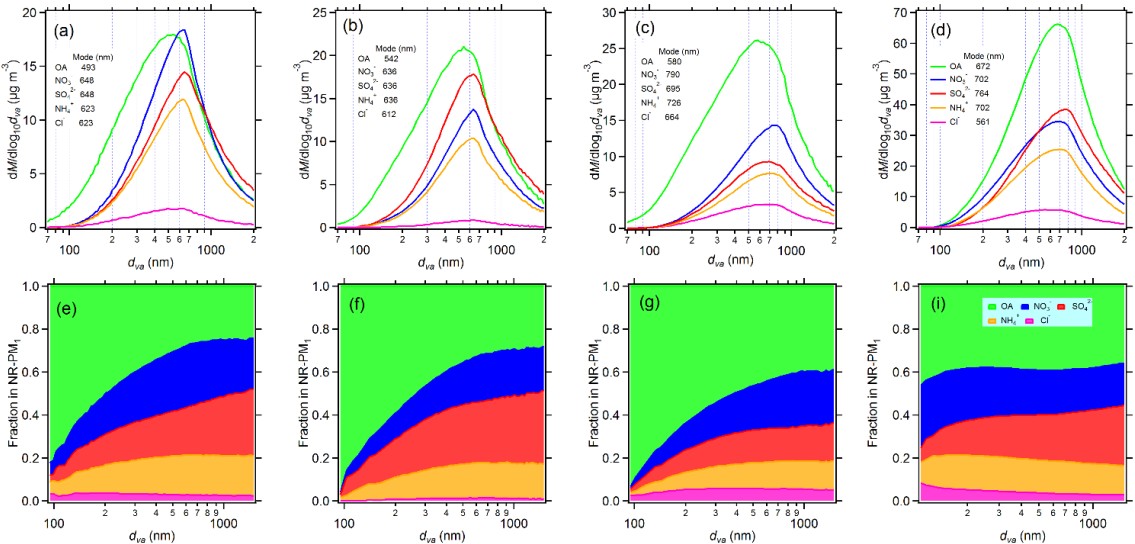

Figure 4. Mass size distributions of chemical compositions in PM$_1$ during the spring (a, e), summer (b, f), autumn (c, g) and winter (d, h) observations.





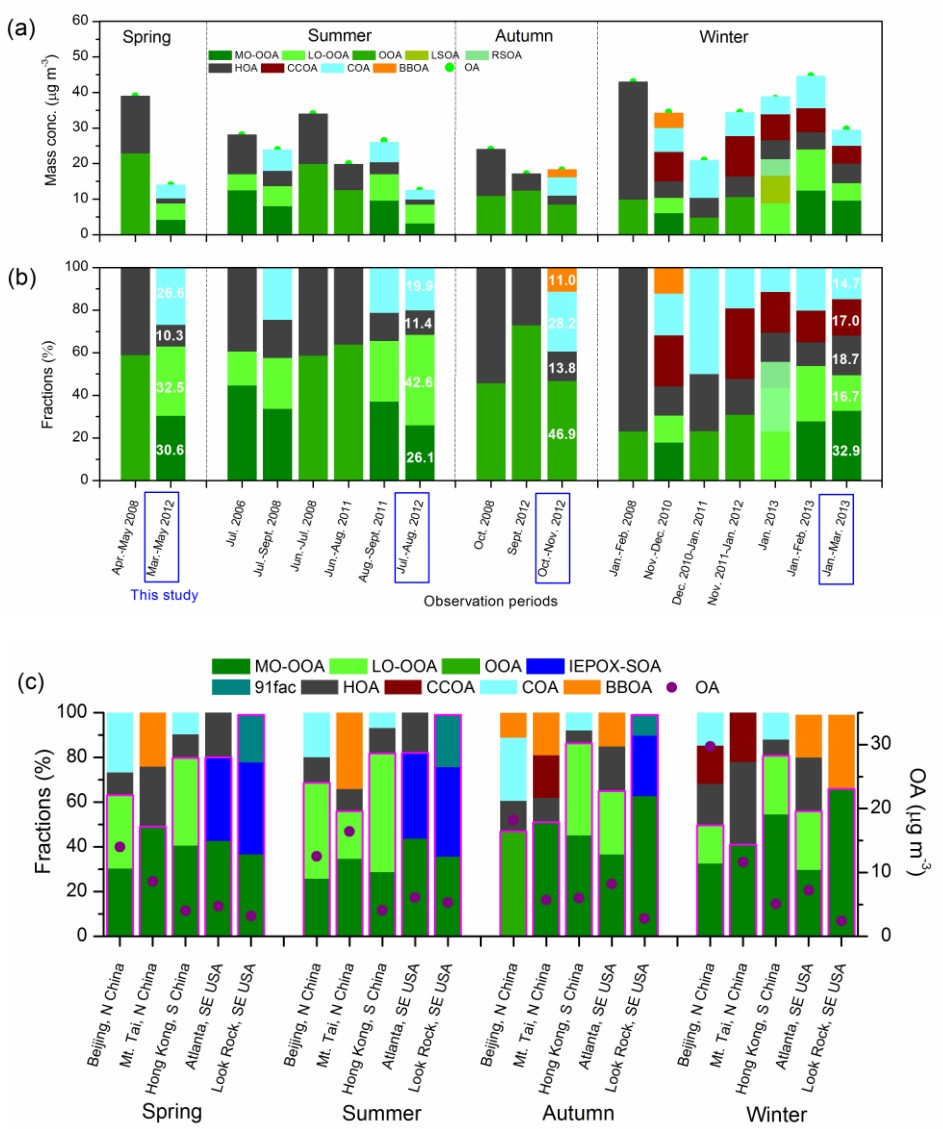

Figure 5. Resolved components of OA during seasonal observations in Beijing (a, b; data and references are listed in Table S5 in the supplement) and at other sites (c) in Mt. Tai (Zhang et al., 2014), Hong Kong (Li et al., 2015) and southeastern USA (Budisulistiorini et al., 2016) in recent years. LSOA, local SOA; RSOA, regional SOA; IEPOX-SOA, isoprene-epoxydiols-derived SOA; 91fac, a biogenically influenced factor characterized by distinct $m/z$ 91. MO-OOA and LO-OOA are identified as LV-OOA and SV-OOA, respectively, except for during Aug.–Sept. 2011 and Nov.–Dec. 2010 (Hu et al., 2016a), and in this study.





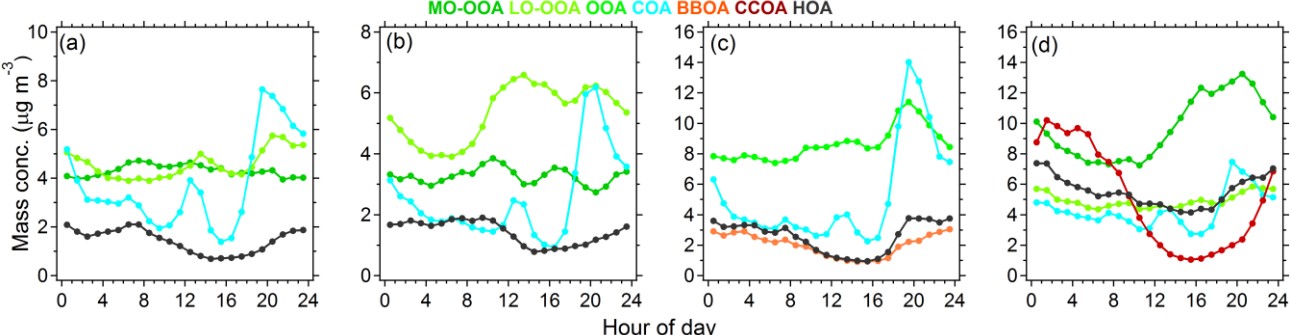

Figure 6. Diurnal variations of OA components in spring (a), summer (b), autumn (c) and winter (d) observations.

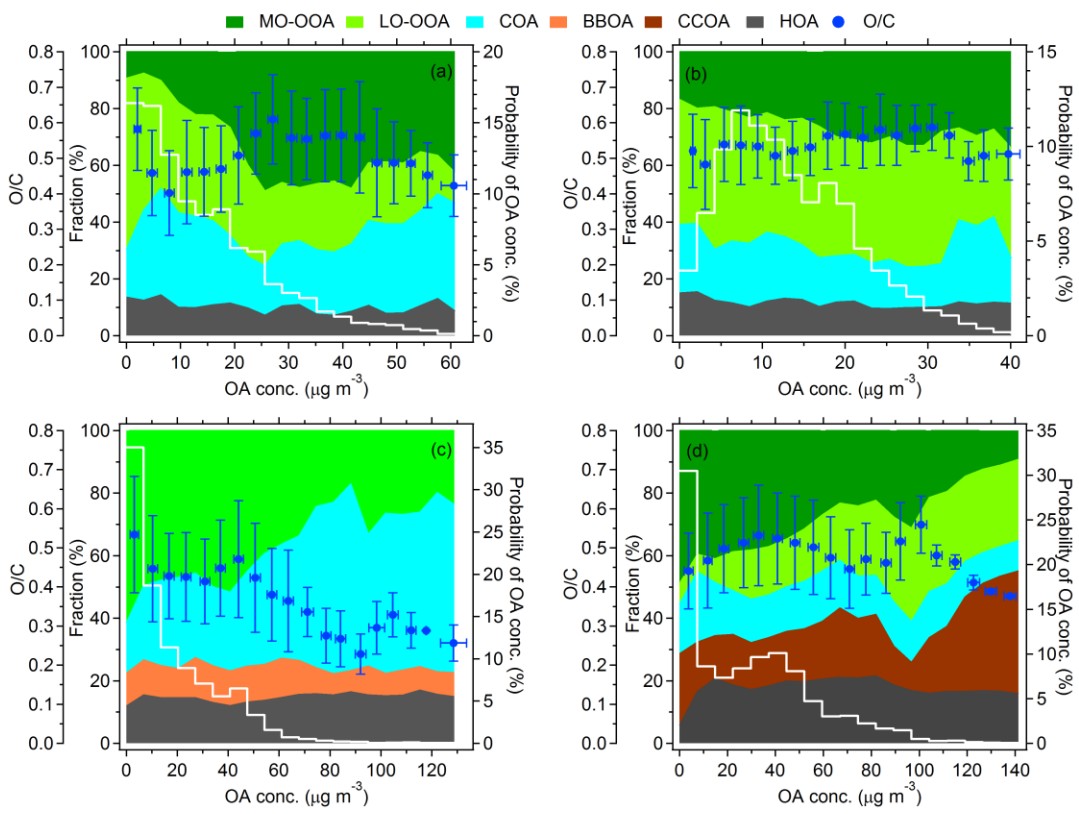

Figure 7. Fractions of OA components at different OA concentration levels and the probability distributions of OA concentrations (white curves) during the spring (a), summer (b), autumn (c) and winter (d) observations. The average O/C ratios in each segment are illustrated and the error bars in the x- and y-axis were the standard deviations of OA concentration and O/C ratios, respectively.



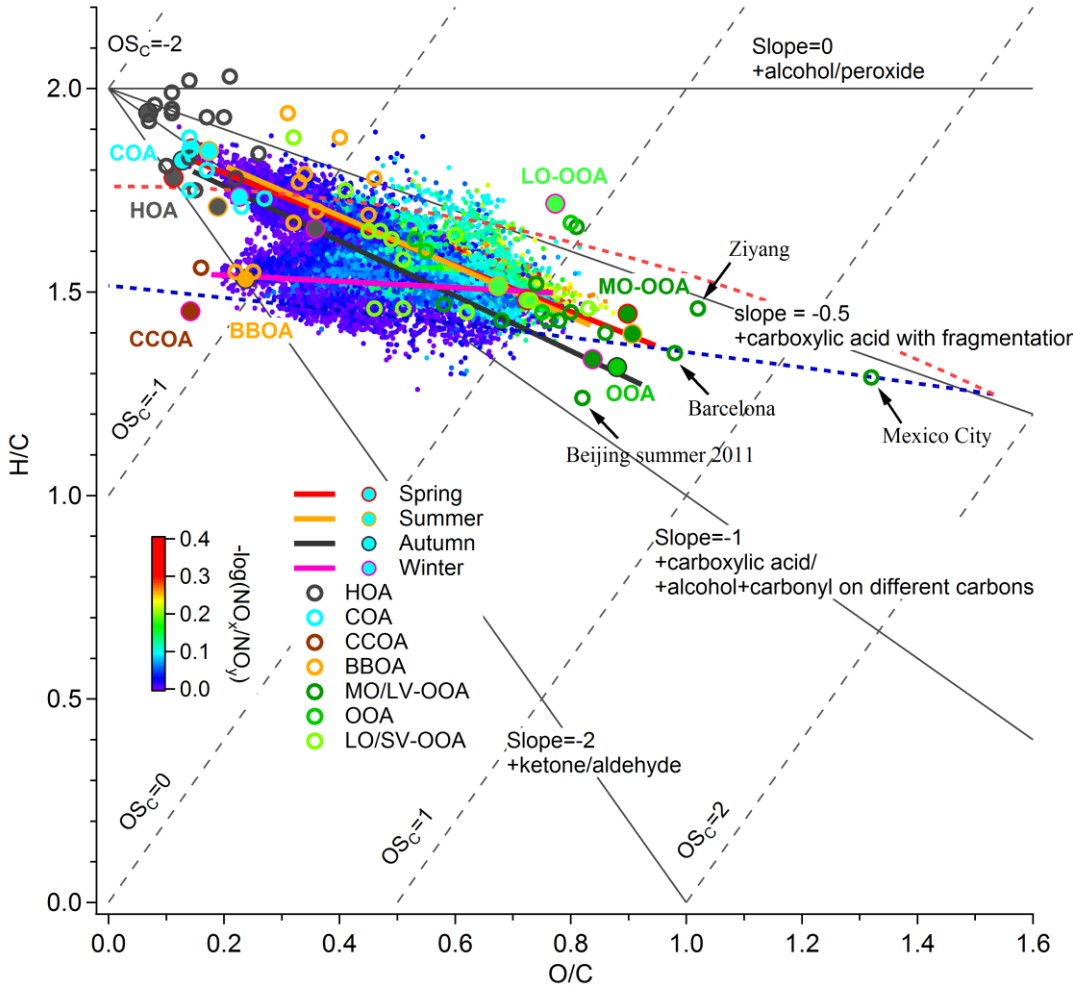

Figure 8. Tri-VK-$\overline{OS_C}$ diagram based on seasonal observations. The scatterplot of H/C vs. O/C ratios is colored by the parameter -log (NO$_x$/NO$_y$). The OA factors resolved by AMS-PMF analysis are also marked in the diagram. The majority of the data fall into colored triangle lines (Ng et al., 2011). Improved-ambient results for OA factors are from Hu et al. (2016a), Hu et al. (2016b) and the summarization of Canagaratna et al. (2015).





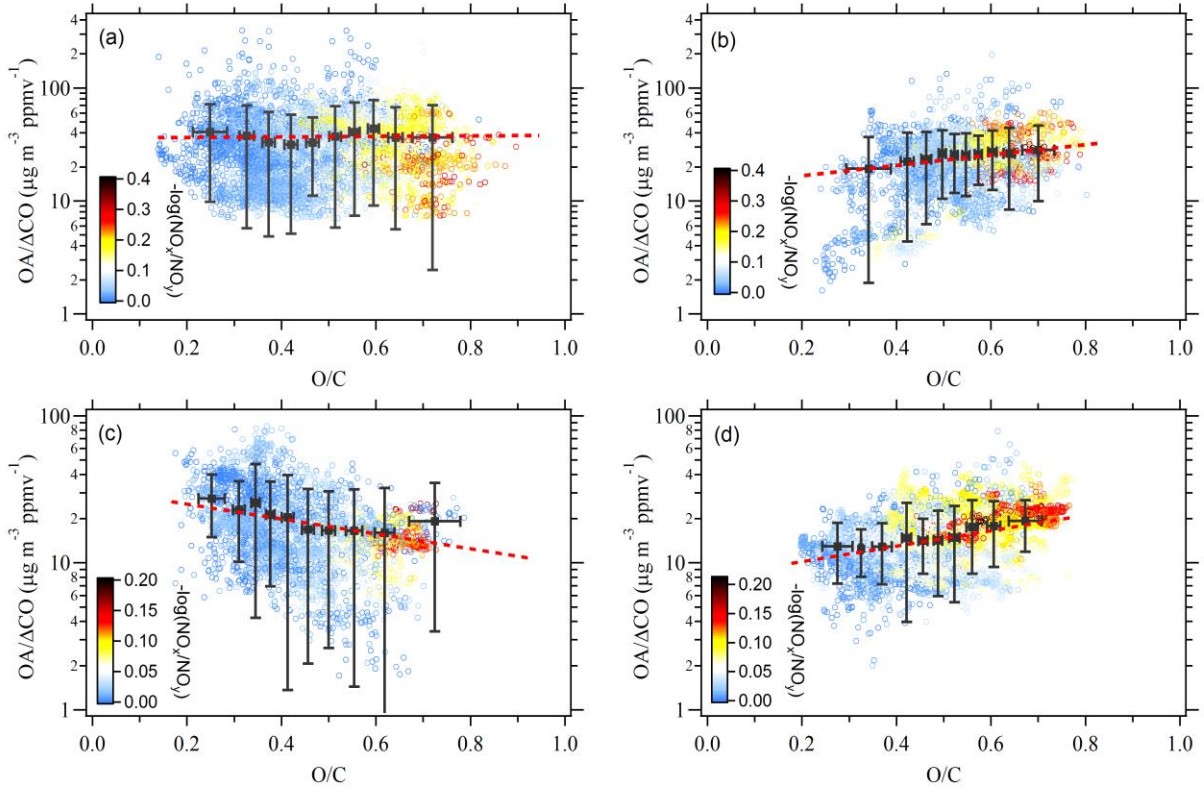

Figure 9. Scatterplots of OA/ΔCO (ΔCO=CO-0.1) ratios vs. O/C atomic ratios in OA during the (a) spring, (b) summer, (c) autumn and (d) winter observations. The OA/ΔCO ratios are in the range of first and ninety-ninth percentile. The scatterplots are colored by the parameter -log ($NO_x/NO_y$).



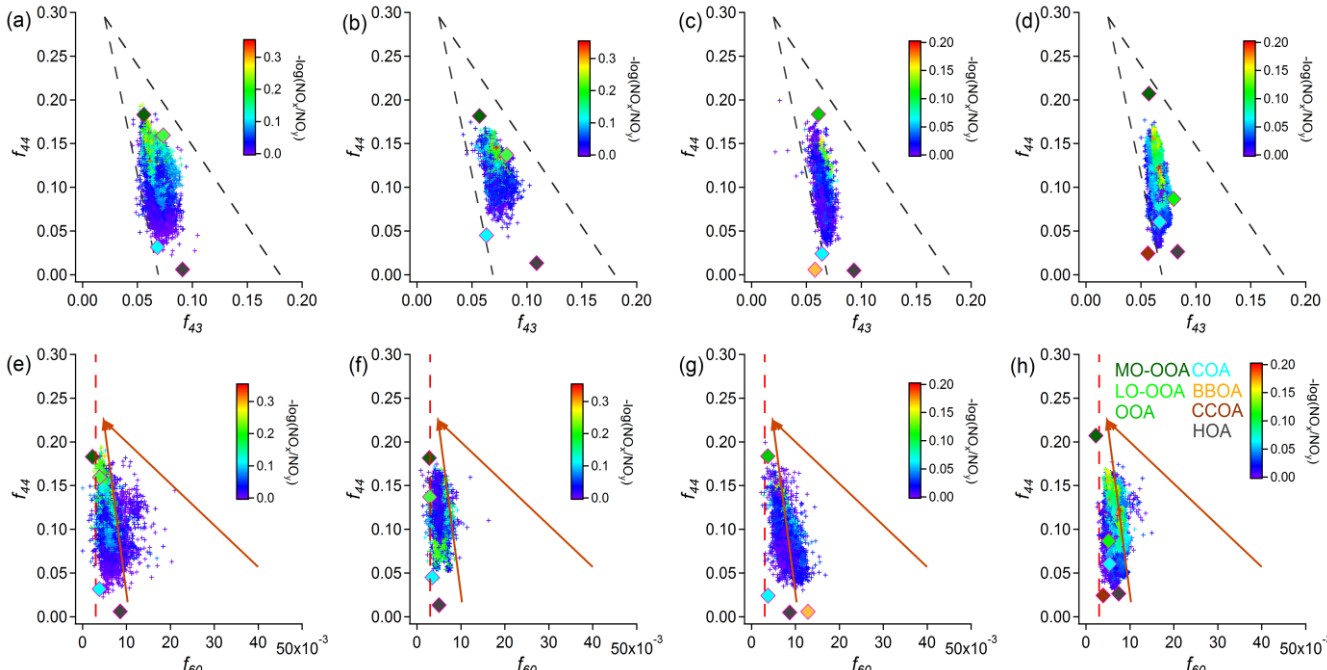

Figure 10. Scatterplots of $f_{44}$ vs. $f_{43}$ and $f_{44}$ vs. $f_{60}$ in OA during the spring (a, e), summer (b, f), autumn (c, g) and winter (d, h) observations. The triangle for $f_{44}$ against $f_{43}$ derived by Ng et al. (2010) is lineated. The conceptual space for BBOA and the nominal background value at 0.3% (Cubison et al., 2011) are marked by arrows and vertical dash lines, respectively. The scatterplots are colored by the metric $-\log(NO_x/NO_y)$. The OA factors resolved by AMS-PMF analysis are also marked in all diagrams.