# Peer review of "Seasonal variations of high time-resolved chemical compositions, sources and evolution for atmospheric submicron aerosols in the megacity Beijing"

_Atmospheric Chemistry and Physics, 2017_

## Referee Comment (RC1) · Anonymous Referee #1 · 23 Mar 2017

Review of

**"Seasonal variations of high time-resolved chemical compositions, sources and evolution for atmospheric submicron aerosols in the megacity Beijing"**

By Hu et al.

General comments

This study presents high time resolution chemical characterization of ambient submicron aerosols measured in Beijing between March 2012 and March 2013 using HR-ToF-AMS. Seasonal comparison of the chemical composition shows that organic aerosol (OA) and inorganic species (i.e. sulfate, nitrate, and ammonium) contributes equally towards total $PM_1$ mass. The chemical component composition and diurnal trend were impacted by atmospheric conditions of each season. Relationship between the atmospheric conditions to $PM_1$ composition need to be further discussed and/or clarified as outlined in the specific comments.

Additionally, sources of OA fraction in Beijing were identified using PMF. Primary OA (i.e. HOA and COA) and secondary OA sources were observed in all seasons, whereas BBOA and CCOA were only observed in autumn and winter, respectively. It is interesting that CCOA factor was resolved in winter dataset indicating significant contribution of coal combustion from domestic heating to ambient submicron aerosol. There seems to be an unresolved factor in winter dataset that could be a BBOA factor. Residual mass spectra should be added as supplementary information for evaluating the PMF solution.

Overall, this study falls within the scope of Atmospheric Chemistry and Physics journal. The results contribute to long-term and seasonal evaluation of air quality and can support air pollution prevention policy in Beijing. The manuscript is written pretty well with only some minor issues as outlined in the technical comments. I support publication of this manuscript after minor revisions.

Specific comments

1. $PM_1$ mass concentration is estimated as a sum of NR-$PM_1$ measured by AMS and BC measured by Aethalometer. However, cutoff size of the AMS's inlet ($PM_1$) is different from the Aethalometer ($PM_{2.5}$). How did you consider the cutoff size when summing the measurements? This information can be added into the experimental section.

2. Pg 6 Lns 9-11: What are considered in the certain atmospheric conditions? How do these conditions relate to the current study?

3. Pg 8 Lns 8-10: "Affected by different meteorological conditions, e.g., solar radiation, temperature, RH, boundary layer, and mountain-valley breeze in summer (Fig. S7), as well as different emission sources, the chemical compositions in $PM_1$ showed distinct diurnal patterns in four seasons." The beginning of sentence implies that the meteorological conditions specific for summer, whereas the ending suggests the distinctive diurnal patterns in four seasons. I think meteorological conditions in four seasons, not only summer, will affect the chemical composition and diurnal patterns in each season. This needs to be clarified.

4. Pg 8 Ln 19: "Flatter diurnal cycles of sulfate were observed in four seasons, …". What does "flatter" here compare with? Does it compare with OA diurnal pattern? Also, add discussion about the diurnal patterns in spring and summer. Sulfate in spring showed small peaks around 13:00 and 20:00, whereas in summer, it peaked at evening only. What could possibly drive changes in sulfate diurnal patterns?

5. Pg 9 Lns 9-11: I think nitrate diurnal pattern in autumn is still significant; nitrate increased in the morning and then later in the evening. Also, what is the evidence of combined effect of sulfate and nitrate to ammonium diurnal pattern? Maybe you can correlate the diurnal variations and provide the coefficient of determination.

6. Pg 10 Ln 3: The size distribution of $NH_4$ is pretty similar to $NO_3$ in all four seasons. The size difference is not obvious in Fig. 4. To help comparison, you can consider increasing the size of the figure (panel for each season), or providing a summary table of size distribution in the SI section.

7. Pg 11 Lns 23-24: Provide reference(s) that show the role of emission reduction of gaseous precursors to SOA formation. For example: Pye, et al. Epoxide pathways improve model predictions of isoprene markers and reveal key role of acidity in aerosol formation. Environ. Sci. Technol. 2013, 47, 11056-11064.

8. Pg 14 Lns 18-20: Are you implying that there is a factor (BBOA) that is not resolved by PMF in the winter dataset? What may cause PMF can't resolve BBOA factor in winter? Is the residual still showing important mass spectrum (e.g. *m/z* 60, 73) and time series features? Diagnostic plots of PMF winter dataset analysis (see Zhang et al. 2011) will be useful for readers to understand the analysis process. These diagnostic plots can be added in the SI section and referred in the main text. In addition to winter, diagnostics for other season should be provided in the SI section to help reader understand the PMF analysis process.

9. Pg 17 Lns 7-8: It is interesting that the LO- and MO-OOA fractions slightly increased around 100 µg/m$^3$ only. What influence the increase of OOA factors during particular OA mass concentration?

10. Pg 18 Lns 27-28: The VK diagram in winter does not show any correlation ($r^2$ = 0.02) between H/C and O/C. The slope is nearly zero (0.08; not broad). This suggests that hydroxylation and/or peroxidation processes were likely to occur in Beijing during winter. On the other hand, carboxylation process might be unlikely to occur since the slope is higher than -0.5. Please elaborate or clarify the discussion.

Technical comments
1. Pg 1 Ln 14: If $PM_1$ here is NR-$PM_1$+BC, it needs to be specified in the sentence.
2. Pg 2 Lns 26-27: Is SNA important only for the urban area or in general? Clarify the sentence and add reference.
3. Pg 5 Lns 7: …, the last well calibrated…
4. Pg 5 Lns 8-9: Do you mean "the size distribution calibration"? Please clarify.
5. Pg 7 Lns 1-2: Do you mean "…$PM_1$ in winter was higher than previous studies in Beijing…"? Also please clarify what does "close periods" mean.
6. Pg 7 Ln 13: Figures S6 and S8 don't display fire points as mentioned in the sentence.

7. Pg 9 Ln 27: …the internally mixed states…
8. Pg 11 Ln 8: The term "freshly secondary formation" is ambiguous. Consider "new SOA formation".
9. Pg 12 Ln 26: Provide the coefficient of determination for any statement about correlation.
10. Pg 15 Lns 24-27: Insert respectively after on average. Breaking this sentence into two will be better as it is pretty long at the current state.
11. Pg 18 Ln 2: …, 1.52-1.63, …
12. Pg 18 Ln 18: What does "shallower" mean? Do you mean flatter slope?
13. Pg 20 Ln 7: …(i.e. lower H/C ratio, and higher O/C ratio and $\overline{OS_C}$)….
14. Pg 20 Ln 28: What does "higher" compare to? Is it higher than "stable level". What does stable level mean here?
15. Pg 21 Ln 28: Provide indicator (value) of the low aging degrees of air masses.
16. Fig 3: Need label (Hours of Day) for the X-axis.
17. Figs 5-8: Colors for MO-OOA and OOA are difficult to distinguish on print.
18. Figs 6-7: The legends for these figures are not consistent. OOA is missing from Fig. 7.
19. Fig. S13: … LV-OOA, MO-OOA…

References

Pye, H. O. T. Epoxide pathways improve model predictions of isoprene markers and reveal key role of acidity in aerosol formation. Environ. Sci. Technol., 47, 11056-11064, 2013.

Zhang, Q. Q. Understanding atmospheric organic aerosols via factor analysis of aerosol mass spectrometry: a review, Analyt. Bioanalyt. Chem., 401, 3045-3067, 2011.

---

## Referee Comment (RC2) · Anonymous Referee #2 · 24 Mar 2017

The manuscript investigates the seasonal variations of compositions, sources, and evolution for atmospheric aerosols in the megacity of Beijing. While field measurements have been carried out by other groups in this city, this work provides a more comprehensive exploration of organic aerosol composition and evolution. Generally, the paper is quite well written. The methodology and results are presented clearly. The measurements provide sufficiently new data that the manuscript may merit publication. However, the qualitative interpretation of the results may limit the application and impacts of this work. This calls for a more detailed data analysis, explaining the mechanisms behind the observations. I suggest the authors respond to the following suggestions before the paper is accepted to publication.

[Figure]

Major Comment:

My main question is that as mentioned by the authors, the characteristics and evolution of aerosol pollution are multifactorial (e.g. meteorological conditions, regional transport, and local sources). These factors can have distinct patterns in different seasons, thus the formation, transformation and removal of pollutants are affected. The authors have provided nice discussions on the sources and evolution of ambient atmospheric aerosols under the influence of different meteorological conditions and sources. However, what would be the impacts of regional or long range transport on the composition, formation and evolution of atmospheric aerosols in different seasons in Beijing? For example, the authors mentioned that regional transport can play a role in the sulfate concentration in the winter season. While regional transport impacts have briefly discussed or mentioned in various sections or paragraphs, it would be nice to have a more coherent, quantitative discussion on the role of regional (or long range) transport in the concentration and evolution of different inorganic and organic components in different seasons.

Minor comments:

Abstract, "The evolution process of OA in different seasons was investigated with multiple metrics and tools. The average carbon oxidation states and other metrics show that the oxidation state of OA was the highest in summer, probably due to both strong photochemical and aqueous-phase oxidations." Any evidence supports the contribution of aqueous-phase oxidation to OA? What is the relative contribution of photochemical vs. aqueous-phase oxidation to the concentration and evolution of OA?

Page 6, "In spring, summer and winter, SNA accounted for about 60% in PM1 due to the secondary aerosol formation through strong photochemical and aqueous-phase reactions." Like OA, what is the relative contribution of photochemical vs. aqueous-phase oxidation to the concentration and evolution of SNA?

Page 6, "Wang et al. (2016) and Cheng et al. (2016) found that high levels of sulfate

and fine PM can be explained by reactive aqueous oxidation of SO2 by NO2 under certain atmospheric conditions." Please elaborate what the meaning of "certain atmospheric conditions". How frequent these atmospheric conditions occur during the field campaign?

Page 6, "Compared with the previous results in Beijing (Fig. 1), PM1 in summer was lower than before, which likely resulted from the more effective rainout (Fig. S7) and lower concentrations of gaseous precursors (Table 1)." What are the frequency of precipitation/rainout and the concentration of gaseous precursors reported in previous studies?

Page 7, "While, PM1 ranged much more broadly, with the highest concentrations of over 200 or 300 $\mu$g m-3, resulting from accumulated pollutants under extremely unfavorable meteorological conditions or strong primary emissions." What is "the extremely unfavorable meteorological conditions"?

Page 7, "The proportions of nitrate increased more significantly, and the nitrate concentration increased rapidly under higher RH (Fig. 2; Pearson correlation coefficients r=0.71, 0.34, 0.49 and 0.79, p<0.01). These results indicate that the aqueous reactions could contribute to nitrate remarkably in highly humid and static air." What are the aqueous reactions referring to? Aerosol phase reactions, in-cloud reactions or both? Ambient RH is a good peroxy for indicting the occurrence and importance of aqueous phase reactions. However, what the physical state of the ambient aerosols (e.g. aqueous or solid)? What is the aerosol water content inferred or predicted from the aerosol speciation data and meteorological conditions? Will the water content and physical state of the aerosols play a role in the aqueous phase reactions and nitrate formation? If the aqueous phase reactions involve in-cloud reactions, any data (e.g. cloud coverage) can be used to support the importance of in-cloud reactions to the formation of nitrate and organic compounds during the campaign?

Page 8, "The peak concentration of OA in the evening in autumn was about two times

higher than in spring and summer, consistent with the results in Oct.-Nov. 2011 (Sun et al., 2015), possibly because of the more intense cooking activities." Please elaborate why there is more instance cooking activity in the evening in autumn, but not in other seasons?

Page 11, "LO-OOA dominated OA in summer (44%) due to the freshly secondary formation from strong photochemical oxidations; whereas, MO-OOA was dominant in OA in winter (33%), maybe because the air masses were more aged on heavy-polluted days." Please elaborate why the air masses are more aged on the heavy-polluted days in winter, but not in other seasons?

Page 13, "Fewer cooking activities during and around the Chinese New Year holiday (7-19 Feb.; Fig. S17), as well as the lower evaporation rate of oil, led to the lower concentration and proportion of COA in winter." This argument is interesting. How does the evaporate rate of oil depend on the temperature? What the contribution of this evaporation process to the total volatile organic compounds generated/originated from the cooking activities?

Page 16" In summer, OOA showed obvious diurnal variations: MO-OOA peaked in the morning and afternoon; LO-OOA showed two pronounced peaks at noon and at night, which was likely influenced by the photochemical oxidations and aqueous-phase formation from POA." It is not clear why the formation of LO-OOA is likely influenced by the photochemical oxidations and aqueous-phase formation from POA. Any other processes that can be contributed to the formation and transformation of LO-OOA?

Page 16, " the concentration and proportion of LO-OOA increased significantly in the afternoon (12:00-16:00), up to 7 ug m-3 and 50%, respectively (Fig. 6b), suggesting that LO-OOA was a strong local/regional photochemical product despite the much higher PBL in the daytime (Hu et al., 2016a)" What is the contribution of regional transport to LO-OOA (and other components)?

Page 17, "In both autumn and winter, the fractions of OOA slightly increased around

100 ug m-3, implying that POA probably transformed to SOA more effectively within this range." Please elaborate why the POA is more likely transformed to SOA under these conditions. What mechanisms or pathways?

Page 18, "In spring, summer and autumn, the slopes fell between -1 (the addition of carboxyl functional groups without fragmentation or carbonyl and hydroxyl in different carbons) and -0.5 (carboxyl functionalization with fragmentation)." In addition to overall oxidation pathways, what other information we could learn from the reported slopes between -1 and -0.5? Are these pathways consistent with the reactions we expected for the formation and chemical transformations of ambient aerosols in Beijing?

Page 19, "In winter, the scatterplot of H/C vs. O/C ratios in the VK Diagram showed "broader" slopes, hinting the more complex sources and evolution processes of OA. The scatterplot indicated that OA in winter mainly evolved between the hydroxylation or peroxidation reactions (slope = 0) and carboxyl groups addition with fragmentation (slope = -0.5)." Any hypothesis or explanation for the complex sources and evolution process of OA in the winter. Why these processes have not been observed or suggested in the other seasons?

Page 19, "compared with the oxidation states of SV-OOA, OOA and LV-OOA summarized by Canagaratna et al. (2015), the oxidation states of OOA in Beijing were generally higher than in other areas, especially for LO-OOA (Fig. 8). The oxidation states of MO-OOA in Beijing were only lower than those in very aged air masses in Ziyang in the basin (Hu et al., 2016b), over Mexico City (DeCarlo et al., 2010) and in Barcelona (Mohr et al., 2012). The oxidation states of LO-OOA were only slightly lower than those of MO-OOA in Beijing, and were comparable to those of MO-OOA in other urban areas (Fig. 8)." It is nice to compare the data with those collected at different locations. However, without explaining the causes/reasons for the differences would not be useful for the readers to understand the formation and evolution of atmospheric aerosols.

---

## Referee Comment (RC3) · Anonymous Referee #3 · 1 Apr 2017

**Comment on "Seasonal variations of high time-resolved chemical compositions, sources and evolution for atmospheric submicron aerosols in the megacity Beijing" by Wei Hu et al.**

This manuscript by Hu et al. presents a comprehensive study on chemical compositions, sources and evolution for atmospheric submicron aerosols in the megacity Beijing in four seasons. Following typical AMS analysis, the source and evolution process of aerosol, especially OA in different seasons are discussed. The contributions of primary and secondary PM are also examined. With the wealth of AMS and ACSM studies in many locations including Beijing city, I was hoping for some unique discussions or scientific insights that were not available in the literature already. In particular, there are a lot of ACSM studies in Beijing in the literature that provide very similar analyses and results of the current paper. The additional analysis of OSc etc with the HR data is very similar to what has been published by many others. While the paper is well written and generally clear, the paper needs to be improved in emphasizing more on new science and insights of the work beyond our current understanding of PM in Beijing.

Some other comments below:

1   Page 7, Line 11-12, please show the satellites data in the supporting information.

2   Page 7, Line 14-15, have the authors examined the contributions of organic nitrate to the high nitrate concentration associated with biomass burning?

3   Page 8, Line 1-3, it would be useful to show the correlations of nitrate with RH under high and low RH conditions in addressing the point that aqueous reactions could contribute to nitrate remarkably in highly humid and static air.

4   Page 9 and Page 10, please clarify the calculation of the particle growth rate.

5   Have the authors tried more factors in PMF or using ME-2 to resolve a BBOA factor in spring and winter? In Page 22 and Figure 10, it seems that in both spring and autumn, a large number of data points affected by biomass burning. Further the author mentioned that in the satellites data, they identified some days with intense biomass burning activities in spring as well. In winter, it seems that HOA and CCOA spectra also bear some BBOA features.

6   Page 16, Line 7-9, the authors state "the peaks of OOA (or LO-OOA) coincided with the peaks of primary emitted COA (spring, summer and autumn) and HOA (winter) in diurnal patterns, probably because strong primary emissions favored the partitioning of oxidized gas precursors to particulate phase" However, on Page 17, Line 7-8, the they also say "In both autumn and winter, the fractions of OOA slightly increased around 100 $\mu g\ m^{-3}$, implying that POA probably transformed to SOA more effectively within this range." Please clarify if it was POA transformed to SOA or primary emissions favored the partitioning of oxidized gas precursors to particulate phase for the increase of OOA.

7   Page 20, Line 12-19, please explain the use of OA/$\Delta$CO ratio rather than $\Delta$OA/$\Delta$CO and $\Delta$POA/$\Delta$CO used in the literature.

8    Figure 3, Please add the standard deviations in the diurnal plots.

9    Figure S5, I suggest making use of the OM:OC ratio in the AMS to convert OA to OC or OC to OA in the comparison with EC/OC analyzer.

---

## Author Comment (AC1) · 25 Jun 2017

Dear Co-Editor Dr. Jason Surratt and Referees, We highly appreciate the detailed valuable comments of the three referees on our manuscript of "'acp-2017-115". The suggestions are quite helpful for us. We have incorporated them in the revised manuscript to improve the quality of our paper. Please see the detailed point-by-point response below and the changes marked blue in the revised manuscript.

Thank you very much!

Best regards, Min Hu On behalf of co-authors

[Figure]

Please also note the supplement to this comment:
http://www.atmos-chem-phys-discuss.net/acp-2017-115/acp-2017-115-AC1-
supplement.pdf

**Supplement:**

Dear Co-Editor Dr. Jason Surratt and Referees,

We highly appreciate the detailed valuable comments of the three referees on our manuscript of "'acp-2017-115". The suggestions are quite helpful for us. We have incorporated them in the revised manuscript to improve the quality of our paper. Please see the detailed point-by-point response below and the changes marked blue in the revised manuscript.

Thank you very much!

Best regards,

Min Hu

On behalf of co-authors

**Point-by-Point Response to Reviewers′ Comments:**

**Referee #1**

**General comments**

This study presents high time resolution chemical characterization of ambient submicron aerosols measured in Beijing between March 2012 and March 2013 using HR-ToF-AMS. Seasonal comparison of the chemical composition shows that organic aerosol (OA) and inorganic species (i.e. sulfate, nitrate, and ammonium) contributes equally towards total $PM_1$ mass. The chemical component composition and diurnal trend were impacted by atmospheric conditions of each season. Relationship between the atmospheric conditions to $PM_1$ composition need to be further discussed and/or clarified as outlined in the specific comments.

Additionally, sources of OA fraction in Beijing were identified using PMF. Primary OA (i.e. HOA and COA) and secondary OA sources were observed in all seasons, whereas BBOA and CCOA were only observed in autumn and winter, respectively. It is interesting that CCOA factor was resolved in winter dataset indicating significant contribution of coal combustion from domestic heating to ambient submicron aerosol. There seems to be an unresolved factor in winter dataset that could be a BBOA factor. Residual mass spectra should be added as supplementary information for evaluating the PMF solution.

Overall, this study falls within the scope of Atmospheric Chemistry and Physics journal. The results contribute to long-term and seasonal evaluation of air quality and can support air pollution prevention policy in Beijing. The manuscript is written pretty well with only some minor issues as outlined in the technical comments. I support publication of this manuscript after minor revisions.

**Response:** Thank you very much for your helpful comments and kind encouragement. According to the specific comments, we strengthened the content on the relationship between the atmospheric conditions and $PM_1$ compositions. The supplementary information for evaluating the PMF solution was added. Please see the detailed response below and the changes marked blue in the revised manuscript.

**Specific comments**

1. $PM_1$ mass concentration is estimated as a sum of NR-$PM_1$ measured by AMS and BC measured by Aethalometer. However, cutoff size of the AMS's inlet ($PM_1$) is different from the

Aethalometer (PM$_{2.5}$). How did you consider the cutoff size when summing the measurements? This information can be added into the experimental section.

**Response:** Ambient BC particles are largely found in the Aitken and accumulation modes (i.e., in the submicron range) because of their formation mechanism (Bond et al., 2013; Huang et al., 2012a; Rose et al., 2006). The sum of non-refractory species measured by the HR-ToF-AMS and BC measured by instruments such as MAAP or aethalometer with the cut-size of 2.5 μm is often treated as total PM$_1$ in previous studies (Huang et al., 2010, 2012b, 2013; He et al., 2011; Hu et al., 2013, 2016). Herein we thought this match has little influence on PM$_1$.

In the revision, "***Atmospheric black carbon (BC) particles are mostly in the submicron range because of their formation mechanisms (Bond et al., 2013)***" was added in Line 11, Page 6.

"***(non-refractory species measured by the AMS and BC by the aethalometer or MAAP)***" was added to clarify "*main chemical components in PM$_1$*" in Line 12, Page 6.

2. Pg 6 Lns 9-11: What are considered in the certain atmospheric conditions? How do these conditions relate to the current study?

**Response:** Wang et al. (2016) showed that the aqueous oxidation of SO$_2$ by NO$_2$ is key to efficient sulfate formation but is only feasible under two atmospheric conditions: on fine aerosols with high relative humidity and NH$_3$ neutralization or under cloud conditions. Similarly, Cheng et al. (2016) found that high reaction rates of SO$_2$ oxidized by NO$_2$ to form sulfate are sustained by the high neutralizing capacity of the atmosphere in northern China. This mechanism is self-amplifying because higher aerosol mass concentration corresponds to higher aerosol water content, resulting in faster sulfate production and more severe haze pollution. In summary, the certain atmospheric conditions represent the fine aerosols with high aerosol water content and NH$_3$ neutralization or under cloud conditions. These two studies included field observations conducted in urban area of Beijing.

In the revision, "*Wang et al. (2016) and Cheng et al. (2016) found that high levels of sulfate and fine PM can be explained by reactive aqueous oxidation of SO$_2$ by NO$_2$ under certain atmospheric conditions*" was changed to "***Recently it was found that high levels of sulfate and fine PM in northern China can be explained by reactive aqueous oxidation of SO$_2$ by NO$_2$ under certain atmospheric conditions, i.e., on the fine aerosols with high aerosol water content and NH$_3$ neutralization or under cloud conditions (Cheng et al., 2016; Wang et al., 2016)***".

3. Pg 8 Lns 8-10: "Affected by different meteorological conditions, e.g., solar radiation, temperature, RH, boundary layer, and mountain-valley breeze in summer (Fig. S7), as well as different emission sources, the chemical compositions in $PM_1$ showed distinct diurnal patterns in four seasons." The beginning of sentence implies that the meteorological conditions specific for summer, whereas the ending suggests the distinctive diurnal patterns in four seasons. I think meteorological conditions in four seasons, not only summer, will affect the chemical composition and diurnal patterns in each season. This needs to be clarified.

**Response:** "*mountain-valley breeze in summer*" means that mountain-valley breeze is the most common in summer in Beijing. In the revision, "*in summer*" was deleted.

4. Pg 8 Ln 19: "Flatter diurnal cycles of sulfate were observed in four seasons, …". What does "flatter" here compare with? Does it compare with OA diurnal pattern? Also, add discussion about the diurnal patterns in spring and summer. Sulfate in spring showed small peaks around 13:00 and 20:00, whereas in summer, it peaked at evening only. What could possibly drive changes in sulfate diurnal patterns?

**Response:** In the revision, "***Compared with OA diurnal patterns,***" was added in Line 15, Page 9. "*In the summer of Beijing, the formation of sulfate is mainly attributed to in-cloud/aqueous-phase reactions (70−80%), and also arises from photochemical oxidation of $SO_2$ (Guo et al., 2010). In this study, sulfate enhanced gradually from morning to late afternoon in summer.*" was changed to "***In spring, sulfate showed small peaks around 13:00 and 20:00. In the daytime, both active photochemical production and more favorable dispersion conditions due to higher planetary boundary layer (PBL) possibly caused such a diurnal pattern of sulfate. In summer, sulfate enhanced gradually from morning to evening and peaked in the evening only, indicating that the photochemical production of sulfate might be significant. In addition to the gas-phase processes, the formation of sulfate is mainly attributed to aqueous-phase reactions in clouds and/or wet aerosols (70−80%) in the summer of Beijing (Guo et al., 2010).***"

5. Pg 9 Lns 9-11: I think nitrate diurnal pattern in autumn is still significant; nitrate increased in the morning and then later in the evening. Also, what is the evidence of combined effect of sulfate and nitrate to ammonium diurnal pattern? Maybe you can correlate the diurnal variations and provide the coefficient of determination.

**Response:** In the revision, "*The diurnal variation of nitrate in spring and autumn was insignificant.*" was changed to "***In autumn, nitrate increased in the morning and then later in***

*the evening, primarily driving by the photochemical production. The diurnal variation of nitrate in spring was insignificant.*"

Because ammonium is usually in the forms of ammonium sulfate and ammonium nitrate, so the diurnal pattern of ammonium should be a combined effect of sulfate and nitrate. The coefficients of determination between the diurnal variations of sulfate and ammonium are calculated as 0.503 ($p<0.05$), 0.191, 0.802 ($p<0.01$) and 0.914 ($p<0.01$) in four seasons, respectively. The coefficients of determination between the diurnal variations of nitrate and ammonium are calculated as 0.803 ($p<0.01$), 0.549 ($p<0.01$), 0.855 ($p<0.01$) and 0.976 ($p<0.01$) ($p<0.05$), respectively.

In the revision, "*In summer, the diurnal variation of ammonium correlated better (r=0.55, p<0.01) with that of nitrate than sulfate (r=0.19, p>0.05). In other seasons, the diurnal variations of ammonium showed good correlations with those of both sulfate and nitrate (r=0.50–0.98, p<0.01 or 0.05).*" was added in Line 11, Page 10.

6. Pg 10 Ln 3: The size distribution of $NH_4$ is pretty similar to $NO_3$ in all four seasons. The size difference is not obvious in Fig. 4. To help comparison, you can consider increasing the size of the figure (panel for each season), or providing a summary table of size distribution in the SI section.

**Response:** Figure 4 is re-illustrated as follows.

[Figure]

Figure 4. Mass size distributions of chemical compositions in NR-PM$_1$ during the spring (a, b), summer (c, d), autumn (e, f) and winter (g, h) observations.

In the revision, "*The size distribution of ammonium was more consistent with that of nitrate in autumn and winter. Nitrate likely mainly existed in the form of NH$_4$NO$_3$, resulted from the condensation of gaseous HNO$_3$ and NH$_3$ on the surfaces of atmospheric particles (Liu et al., 2008).*" was changed to "***The size distribution of ammonium was more consistent with that of nitrate than sulfate in winter. More nitrate likely existed in the form of NH$_4$NO$_3$, formed by the reaction of gaseous HNO$_3$ and NH$_3$ condensed on atmospheric particles (Weimer et al., 2006).***"

7. Pg 11 Lns 23-24: Provide reference(s) that show the role of emission reduction of gaseous precursors to SOA formation. For example: Pye, et al. Epoxide pathways improve model predictions of isoprene markers and reveal key role of acidity in aerosol formation. Environ. Sci. Technol. 2013, 47, 11056-11064.

**Response:** In the revision, "*(Guo et al., 2014; Pye, et al., 2013)*" was added.

8. Pg 14 Lns 18-20: Are you implying that there is a factor (BBOA) that is not resolved by PMF in the winter dataset? What may cause PMF can't resolve BBOA factor in winter? Is the residual still showing important mass spectrum (e.g. m/z 60, 73) and time series features? Diagnostic plots of PMF winter dataset analysis (see Zhang et al. 2011) will be useful for readers to understand the analysis process. These diagnostic plots can be added in the SI section and referred in the main text. In addition to winter, diagnostics for other season should be provided in the SI section to help reader understand the PMF analysis process.

**Response:** There might be relatively limited BBOA contribution to the OA in winter, so the BBOA factor cannot be resolved in our dataset by free PMF. In recently published papers (Sun et al., 2013, 2014; Zhang et al., 2014), there were also no BBOA factor resolved by PMF analysis in urban Beijing during the similar periods of 2012 and 2013.

The mass spectrum of the residual doesn't show obvious feature of characteristic ion fragments.

In the revised manuscript, "***The optimum solutions were selected following the steps as described in Zhang et al. (2011). The key diagnostic plots of the PMF analysis are shown in Sect. S5 in the supplement.***" was added in Line 5, Page 6.

The PMF diagnostic plots and related tables are added in Sect. S5 in the supplement as follows.

Factor number from 1 to 10 and the different seeds (0-50) were selected to run in the PMF model. For the spring observation, diagnostic plots of the PMF analysis are shown in Fig. S16. When OA was separated into four fractions, it included more oxidized (MO-OOA) and less oxidized OOA (LO-OOA), cooking OA (COA) and hydrocarbon-like OA (HOA). The performances of spectra and time series of the four factors at different $f_{peak}$ are shown in Fig. S17. When OA was separated into five fractions, OOA was also split into two factors, but more information on the OA sources (BBOA) could be provided. When more than five factors, OOA decomposed into three or more factors. After comparing the performances of MS spectra and

time series of five factors at different $f_{peak}$, the five factors, $f_{Peak}=1$ solution is chosen as the optimal solution for this PMF analysis because the signal of the characteristic ion fragment m/z is more obvious in one factor. In the five-factor solution, the mass spectra of two OOA factors are similar (r= 0.955), and the elemental ratios and OA/OC ratios (O/C: 0.99, 1.00; H/C: 1.50, 1.26; OA/OC: 2.51, 2.47) are close. It is unclear if the two OOA components represent distinct sources or chemical types. Thus, two OOA factors were combined into total OOA for further analysis (Hayes et al., 2013). Finally, four factors of OA were obtained, i.e., oxygenated OA (OOA), cooking OA (COA), hydrocarbon-like OA (HOA), and biomass burning OA(BBOA), as shown in Fig. S28. The detailed information on how to select the optimum PMF solution is available in Table S4.

[Figure]

Figure S16. Diagnostic plots of the PMF analysis on OA mass spectral matrix for the spring observation.

[Figure]

Figure S17. The spectra and time series of 4-factor solution at different $f_{peak}$ values for the spring observation.

[Figure]

Figure S18. The spectra and time series of 5-factor solution at different $f_{peak}$ values for the spring observation.

**Table S4** Descriptions of PMF solutions for the spring observation in Beijing.

| Factor number | $F_{peak}$ | Seed | $Q/Q_{exp}$ | Solution Description |
|---|---|---|---|---|
| 1 | 0 | 0 | 2.90 | Too few factors, large residuals at time periods and key *m/z*'s |
| 2 | 0 | 0 | 1.79 | Too few factors, large residuals at time periods and key *m/z*'s |
| 3 | 0 | 0 | 1.49 | Too few factors (OOA, HOA and COA). The $Q/Q_{exp}$ at different seeds (0-50) are very unstable. Factors are mixed to some extent based on the time series and spectra. |
| 4 | 0 | 0 | 1.32 | OA factors could be identified as MO-OOA, LO-OOA, COA and HOA. Time series and diurnal variations of OA factors are consistent with the external tracers. But, the signal of characteristic ion *m/z* 60 biomass burning is strong in HOA factor. |
| **5** | **1** | **0** | **1.25** | **Final choice for the PMF solution. Two OOA factors, COA, HOA and BBOA are identified. Two similar OOA factors are combined for further analysis. Time series and diurnal variations of OA factors are consistent with the external tracers.** |
| 6-10 | 0 | 0 | 1.20-1.06 | Factor split. OOA was split into three or more factors with similar spectra, however, different time series. |
| 5 | -3 to 3 | 0 | 1.25-1.39 | In $f_{peak}$ range from −1.0 to 1.0, factor MS of OOA and COA are nearly identical, but there is a shift between HOA and BBOA for some ion fragments. The time series of OOA and HOA are nearly identical, but the other show some changes. |

For the summer observation, the 4-factor, $f_{peak}$=0 solution was selected as the optimum solution. Four OA factors are more oxidized (MO-OOA) and less oxidized OOA (LO-OOA), cooking OA (COA) and hydrocarbon-like OA (HOA). The performances of spectra and time series of the four factors at different $f_{peak}$ were also investigated. The detailed information on how to select the optimum PMF solution can be found in Figure S19-S21 and Table S5.

[Figure]

Figure S19. Diagnostic plots of the PMF analysis on OA mass spectral matrix for the summer observation.

[Figure]

Figure S20. The spectra and time series of 4-factor solution at different $f_{peak}$ values for the summer observation.

[Figure]

Figure S21. Unit mass spectra and time series of OA factors for 5-factor solution. The factors are marked as OOA1, OOA2, COA, OOA3 and HOA, respectively. OOA1, OOA2 and OOA3 show similar MS features (r=0.87–0.90). It is unclear if these OOA components represent distinct sources or chemical types. The elemental ratios and OA/OC ratios of each component are added.

**Table S5** Descriptions of PMF solutions for the summer observation in Beijing.

| Factor number | Fpeak | Seed | Q/Q$_{exp}$ | Solution Description |
|---|---|---|---|---|
| 1 | 0 | 0 | 7.20 | Too few factors, large residuals at time periods and key *m/z*'s |
| 2 | 0 | 0 | 5.31 | Too few factors, large residuals at time periods and key *m/z*'s |
| 3 | 0 | 0 | 4.73 | Too few factors (OOA, HOA and COA). Factors are mixed to some extent based on the time series and spectra. |
| **4** | **0** | **0** | **4.53** | **Optimum solution for the PMF analysis (MO-OOA, LO-OOA, COA and HOA). Time series and diurnal variations of OA factors are consistent with the external tracers. The spectra of four factors are consistent with the source spectra in AMS spectra database.** |
| 5-10 | 0 | 0 | 4.30-3.74 | Factor split. Take 5 factor number solution as an example, OOA is likely split into three factors with similar mass spectra and different time series. However, it is difficult to explain if they represent distinct sources or chemical types. |
| 4 | -3 to 3 | 0 | 4.53-4.58 | In $f_{peak}$ range from −1.0 to 1.0, factor MS and time series are nearly identical. |

The solution of the PMF analysis for the autumn observation is similar to that for the spring observation. When OA was separated into five fractions, OOA was also split into two factors, but a BBOA factor of distinct characteristics ($f_{60}$=1.3%) could be identified. When more than five factors, OOA decomposed into three or more factors. The performances of spectra and time series of the four factors at different $f_{peak}$ are nearly identical. The five factors, $f_{Peak}$=0 and seed=0 solution is chosen as the optimal solution for this PMF analysis. In the five-factor solution, two OOA factors have similar MS characteristics (r= 0.976) and the elemental ratios and OA/OC ratios (O/C: 0.85–0.91; H/C: 1.24–1.40; OA/OC: 2.24–2.37) are close. It is unclear if the two OOA components represent distinct sources or chemical types. Thus, two OOA factors were combined into total OOA for further analysis (Hayes et al., 2013). Finally, four factors of OA were obtained, i.e., oxygenated OA (OOA), cooking OA (COA), hydrocarbon-like OA (HOA), and biomass burning OA(BBOA), as shown in Fig. S30. The detailed information on how to select the optimum PMF solution are given as Figs. S22-S24 and Table S6.

[Figure]

Figure S22. Diagnostic plots of the PMF analysis on OA mass spectral matrix for the autumn observation.

[Figure]

Figure S23. The spectra and time series of 5-factor solution at different $f_{peak}$ values for the autumn observation.

[Figure]

Figure S24. Unit mass spectra and time series of OA factors for 6-factor solution. The factors are marked as OOA1, BBOA, OOA2, COA, HOA1 and HOA2, respectively. The time series of BBOA and OOA2 trend well (r=0.78). HOA1 and HOA2 have similar MS (r=0.94) and diurnal variations (r=0.93). These factors appear mixed with each other.

**Table S6** Descriptions of PMF solutions for the autumn observation in Beijing.

| Factor number | Fpeak | Seed | Q/Q$_{exp}$ | Solution Description |
|---|---|---|---|---|
| 1 | 0 | 0 | 4.55 | Too few factors, large residuals at time periods and key *m/z*'s |
| 2 | 0 | 0 | 3.09 | Too few factors, large residuals at time periods and key *m/z*'s |
| 3 | 0 | 0 | 2.25 | Too few factors (OOA, COA and HOA). The Q/Q$_{exp}$ at different seeds (0-50) are very unstable. The HOA factor contain high abundance (1.0%) of *m/z* 60. |
| 4 | 0 | 0 | 2.07 | Four factors include two similar OOA factors, COA and HOA. The HOA factor contain high abundance (1.1%) of *m/z* 60. |
| **5** | **0** | **0** | **1.97** | **Optimum solution for the PMF analysis (two OOA factor, COA, HOA and BBOA). Two similar OOA factors are combined for further analysis. Time series and diurnal variations of OA factors are consistent with the external tracers.** |
| 6-10 | 0 | 0 | 1.88-1.71 | Factor split. Some of the split factors have time series and MS that appear mixed. |
| 5 | -3 to 3 | 0 | 1.97-2.10 | In $f_{peak}$ range from −1.0 to 1.0, factor MS and time series are nearly identical. |

For the winter observation, a 5-factor, $f_{peak}$=0 solution was selected as the optimum solution. Five OA factors are more oxidized (MO-OOA) and less oxidized OOA (LO-OOA), cooking OA (COA), coal combustion OA (CCOA) and hydrocarbon-like OA (HOA), respectively. The performances of spectra and time series of the five factors at different $f_{peak}$ were also investigated. The detailed information on how to select the optimum PMF solution can be found in Figs. S25-S27 and Table S7.

[Figure]

Figure S25. Diagnostic plots of the PMF analysis on OA mass spectral matrix for the winter observation.

[Figure]

Figure S26. The spectra and time series of 5-factor solution at different $f_{peak}$ values for the winter observation.

[Figure]

Figure S27. Unit mass spectra and time series of OA factors for 6-factor solution. The factors are marked as OOA1, OOA2, COA, OOA3, HOA and CCOA, respectively. OOA1, OOA2 and OOA3 show similar time series or MS features (r=0.56–0.95). The characteristics of OOA3 factor is not obvious. It is unclear if these factors represent distinct sources or chemical types.

**Table S7** Descriptions of PMF solutions for the winter observation in Beijing.

| Factor number | Fpeak | Seed | Q/Q$_{exp}$ | Solution Description |
|---|---|---|---|---|
| 1 | 0 | 0 | 7.09 | Too few factors, large residuals at time periods and key *m/z*'s |
| 2 | 0 | 0 | 3.57 | Too few factors, large residuals at time periods and key *m/z*'s |
| 3 | 0 | 0 | 3.14 | Too few factors (OOA-, HOA- and COA-like). The Q/Q$_{exp}$ at different seeds (0-50) are very unstable. Factors are mixed to some extent based on the time series and spectra. |
| 4 | 0 | 0 | 2.84 | OA is split to two OOA factors, COA and HOA. It seems that HOA mixed with CCOA. |
| **5** | **0** | **0** | **2.70** | **Optimum choice for PMF factors (MO-OOA, LO-OOA, COA, HOA and CCOA). Time series and diurnal variations of OA factors are consistent with the external tracers. The spectra of four factors are consistent with the source spectra in AMS spectra database.** |
| 6-10 | 0 | 0 | 2.59-2.33 | Factor split. Take 6 factor number solution as an example, OOA was split into three factors with similar spectra and/or time series. |
| 5 | -3 to 3 | 0 | 2.70-2.78 | In *f*$_{peak}$ range from −1.0 to 1, factor MS and time series are nearly identical, but there is likely a shift of the time series for LO-OOA and COA during the heavy-pollution episodes. |

 It is interesting that the LO- and MO-OOA fractions slightly increased around 100 μg m$^{-3}$ only. What influence the increase of OOA factors during particular OA mass concentration?

**Response:** It has been found that a substantial fraction (50–75%) of POA is semivolatile, evaporates when the plume becomes more dilute, and is then available in the gas phase to take part in photochemical reactions (Shrivastava et al., 2006; Robinson et al., 2007). This material has the physicochemical properties of SOA. Murphy and Pandis (2009) define that fresh POA is emitted in the particulate phase and has not undergone chemical processing, while oxidized POA (OPOA) refers to POA compounds that evaporate and undergo oxidation in the gas phase, which allows them to reduce their volatility and re-condense back to the particulate phase. SOA produced from the oxidation of intermediate-volatility compounds (IVOCs) was also included in OPOA mainly because the IVOC emissions were calculated based on the POA emissions (Fountoukis et al., 2014). Intermediate-volatility organic compounds (IVOCs) have been proposed to be an important source of SOA (Zhao et al., 2014).

SOA chemistry is complex and the contribution of different pathways is not well understood. More work is needed to accurately identify the volatility and aging of primary emissions, and to quantify the contributions of SOA from different sources and formation mechanisms.

In this study, we have no strong evidence to elaborate the mechanisms or pathways. We only give our hypothesis in this sentences.

In the revision, "*In both autumn and winter, the fractions of OOA slightly increased around 100 μg m$^{-3}$, implying that POA probably transformed to SOA more effectively within this range*" was changed to "***In both autumn and winter, the fractions of OOA slightly increased around 100 μg m$^{-3}$. More work is needed to accurately clarify the cause of the OOA increase within this range.***"

 The VK diagram in winter does not show any correlation (r$^2$ = 0.02) between H/C and O/C. The slope is nearly zero (0.08; not broad). This suggests that hydroxylation and/or peroxidation processes were likely to occur in Beijing during winter. On the other hand, carboxylation process might be unlikely to occur since the slope is higher than -0.5. Please elaborate or clarify the discussion.

**Response:** Currently, there is no substantial progress on explicating specific oxidation pathways for OA evolution (Chen et al., 2015; Heald et al., 2010). In winter, CCOA was resolved by the PMF analysis, which was different from other seasons. This is a possible reason

the VK Diagram in winter was different from those in other seasons.

In the revision, "*However, in winter, the correlation was not good (r² = 0.02), with a fitted slope -0.08, exhibiting a "broader" range in the VK Diagram than in other seasons (Fig. 8 and Fig. S21)*" was changed to "**However, the VK diagram in winter (Fig. 8 and Fig. S39) does not show any correlation (r² = 0.02) between H/C and O/C. The slope is nearly zero (-0.08). This is possibly caused by the more complex sources of OA in winter, e.g., CCOA was only resolved in winter**" in Line 25, Page 19.

"*In winter, the scatterplot of H/C vs. O/C ratios in the VK Diagram showed "broader" slopes, hinting the more complex sources and evolution processes of OA. The scatterplot indicated that OA in winter mainly evolved between the hydroxylation or peroxidation reactions (slope = 0) and carboxyl groups addition with fragmentation (slope = -0.5).*" was changed to "**In winter, the nearly zero slope of the VK Diagram suggests that hydroxylation and/or peroxidation processes (slope = 0) were likely to occur in Beijing during winter. On the other hand, carboxylation process might be unlikely to occur since the slope is higher than -0.5.**" in Line 5, Page 20.

**Technical comments**

1. Pg 1 Ln 14: If $PM_1$ here is NR-PM1+BC, it needs to be specified in the sentence.

**Response:** "*An Aerodyne high resolution time-of-flight aerosol mass spectrometry (HR-ToF-AMS) and other relevant instrumentations for gaseous and particulate pollutants were deployed. The average mass concentrations of submicron particulate matter ($PM_1$)…*" was changed into "**An Aerodyne high resolution time-of-flight aerosol mass spectrometry (HR-ToF-AMS) was deployed to measure non-refractory chemical components of submicron particulate matter (NR-$PM_1$). The average mass concentrations of $PM_1$ (NR-$PM_1$+black carbon)**…".

2. Pg 2 Lns 26-27: Is SNA important only for the urban area or in general? Clarify the sentence and add reference.

**Response:** "**in urban regions (Guo et al., 2014; Huang et al., 2014; Lee, 2015)**" was added.

3. Pg 5 Lns 7: ..., the last well calibrated…

**Response:** "*last well calibrated*" was corrected into "**the last well calibrated**".

4. Pg 5 Lns 8-9: Do you mean "the size distribution calibration"? Please clarify.

**Response:** "*the size distribution*" was corrected into "***the size distribution calibration***".

5. Pg 7 Lns 1-2: Do you mean "…PM₁ in winter was higher than previous studies in Beijing…"? Also please clarify what does "close periods" mean.

**Response:** "*PM₁ in winter was higher than before, and equivalent to those in close periods (Zhang et al., 2014; Sun et al., 2014)*" was revised into "*PM₁ in winter was higher than **the previous results in Beijing before 2013**, and equivalent to those in close periods **from January to February 2013 (Fig. 1;** Zhang et al., 2014; Sun et al., 2014)."*

6. Pg 7 Ln 13: Figures S6 and S8 don't display fire points as mentioned in the sentence.

**Response:** The maps of fire points from satellites were added in the supplementary material as follows.

[Figure]

Fig. S10. Fire points observed by satellites (https://firms.modaps.eosdis.nasa.gov/firemap) in Beijing and surrounding areas during 7−8 (a) and 26−28 (b) Apr. 2012.

[Figure]

Fig. S11. Fire points observed by satellites (https://firms.modaps.eosdis.nasa.gov/firemap) in Beijing and surrounding areas during the autumn observation.

7. Pg 9 Ln 27: …the internally mixed states…

**Response:** "*the internal mixed states*" was changed into "***the internally mixed states***".

8. Pg 11 Ln 8: The term "freshly secondary formation" is ambiguous. Consider "new SOA formation".

**Response:** This sentence was removed because it is arguable (Comment 8 of Reviewer#2).

9. Pg 12 Ln 26: Provide the coefficient of determination for any statement about correlation.

**Response:** Here the uncentered coefficient (UC) represents the cosine of the angle between a

pair of mass spectra (MS) or time series (TS) as vectors, such that

$$UC = \cos\theta = (x \cdot y)/(|x| \cdot |y|)$$

where x and y denote a pair of MS or TS as vectors. The uncentered correlation is very similar to the well-known Pearson coefficient (R) for mass spectra, and quite correlated with Pearson's R for time series (Ulbrich et al., 2009). Whereas, it is a value of cosine and has no the coefficient of determination.

The coefficients of determination for Pearson correlations were checked in the whole text.

10. Pg 15 Lns 24-27: Insert respectively after on average. Breaking this sentence into two will be better as it is pretty long at the current state.

**Response:** "*respectively*" was added. The original sentence was broken into two sentences, that is "***The concentrations of COA factors at the noon peak (about 12:00–14:00) were 3.1, 2.5, 4.0 and 4.4 μg m^{-3} on average, respectively. COA reached the highest concentrations in the evening (18:00–21:00), as 5.6, 6.2, 14.0 and 7.5 μg m^{-3} on average, respectively.***"

11. Pg 18 Ln 2: …, 1.52-1.63, …

**Response:** Corrected.

12. Pg 18 Ln 18: What does "shallower" mean? Do you mean flatter slope?

**Response:** "*shallower*" was changed into "***flatter***".

13. Pg 20 Ln 7: …(i.e. lower H/C ratio, and higher O/C ratio and ***OSC***)….

**Response:** Corrected.

14. Pg 20 Ln 28: What does "higher" compare to? Is it higher than "stable level". What does stable level mean here?

**Response:** The content was addressed in the text above, "*In laboratory and field studies on OA aging under strong oxidizing conditions, it was found that the OA/ΔCO ratios remain relatively*

*stable at high O/C ratios with the increase of O/C ratios because organics obtain oxygen atoms but loss carbon atoms in the oxidation processes (DeCarlo et al., 2008, 2010)."* "*Higher*" is compared to the results we obtained. The *"stable level"* means *"OA/ΔCO ratios remain relatively stable at high O/C ratios with the increase of O/C ratios*".

In the revision, "*get higher*" was changed to "***reach higher values***".

15. Pg 21 Ln 28: Provide indicator (value) of the low aging degrees of air masses.

**Response:** "**(-log (NOₓ/NOᵧ) < 0.05)**" was added.

16. Fig 3: Need label (Hours of Day) for the X-axis.

**Response:** "Hour of day" was added as X-axis label.

17. Figs 5-8: Colors for MO-OOA and OOA are difficult to distinguish on print.

**Response:** The difference between the colors for MO-OOA and OOA was largened in these figures. Please see these figures in the manuscript.

18. Figs 6-7: The legends for these figures are not consistent. OOA is missing from Fig. 7.

**Response:** The legend of OOA was added in Fig. 7.

19. Fig. S13: … LV-OOA, MO-OOA…

**Response:** The figure caption was corrected.

**References**

Pye, H. O. T. Epoxide pathways improve model predictions of isoprene markers and reveal key role of acidity in aerosol formation. Environ. Sci. Technol., 47, 11056-11064, 2013.

Zhang, Q. Q. Understanding atmospheric organic aerosols via factor analysis of aerosol mass spectrometry: a review, Analyt. Bioanalyt. Chem., 401, 3045-3067, 2011.

Thank you very much for your comments and suggestions. Your any further comments and suggestions are appreciated.

**Referee #2**

The manuscript investigates the seasonal variations of compositions, sources, and evolution for atmospheric aerosols in the megacity of Beijing. While field measurements have been carried out by other groups in this city, this work provides a more comprehensive exploration of organic aerosol composition and evolution. Generally, the paper is quite well written. The methodology and results are presented clearly. The measurements provide sufficiently new data that the manuscript may merit publication. However, the qualitative interpretation of the results may limit the application and impacts of this work. This calls for a more detailed data analysis, explaining the mechanisms behind the observations. I suggest the authors respond to the following suggestions before the paper is accepted to publication.

**Response:** Thank you very much for your helpful comments and kind encouragement. We responded the specific suggestions of the reviewer, and revised the manuscript correspondingly to improve the quality of this manuscript. Please see the detailed response below and the changes marked blue in the revised manuscript.

**Major Comment:**

My main question is that as mentioned by the authors, the characteristics and evolution of aerosol pollution are multifactorial (e.g. meteorological conditions, regional transport, and local sources). These factors can have distinct patterns in different seasons, thus the formation, transformation and removal of pollutants are affected. The authors have provided nice discussions on the sources and evolution of ambient atmospheric aerosols under the influence of different meteorological conditions and sources. However, what would be the impacts of regional or long range transport on the composition, formation and evolution of atmospheric aerosols in different seasons in Beijing? For example, the authors mentioned that regional transport can play a role in the sulfate concentration in the winter season. While regional transport impacts have briefly discussed or mentioned in various sections or paragraphs, it would be nice to have a more coherent, quantitative discussion on the role of regional (or long range) transport in the concentration and evolution of different inorganic and organic components in different seasons.

**Response:** Ideally, regional transport may be evaluated using chemical transport models that interactively consider emissions, chemistry, meteorology, and removal (Zhang et al., 2015b). There have been several studies on the contribution of regional/long-distance transport to atmospheric aerosols in Beijing. Hu et al. (2015) applied a source-oriented Community Multiscale Air Quality (CMAQ) simulation to determine source sector/region contributions to primary aerosols. They found that local residential/transportation emissions and residential/industrial emissions from Heibei contributed predominately to primary aerosols in the spring of Beijing. In summer and fall, local industrial emissions were the largest contributor. In winter, local/regional residential/industrial sources contributed to over 90% of primary aerosols in Beijing. Wu et al. (2011) investigated the impact of local and regional sources on the air pollutants in Beijing with an online air pollutant tagged module in the Nested Air Quality Prediction Model System (NAQPMS), and estimated the air pollutant contributions from local and regional sources to the surface layer (about 30 m) and the upper layer (about 1.1 km) in the summer of 2006 in Beijing. The contribution of local sources to $PM_{10}$ at the surface layer in Beijing was dominated (75%). Comparatively, the contribution of the surrounding regions (e.g., southern Beijing) was large (more than 50%) to $PM_{10}$ at the 1.1 km layer. Lin et al. (2015) extended the Comprehensive Air Quality Model with Extensions (CAMX) v 5.4, and found that in August 2007, the local source contributions to anthropogenic and biogenic SOA in Beijing were 23.8% and 16.6%, respectively; regional/long-distance transport dominated for both anthropogenic and biogenic SOA in Beijing. Zhang et al. (2015a) quantified the source contributions to surface $PM_{2.5}$ pollution over North China from January 2013 to 2015 using the GEOS-Chem transport model and its improved adjoint, and attributed about half of the $PM_{2.5}$ pollution in Beijing to sources outside of the city.

Other than chemical transport models, Jia et al. (2008) describe a novel technique for quantifying regional aerosol solely from a series of fast-response aerosol measurements at a single site. Based on the strong asymmetric "sawtooth cycles" of aerosol in Beijing, they suggest the regional component averages about 50% and can range from 10%–20% during northwesterly flow to 70% or so during southerly flow. The uncertainties of the concentrations of regional aerosol can be up to 50% for one day but <10% when totaled over a sawtooth. Using data from two sites, Guo et al. (2010) give a rough estimation that 87% of the $PM_{1.8}$ in the summer of urban Beijing were regional contributions. The regional contributions to sulfate, ammonium and oxalate in $PM_{1.8}$ were 90%, 87% and 95%, respectively. Nitrate formation was locally dominant.

Li et al. (2015) report the results of the hybrid receptor modeling with observations at 10 sites in Beijing and cluster analysis of the 48-h air mass back trajectories for the same study periods. They show that the source areas leading to high $PM_{2.5}$ in Beijing were primarily located in the southwest and south of Beijing. However, Zhang et al. (2015b) considered the back trajectory analysis is unsuitable for urban-scale studies because it employs large-scale wind fields with coarse resolutions and does not consider the complex urban canopies.

The contribution of regional (long-distance) transport to aerosol pollution is important for reducing regional air pollution under a joint control policy. The contributions of local/regional transport are difficult to quantify, however, even with transport models (Jia et al., 2008). In this study, this issue will not be focused on.

Here for reference, the results of backward trajectories of air parcels calculated with the NOAA's HYSPLIT4 trajectory model (http://www.arl.noaa.gov/hysplit.html) are shown in Fig. S43 in the supplementary, to give an insight into the impacts of regional/long-distance on the composition, formation and evolution of atmospheric aerosols in different seasons in Beijing.

The following texts were added in the manuscript and the supplementary, respectively.

In the manuscript, "***To give an insight into the impacts of regional/long-distance transport on atmospheric aerosols in Beijing, the backward trajectories of air parcels during the observation periods were calculated with the NOAA's HYSPLIT4 trajectory model (http://www.arl.noaa.gov/hysplit.html). A new 3-day backward trajectory was traced from the observation site at an altitude of 500 m above ground level every hour. Cluster analyses of backward trajectories were applied to reveal the major pathways during different campaigns (Sect. S11 in the supplement).***" was added in Sect. 2.1.

In the supplementary, the following contents "***During the seasonal observations in Beijing, the pathways of dominant air masses are different. Both long-distance transported and regional/local air masses influenced Beijing. In summer, the transport distance of long-distance transported air masses was shorter than in other seasons. In general, with the decrease of transport distance, the concentration of PM$_1$ gradually increased. When Beijing was dominated by regional/local air masses, the fractions of secondary inorganic species (SNA) increased, while the contributions of carbonaceous components (OA+BC) decreased, which is consistent with the previous results in Beijing (Sun et al., 2010; Huang et al., 2010; Zhang et al., 2014). Higher concentrations of SNA and PM$_1$ under the control of regional/local air masses reflected the great contribution of secondary formation from the gaseous precursors (e.g., NO$_x$ and SO$_2$) emitted by vehicles and coal combustion in urban areas.***"

"***During the observations in spring, summer and autumn, the contributions of OOA (MO-OOA+LO-OOA) increased when Beijing was dominated by regional or local air masses. In summer, the fractions of LO-OOA in OA were high (29–48%) regardless of the different trajectories, signifying that the secondary formation from photochemical oxidations probably***

*made an important contribution to OA. During the winter observation, POA and OOA contributed equally to OA in most cases due to the long-lasting stable weather conditions, indicating that both primary pollutants and regional secondary formation made important contributions to OA. When Beijing was dominated by long-distance transported air masses from north polar regions in winter, OOA contributed more significantly to OA, implying that organic aerosols were fully aged during long-distance transport.*" was added.

[Figure]

Figure S43. Back trajectories for each of the identified clusters and corresponding average main components of PM$_1$ and OA in PM$_1$ during the seasonal campaigns. (a) spring; (b)summer; (c) autumn and (d) winter. The filling color of main chemical species in PM$_1$ is the same with other figures.

**Minor comments:**

1. Abstract, "The evolution process of OA in different seasons was investigated with multiple metrics and tools. The average carbon oxidation states and other metrics show that the oxidation state of OA was the highest in summer, probably due to both strong photochemical and aqueous-phase oxidations." Any evidence supports the contribution of aqueous-phase oxidation to OA? What is the relative contribution of photochemical vs. aqueous-phase oxidation to the concentration and evolution of OA?

**Response:** Frankly, the contributions of different formation pathways have not been quantified yet, and it is difficult to give a quantitative result for the relative contribution of photochemical vs. aqueous-phase oxidation to the secondary formation based on field observation data only. Further studies including laboratory experiments, field observations and model simulations are needed to close the gaps in the current understanding of SOA formation pathways (Ervens et al., 2011).

Based on laboratory experiments and simulations, Ervens et al. (2011) suggest that SOA formed in cloud and aerosol water (aqSOA) might contribute almost as much mass as SOA formed in the gas phase to the SOA budget, with highest contributions from biogenic emissions of VOCs in the presence of anthropogenic pollutants (i.e., $NO_x$) at high RH and cloudiness. Xu et al. (2017) show that aqueous-phase processes have a dominant impact on the formation of MO-OOA, and the contribution of MO-OOA to OA increases substantially as a function of RH or liquid water content (LWC) in aerosols. In contrast, photochemical processing plays a major role in the formation of LO-OOA, as indicated by the strong correlations between LO-OOA and odd oxygen ($O_x=O_3+NO_2$) during periods of photochemical production.

Here the LWC in aerosols in this study was roughly estimated with the ISORROPIAII model. The input data were the four species (sulfate, nitrate, ammonium and chloride) measured by the AMS, RH and temperature. The reverse mode and the metastable state of aerosols were selected. According to the results, the ambient aerosols were generally in aqueous phase. The average values of aerosol LWC in four seasons were $17.3\pm28.5$, $18.8\pm24.9$, $12.8\pm27.3$ and $25.2\pm32.8$ $\mu g\ m^{-3}$, respectively. During the heavy-polluted episodes, the LWC was frequently higher than $100\ \mu g\ m^{-3}$.

Table S14 shows the correlation coefficients between OOA and some indicators (RH, LWC, $O_3$ and $O_x$). The good correlations between OOA and RH and/or LWC indicate that aqueous-phase reactions play a more significant role in OOA formation. The slope of OOA against $O_x$ steepened with the increase of RH and LWC (Figs. S41 and S42), also implying that the

aqueous-phase oxidation was an important pathway of the OOA formation. The strong correlations between total oxidant ($O_x$) and LO-OOA in summer, and between $O_x$ and OOA in autumn and winter, suggesting photochemical processes contributed substantially to OOA, especially LO-OOA, in these seasons.

**Table S14.** Pearson correlation coefficients between secondary organic and inorganic species and some indicators (RH, LWC, $O_3$ and $O_x$). Coefficients greater than 0.5 are in bold. Correlation is significant at the 0.01 level (2-tailed) except for those marked by [#].

|  |  | RH | LWC | $O_3$ | $O_x$ |
|---|---|---|---|---|---|
| Spring | OOA | **.661** | **.754** | -.199 | .345 |
|  | $SO_4^{2-}$ | **.764** | **.901** | -.207 | .186 |
|  | $NO_3^-$ | **.705** | **.827** | -.318 | .254 |
| Summer | MO-OOA | .176 | **.751** | .131 | .264 |
|  | LO-OOA | .005[#] | .469 | .360 | **.527** |
|  | $SO_4^{2-}$ | .114 | **.686** | .262 | .359 |
|  | $NO_3^-$ | .335 | **.873** | -.123 | .000[#] |
| Autumn | OOA | .483 | **.803** | -.433 | **.571** |
|  | $SO_4^{2-}$ | **.552** | **.919** | -.340 | .338 |
|  | $NO_3^-$ | .489 | **.854** | -.379 | **.548** |
| Winter | MO-OOA | **.624** | **.647** | -.504 | **.640** |
|  | LO-OOA | **.692** | **.840** | -.534 | **.726** |
|  | $SO_4^{2-}$ | **.801** | **.899** | -.613 | **.597** |
|  | $NO_3^-$ | **.785** | **.819** | -.637 | **.655** |

[Figure]

Figure S41. Scattering plots of OOA mass concentrations against $O_x$ concentrations. (a)

spring; (b) summer; (c) autumn and (d) winter. Data points are color coded by RH.

[Figure]

Figure S42. The same as above. Data points are color coded by estimated LWC in aerosols.

In the revision, the related content above is added in Sect. S5 in the Supplement.

In the abstract, "*It was indicated by the good correlations (Pearson correlation coefficients r=0.53−0.75, p<0.01) between LO-OOA and odd oxygen ($O_x=O_3+NO_2$), and between MO-OOA and liquid water content in aerosols.*" was added.

In the manuscript, "*OOA had strong correlations with RH and/or LWC (Table S14), indicating that aqueous-phase reactions play a dominant role in OOA formation. The slope of OOA against $O_x$ steepened with the increase of RH and LWC (Figs. S41 and S42), also implying that the aqueous-phase oxidation was an important pathway of the OOA formation. The good correlations (r=0.53−0.73, p<0.01) between $O_x$ and LO-OOA in summer, and between $O_x$ and OOA in autumn and winter, suggesting photochemical processes also contributed substantially to SOA, especially LO-OOA, in these seasons.*" was added in Line 17, Page 16.

2. Page 6, "In spring, summer and winter, SNA accounted for about 60% in $PM_1$ due to the secondary aerosol formation through strong photochemical and aqueous-phase reactions." Like OA, what is the relative contribution of photochemical vs. aqueous-phase oxidation to the

concentration and evolution of SNA?

**Response:** As mentioned in the response to comment 1, the contributions of photochemical and aqueous-phase oxidation to the concentration and evolution of SNA have not been quantified yet.

It is well known that most of aerosol sulfate are formed from heterogeneous or aqueous-phase/cloud processes (Kulmala et al., 2016). On a global scale, about 80% of the sulfate formation occurs within clouds. Ambient aerosol populations often show two distinct submicron modes (<0.2 μm and 0.5–1 μm) where the larger (droplet) mode is formed from the smaller (condensation) mode through volume-phase reactions in clouds and wet aerosols (Ervens et al., 2011). Based on this assumption, Guo et al. (2010) found that the gas-to-particle condensation process was important for aerosol pollution in the summer of Beijing. In urban Beijing, the formation of sulfate was mainly attributed to in-cloud or aerosol droplet process (80%) and gas condensation process (14%).

As shown in Table S14, sulfate and nitrate correlated well with RH and/or LWC in four seasons, indicating that the aqueous-phase reactions in aerosols played an important role in secondary inorganic formation in Beijing. The contributions of photochemical processes to sulfate and nitrate in four seasons were probably less than those of aqueous-phase reactions according to the weaker correlations between secondary inorganics and $O_x$. Especially, in summer nitrate showed no correlation with $O_x$.

As shown in Fig. S40, when RH was higher than 40% (or 30% in winter), aqueous-phase processed likely played a dominant role in secondary inorganic formation.

[Figure]

Figure S40. Influences of RH and $O_3$ concentrations on sulfate and nitrate formation.

Conversion ratios for sulfur and nitrogen ($F_S$ and $F_N$) were calculated as follows:

$$F_S = n - SO_4^{2-}/(n - SO_4^{2-} + n - SO_2) \qquad (3)$$
$$F_N = n - NO_3^-/(n - NO_3^- + n - NO_2) \qquad (4)$$

where $n$ means the amount of substance of the gaseous and particulate pollutants, mol m$^{-3}$.

Table S15 Pearson correlation coefficients between $F_S$ and $F_N$ with RH, LWC, $O_3$, $O_x$ and $NH_4^+$.

|  | Spring | | Summer | | Autumn | | Winter | |
|---|---|---|---|---|---|---|---|---|
|  | $F_S$ | $F_N$ | $F_S$ | $F_N$ | $F_S$ | $F_N$ | $F_S$ | $F_N$ |
| RH | .339 | **.722** | **.639** | .393 | .432 | **.574** | **.531** | **.744** |
| LWC | .475 | **.816** | .464 | **.816** | **.647** | **.874** | **.583** | **.676** |
| $O_3$ | .024* | -.146 | -.100 | .024* | .035 | -.321 | -.268 | -.518 |
| $O_x$ | -.166 | .277 | -.096 | .052 | -.122 | .359 | .342 | .368 |
| $NH_4^+$ | .324 | **.924** | .353 | **.822** | .495 | **.938** | .598 | **.855** |

Note: Coefficients greater than 0.5 are in bold. Correlation is significant at the 0.01 level (2-tailed) except for those marked with *.

The good correlation between $F_N/F_S$ and RH/LWC also support that aqueous-phase reactions in aqueous aerosols and/or clouds could contribute to secondary inorganic formation remarkably in highly humid air.

In the revision, the related content above is added in Sect. S10 in the Supplement.

In the manuscript, "*In spring, summer and winter, SNA accounted for about 60% in PM₁ due to the secondary aerosol formation through strong photochemical and aqueous-phase reactions.*" was changed to "***In four seasons, SNA accounted for about 40−60% in PM₁. Secondary inorganics (sulfate and nitrate) correlated well with RH and/or LWC (Table S14), indicating that aqueous-phase reactions in aerosols played an important role in secondary inorganic formation in Beijing. The contribution of photochemical processes to secondary inorganics was likely less than those of aqueous-phase reactions according to the weaker correlations between secondary inorganics and odd oxygen (Oₓ=O₃+NO₂). Especially, nitrate did not correlate with Oₓ in summer (Table S14).***"

3. Page 6, "Wang et al. (2016) and Cheng et al. (2016) found that high levels of sulfate and fine PM can be explained by reactive aqueous oxidation of $SO_2$ by NO₂ under certain atmospheric conditions." Please elaborate what the meaning of "certain atmospheric conditions". How frequent these atmospheric conditions occur during the field campaign?

**Response:** Wang et al. (2016) showed that the aqueous oxidation of $SO_2$ by $NO_2$ is key to

efficient sulfate formation but is only feasible under two atmospheric conditions: on fine aerosols with high relative humidity and $NH_3$ neutralization or under cloud conditions. Similarly, Cheng et al. (2016) found that high reaction rates of $SO_2$ oxidized by $NO_2$ to form sulfate are sustained by the high neutralizing capacity of the atmosphere in northern China. This mechanism is self-amplifying because higher aerosol mass concentration corresponds to higher aerosol water content, resulting in faster sulfate production and more severe haze pollution. In summary, the certain atmospheric conditions represent the fine aerosols with high aerosol water content and $NH_3$ neutralization or under cloud conditions. These two studies included field observations conducted in urban area of Beijing. The PM pollution episodes exhibit a periodic cycle of several days (Guo et al., 2014; Fig. S6–S9), i.e., these atmospheric conditions occur quite often during the field campaigns.

In the revision, "*Wang et al. (2016) and Cheng et al. (2016) found that high levels of sulfate and fine PM can be explained by reactive aqueous oxidation of $SO_2$ by $NO_2$ under certain atmospheric conditions*" was changed to "***Recently it was found that high levels of sulfate and fine PM in northern China can be explained by reactive aqueous oxidation of $SO_2$ by $NO_2$ under certain atmospheric conditions, i.e., on the fine aerosols with high aerosol water content and $NH_3$ neutralization or under cloud conditions (Cheng et al., 2016; Wang et al., 2016)***".

4. Page 6, "Compared with the previous results in Beijing (Fig. 1), $PM_1$ in summer was lower than before, which likely resulted from the more effective rainout (Fig. S7) and lower concentrations of gaseous precursors (Table 1)." What are the frequency of precipitation/rainout and the concentration of gaseous precursors reported in previous studies?

**Response:** The time series of meteorological conditions and submicron aerosol species during 9 to 21 July 2006 (Sun et al., 2010) and 26 June to 28 August 2011 (Sun et al., 2012) are copied as follows. It is likely that the concentration of submicron aerosols was reduced litter by washout in some rain events (e.g., 13 and 18 Jul. 2006). So, we hypothesized that "*$PM_1$ in summer was lower than before, which **likely** resulted from the more effective **washout** (Fig. S7)*".

[Figure]

Figure R1. Time series of meteorological conditions and submicron aerosol species during 9 to 21 July 2006 (Sun et al., 2010).

[Figure]

Figure R2. Time series of meteorological conditions and submicron aerosol species during 26 June to 28 August 2011 (Sun et al., 2012).

Compared with the previous results of gaseous precursors ($SO_2$ and $NO_x$) during July-August 2011 and June 2012 (Sun et al., 2015), the concentrations in this study were lower (Table R1).

In the revision, "*rainout*" was corrected to "***washout***"; "*(Table 1)*" was changed to "***(Table 1 and Table 1 in Sun et al. (2015))***".

Table R1. Comparison of the concentrations of gaseous pollutants during July-August 2011 and June 2012 (Sun et al., 2015), Aug.–Sept. 2011(Hu et al., 2016) and Jul. 29–Aug. 29 2012 (this study).

|  | Jul.-Aug. 2011, Jun. 2012 | Aug.–Sept. 2011 | Jul. –Aug. 2012 |
|---|---|---|---|
| $SO_2$ (ppb) | 5.4±0.8 | 4.4±2.6 | 3.1±3.3 |
| CO (ppm) | 1.8±1.3 | 0.9±0.5 | 0.7±0.3 |
| NO (ppb) | 7.8±10.8 | 8.0±14.5 | 3.5±6.6 |
| $NO_y$ (ppb) | 35.6±17.9 | 35.6±21.6 ($NO_2$) | 30.1±17.5 |
| $O_3$ (ppb) | 33.3±29.1 | 63.2±30.6 | 41.6±34.7 |

5. Page 7, "While, $PM_1$ ranged much more broadly, with the highest concentrations of over 200 or 300 µg m$^{-3}$, resulting from accumulated pollutants under extremely unfavorable meteorological conditions or strong primary emissions." What is "the extremely unfavorable meteorological conditions"?

**Response:** "*extremely unfavorable meteorological conditions or strong primary emissions*" was changed to "***strong primary emissions coupled with extremely unfavorable meteorological conditions, e.g., long-lasting stagnant weather, high humidity, and temperature inversion.***"

6. Page 7, "The proportions of nitrate increased more significantly, and the nitrate concentration increased rapidly under higher RH (Fig. 2; Pearson correlation coefficients r=0.71, 0.34, 0.49 and 0.79, p<0.01). These results indicate that the aqueous reactions could contribute to nitrate remarkably in highly humid and static air." What are the aqueous reactions referring to? Aerosol phase reactions, in-cloud reactions or both? Ambient RH is a good proxy for indicting the occurrence and importance of aqueous phase reactions. However, what the physical state of the ambient aerosols (e.g. aqueous or solid)? What is the aerosol water content inferred or predicted from the aerosol speciation data and meteorological conditions? Will the water content and physical state of the aerosols play a role in the aqueous phase reactions and nitrate formation? If the aqueous phase reactions involve in-cloud reactions, any data (e.g. cloud coverage) can be used to support the importance of in-cloud reactions to the formation of nitrate and organic compounds during the campaign?

**Response:** Aqueous-phase uptake (in clouds or aqueous aerosols) of nitric acid and condensation onto preexisting particles are two possible pathways to form fine mode nitrates (Guo et al., 2010). In this study, whether the aqueous-phase reactions were aerosol phase

reactions, in-cloud reactions or both was not considered.

As mentioned in the response to Comment 1 and 2, the liquid water content (LWC) in aerosols was roughly estimated with the ISORROPIAII model. According to the results, the ambient aerosols were generally in aqueous phase. The Pearson correlation coefficients between nitrate and LWC in four seasons were 0.827, 0.873, 0.854 and 0.819 ($p<0.01$), respectively, indicating that the aqueous-phase reactions in aerosols played an important role in nitrate formation in Beijing. The good correlation between $F_N$ and RH/LWC also support that aqueous-phase reactions in aqueous aerosols and/or clouds could contribute to nitrate remarkably in highly humid air.

In the revision, the average LWC was added in Table 1.

"*The proportions of nitrate increased more significantly, and the nitrate concentration increased rapidly under higher RH (Fig. 2; Pearson correlation coefficients r=0.71, 0.34, 0.49 and 0.79, p<0.01). These results indicate that the aqueous reactions could contribute to nitrate remarkably in highly humid and static air.*" was changed to "***The proportions of nitrate increased significantly, and the nitrate concentration increased rapidly under higher RH (Fig. 2 and Fig. S6–S9). Nitrate showed good correlations with RH (Pearson correlation coefficients r=0.34–0.79, p<0.01) and the LWC in aerosols (r=0.82–0.87, p<0.01). These results indicate that the aqueous-phase reactions in wet aerosols and/or clouds could substantially contribute to nitrate formation.***"

7. Page 8, "The peak concentration of OA in the evening in autumn was about two times higher than in spring and summer, consistent with the results in Oct.-Nov. 2011 (Sun et al., 2015), possibly because of the more intense cooking activities." Please elaborate why there is more instance cooking activity in the evening in autumn, but not in other seasons?

**Response:** In contrast to other seasons (including spring, summer and winter), the weather conditions in autumn are more moderate and pleasant (Table 1). As mentioned in the manuscript, "*In autumn charcoal-grilling or barbecuing has become one of the most popular outdoor recreational activities in urban Beijing and surrounding areas.*"

In the revision, "***In autumn, charcoal-grilling or barbecuing has become one of the most popular outdoor recreational activities in urban Beijing and surrounding areas, due to more moderate and pleasant weather conditions than in other seasons (Table 1).***" was moved from Page 17 to Page 9.

Page 18, "*in autumn charcoal-grilling or barbecuing has become one of the most popular outdoor recreational activities in urban Beijing and surrounding areas, which could be an important source of COA.*" was changed to "***in autumn intensive barbecue activities in urban Beijing and surrounding areas could be an important source of COA as mentioned above.***"

8. Page 11, "LO-OOA dominated OA in summer (44%) due to the freshly secondary formation from strong photochemical oxidations; whereas, MO-OOA was dominant in OA in winter (33%), maybe because the air masses were more aged on heavy-polluted days." Please elaborate why the air masses are more aged on the heavy-polluted days in winter, but not in other seasons?

**Response:** After deeper discussion among coauthors, we think this comparison is arguable because the oxidation degrees of LO-OOA and MO-OOA during summer and winter were different. So, in the revision, we **removed** these contents.

9. Page 13, "Fewer cooking activities during and around the Chinese New Year holiday (7-19 Feb.; Fig. S17), as well as the lower evaporation rate of oil, led to the lower concentration and proportion of COA in winter." This argument is interesting. How does the evaporate rate of oil depend on the temperature? What the contribution of this evaporation process to the total volatile organic compounds generated/originated from the cooking activities?

**Response:** In China, both vegetable oils and animal fats are commonly used as cooking oils. In China, village residents use homemade vegetable oils and animal fats in cooking, and store them in the kitchens for several months even longer than one year. The edible oils partially evaporate very slowly over months to years leaving a sticky varnish on the inner wall of the container. The odors of stored cooking oils can be more obviously smelt in summer than in winter. Thus, the evaporation process of cooking oils is much slower and significantly less visible in winter.

Vegetable oils and animal fats used in cooking contain saturated and unsaturated fatty acid esters of glycerol. The Clausius–Clapeyron relation for a system consisting of vapor and liquid of a pure substance is in applicable to cooking oils. Several previous studies (e.g., Ndiaye et al., 2005; Murata et al., 1993) measured the vapor pressures of some kinds of edible oils at different temperatures. For vegetable oils, vapor pressure values varied in the range of 0.19–2.16 kPa, whereas for fatty acid ethyl ester (FAEE) mixtures a maximum value of 4.85 kPa was found (Ndiaye et al., 2005).

The vapor pressures of edible oils at different temperatures measured by Ndiaye et al. (2005) are listed as follows:

Table R2. Experimental vapor pressure data for soybean oil, castor oil, and their FAEE derivatives.

| soybean oil | | | castor oil | | | FAEE from soybean oil | | | FAEE from castor oil | | |
|---|---|---|---|---|---|---|---|---|---|---|---|
| | $p$/kPa | | | $p$/kPa | | | $p$/kPa | | | $p$/kPa | |
| $T$/K | measured | $\sigma$ | $T$/K | measured | $\sigma$ | $T$/K | measured | $\sigma$ | $T$/K | measured | $\sigma$ |
| 294.1 | 0.354 | 0.019 | 299.1 | 0.192 | 0.023 | 290.4 | 0.437 | 0.035 | 290.7 | 0.259 | 0.006 |
| 299.5 | 0.453 | 0.015 | 304.2 | 0.258 | 0.009 | 294.6 | 0.549 | 0.018 | 293.7 | 0.323 | 0.017 |
| 304.0 | 0.543 | 0.043 | 308.9 | 0.328 | 0.003 | 299.5 | 0.718 | 0.014 | 298.0 | 0.427 | 0.053 |
| 309.3 | 0.661 | 0.075 | 313.6 | 0.416 | 0.021 | 304.7 | 0.916 | 0.033 | 303.8 | 0.577 | 0.084 |
| 313.8 | 0.753 | 0.083 | 319.1 | 0.529 | 0.032 | 309.0 | 1.144 | 0.077 | 308.9 | 0.753 | 0.126 |
| 318.8 | 0.892 | 0.114 | 323.5 | 0.647 | 0.049 | 314.5 | 1.447 | 0.086 | 313.7 | 0.951 | 0.167 |
| 323.5 | 1.033 | 0.140 | 329.2 | 0.806 | 0.048 | 319.5 | 1.781 | 0.086 | 318.5 | 1.198 | 0.214 |
| 328.7 | 1.195 | 0.151 | 334.0 | 0.979 | 0.053 | 324.3 | 2.196 | 0.104 | 323.4 | 1.505 | 0.267 |
| 333.9 | 1.364 | 0.146 | 339.2 | 1.182 | 0.031 | 328.9 | 2.652 | 0.091 | 328.0 | 1.857 | 0.315 |
| 338.7 | 1.513 | 0.111 | 344.2 | 1.402 | 0.020 | 334.4 | 3.228 | 0.025 | 333.2 | 2.301 | 0.337 |
| 344.0 | 1.717 | 0.073 | 348.2 | 1.648 | 0.031 | 338.9 | 3.958 | 0.014 | 338.6 | 2.806 | 0.273 |
| 348.9 | 1.962 | 0.063 | 354.0 | 1.950 | 0.185 | 344.3 | 4.658 | 0.165 | 343.5 | 3.415 | 0.224 |
| 353.2 | 2.155 | 0.006 | | | | | | | 348.5 | 4.000 | 0.044 |
| | | | | | | | | | 354.6 | 4.848 | 0.540 |

Because edible oils have significantly lower evaporation rates than isoviscous hydrocarbon and synthetic base fluids at room temperatures, the contribution of the evaporation process to the total VOCs generated/originated from the cooking activities might be insignificant. Further studies are essential to confirm it.

In the revision, "*as well as the lower evaporation rate of oil*" was changed to "***and the lower evaporation rates of cooking oils (Ndiaye et al., 2005)***".

10. Page 16" In summer, OOA showed obvious diurnal variations: MO-OOA peaked in the morning and afternoon; LO-OOA showed two pronounced peaks at noon and at night, which was likely influenced by the photochemical oxidations and aqueous-phase formation from POA." It is not clear why the formation of LO-OOA is likely influenced by the photochemical oxidations and aqueous-phase formation from POA. Any other processes that can be contributed to the formation and transformation of LO-OOA?

**Response:** LO-OOA is usually considered to be fresh SOA. Traditionally, SOA is formed from gas-phase oxidation, heterogeneous reactions on aerosol surfaces, and multiphase chemistry of gas-phase organic compounds (Kroll et al., 2005; Hallquist et al., 2009). It has been recognized that a substantial fraction (50–75%) of POA is semivolatile. This fraction of POA evaporates when the plume becomes more dilute, and is then available in the gas phase to take part in photochemical reactions, producing SOA far exceeding that from traditional precursors (Robinson et al., 2007; Shrivastava et al., 2006). Murphy and Pandis (2009) define that oxidized

POA (OPOA) refers to POA compounds that evaporate and undergo oxidation in the gas phase, which allows them to reduce their volatility and re-condense back to the particulate phase. SOA produced from the oxidation of intermediate-volatility compounds (IVOCs) was also included in OPOA mainly because the IVOC emissions were calculated based on the POA emissions (Fountoukis et al., 2014).

In addition to SOA formation in the gas phase (gasSOA), SOA (aqSOA) forms in the aqueous phase of cloud/fog and aerosol water through complex chemical reactions. Based on model studies, Ervens et al. (2011) compared aqSOA and gasSOA yields and mass predictions for selected conditions, and suggest that aqSOA might contribute almost as much mass as gasSOA to the SOA budget.

As mentioned above in the response to Comment 1, in summer, LO-OOA correlated well with $O_x$ (r=0.53, $p$<0.01) and LWC (r=0.47, $p$<0.01).

In the revision, "*In summer, OOA showed obvious diurnal variations: MO-OOA peaked in the morning and afternoon; LO-OOA showed two pronounced peaks at noon and at night, which was likely influenced by the photochemical oxidations and aqueous-phase formation from POA.*" was changed to "***In summer, OOA showed obvious diurnal variations: MO-OOA peaked in the morning and afternoon; LO-OOA showed two pronounced peaks at noon and at night. This was likely the co-effect of SOA formation via gas-phase photochemical reactions in the daytime and aqueous chemistry in humid air at night (Ervens et al., 2011; Lim et al., 2010).***"

11. Page 16, "the concentration and proportion of LO-OOA increased significantly in the afternoon (12:00-16:00), up to 7 ug m$^{-3}$ and 50%, respectively (Fig. 6b), suggesting that LO-OOA was a strong local/regional photochemical product despite the much higher PBL in the daytime (Hu et al., 2016a)" What is the contribution of regional transport to LO-OOA (and other components)?

**Response:** The contribution of regional (long-distance) transport to aerosol pollution is important for reducing regional air pollution under a joint control policy. The contributions of local/regional transport are difficult to quantify, however, even with transport models (Jia et al., 2008). In this study, this issue will not be focused on.

Please refer to the response to general comments for details.

In the revision, "***Regardless of the air mass history (local, regional and long-distance transported), LO-OOA dominated OOA, accounting for 29–48% of OA (Fig. S43).***" was

added in Line 20, Page 17.

12. Page 17, "In both autumn and winter, the fractions of OOA slightly increased around 100 ug m$^{-3}$, implying that POA probably transformed to SOA more effectively within this range." Please elaborate why the POA is more likely transformed to SOA under these conditions. What mechanisms or pathways?

**Response:** As mentioned in the response to Comment 10, it has been found that a substantial fraction (50–75%) of POA is semivolatile, evaporates when the plume becomes more dilute, and is then available in the gas phase to take part in photochemical reactions (Shrivastava et al., 2006; Robinson et al., 2007). This material has the physicochemical properties of SOA. Murphy and Pandis (2009) define that fresh POA is emitted in the particulate phase and has not undergone chemical processing, while oxidized POA (OPOA) refers to POA compounds that evaporate and undergo oxidation in the gas phase, which allows them to reduce their volatility and re-condense back to the particulate phase. SOA produced from the oxidation of intermediate-volatility compounds (IVOCs) was also included in OPOA mainly because the IVOC emissions were calculated based on the POA emissions (Fountoukis et al., 2014). Intermediate-volatility organic compounds (IVOCs) have been proposed to be an important source of SOA (Zhao et al., 2014).

SOA chemistry is complex and the contribution of different pathways is not well understood. More work is needed to accurately identify the volatility and aging of primary emissions, and to quantify the contributions of SOA from different sources and formation mechanisms.

In this study, we have no strong evidence to elaborate the mechanisms or pathways. We only give our hypothesis in this sentences.

In the revision, "*In both autumn and winter, the fractions of OOA slightly increased around 100 μg m$^{-3}$, implying that POA probably transformed to SOA more effectively within this range*" was changed to "***In both autumn and winter, the fractions of OOA slightly increased around 100 μg m$^{-3}$. More work is needed to accurately clarify the cause of the OOA increase within this range.***"

13. Page 18, "In spring, summer and autumn, the slopes fell between -1 (the addition of carboxyl functional groups without fragmentation or carbonyl and hydroxyl in different carbons) and -0.5 (carboxyl functionalization with fragmentation)." In addition to overall oxidation pathways,

what other information we could learn from the reported slopes between -1 and -0.5? Are these pathways consistent with the reactions we expected for the formation and chemical transformations of ambient aerosols in Beijing?

**Response:** We did not do chamber experiments to verify the consistency between measurements and simulation. Currently there is no substantial progress on the issue of specific oxidation pathways.

A large data set including surface, aircraft, and laboratory studies of the atomic O/C and H/C ratios of OA is synthesized by Heald et al. (2010) and Chen et al. (2015). The overall fit for ambient data characterizes measurements that span a wide range of OA oxidation, and the complex chemical evolution of OA in the atmosphere could be simply represented in models. Chen et al. (2015) show that laboratory OA including both source and aged types explains some of the key differences in OA observed across different environments. However, the comparisons also reveal significant gaps between ambient and laboratory measured OA composition.

More works are required to bridge the gaps. For instance, it is needed to characterize the yields and bulk composition of SOA produced by aqueous-phase reactions under atmospherically relevant conditions (Chen et al., 2015). In addition, a complete description of OA processing must link the compositional changes with key physical properties (e.g., volatility, hygroscopicity, light absorption) of the aerosol (Heald et al., 2010).

14. Page 19, "In winter, the scatterplot of H/C vs. O/C ratios in the VK Diagram showed "broader" slopes, hinting the more complex sources and evolution processes of OA. The scatterplot indicated that OA in winter mainly evolved between the hydroxylation or peroxidation reactions (slope = 0) and carboxyl groups addition with fragmentation (slope = -0.5)." Any hypothesis or explanation for the complex sources and evolution process of OA in the winter. Why these processes have not been observed or suggested in the other seasons?

**Response:** As mentioned above in the response to Comment 13, there is no substantial progress on explicating specific oxidation pathways for OA evolution. In winter, CCOA was resolved by the PMF analysis, which was different from other seasons. This is a possible reason these processes were not suggested in the other seasons.

[revised manuscript text omitted]

Thank you very much for your comments and suggestions. Your any further comments and suggestions are appreciated.

**Referee #3**

This manuscript by Hu et al. presents a comprehensive study on chemical compositions, sources and evolution for atmospheric submicron aerosols in the megacity Beijing in four seasons. Following typical AMS analysis, the source and evolution process of aerosol, especially OA in different seasons are discussed. The contributions of primary and secondary PM are also examined. With the wealth of AMS and ACSM studies in many locations including Beijing city, I was hoping for some unique discussions or scientific insights that were not available in the literature already. In particular, there are a lot of ACSM studies in Beijing in the literature that provide very similar analyses and results of the current paper. The additional analysis of OSc etc with the HR data is very similar to what has been published by many others. While the paper is well written and generally clear, the paper needs to be improved in emphasizing more on new science and insights of the work beyond our current understanding of PM in Beijing.

**Response:** Thank the reviewer very much for the helpful comments. In previous studies, the studies conducted in spring and autumn are relatively lacking, this study might be a supplementary of previous studies. As the general comment of Reviewer#2 said, "While field measurements have been carried out by other groups in this city, this work provides a more comprehensive exploration of organic aerosol composition and evolution." "The measurements provide sufficiently new data…" After response to all the comments of three reviewers and revised the manuscript correspondingly, we hope the quality of this manuscript has been improved.

Some other comments below:

1. Page 7, Line 11-12, please show the satellites data in the supporting information.

**Response:** The maps of fire points from satellites were added in the supplementary material. Please refer to the response to Technical Comment 6 of Reviewer#1.

2. Page 7, Line 14-15, have the authors examined the contributions of organic nitrate to the high nitrate concentration associated with biomass burning?

**Response:** In this study, as the title shown, we mainly focused on "*Seasonal variations of high time-resolved chemical compositions, sources and evolution for atmospheric submicron aerosols in the megacity Beijing*". So, we did not mention the details of the contributions of organonitrate (ON) to the high nitrate concentration.

We tried to examine the contributions of organic nitrate during the autumn observation. The concentration of ON was estimated using the PMF analysis including the mass spectra of OA, $NO^+$ and $NO_2^+$ (Sun et al., 2012; Xu et al., 2015; Zhang et al., 2016). As the factor of nitrate inorganic aerosol (NIA) was resolved, it is considered that the $NO^+$ and $NO_2^+$ in other OA factors were from ON. The concentration of ON was estimated using the following equations:

$$NO_{org}^+ = \sum([OA\ factor]_i \times f\ NO^+_i) \quad (1)$$

$$NO_{2,org}^+ = \sum([OA\ factor]_i \times f\ NO_2^+_i) \quad (2)$$

The estimated results are shown in Fig. R3. The contributions of ON to OOA, BBOA and COA factors were 0.4, 0.3 and 0.1 μg m$^{-3}$. On average, ON accounted for about 10% of nitrate measured by the AMS. There are some limitations of the estimation method, such as the collinearity of ion fragments and the PMF analysis lacking the knowledge of standard spectra of different sources, which may result in uncertainties. So, we omitted the related contents in the manuscript.

[Figure]

Figure R3. Estimated organonitrate concentrations in resolved OA factors and nitrate directly measured by the AMS.

3. Page 8, Line 1-3, it would be useful to show the correlations of nitrate with RH under high and low RH conditions in addressing the point that aqueous reactions could contribute to nitrate remarkably in highly humid and static air.

**Response:** Because under higher RH conditions (RH>60%), the ranges of the nitrate concentration are wide, the correlations between nitrate and RH were not good. Instead, we gave the Pearson correlation coefficients between the time series of nitrate and RH, as well as

Fig. 2 in the manuscript.

Conversion ratios for sulfur and nitrogen ($F_S$ and $F_N$) were calculated as follows:

$$F_S = n-SO_4{}^{2-}/(n-SO_4{}^{2-} + n-SO_2) \tag{3}$$

$$F_N = n-NO_3{}^{-}/(n-NO_3{}^{-} + n-NO_2) \tag{4}$$

where $n$ means the amount of substance of the gaseous and particulate pollutants, mol m$^{-3}$。

Table S15 Pearson correlation coefficients between $F_S$ and $F_N$ with RH, LWC, $O_3$, $O_x$ and $NH_4{}^{+}$.

| | Spring | | Summer | | Autumn | | Winter | |
|---|---|---|---|---|---|---|---|---|
| | $F_S$ | $F_N$ | $F_S$ | $F_N$ | $F_S$ | $F_N$ | $F_S$ | $F_N$ |
| **RH** | .339 | **.722** | **.639** | .393 | .432 | **.574** | **.531** | **.744** |
| **LWC** | .475 | **.816** | .464 | **.816** | **.647** | **.874** | **.583** | **.676** |
| **O$_3$** | .024* | -.146 | -.100 | .024* | .035 | -.321 | -.268 | -.518 |
| **O$_x$** | -.166 | .277 | -.096 | .052 | -.122 | .359 | .342 | .368 |
| **NH$_4{}^{+}$** | .324 | **.924** | .353 | **.822** | .495 | **.938** | **.598** | **.855** |

Note: Coefficients greater than 0.5 are in bold. Correlation is significant at the 0.01 level (2-tailed) except those marked with *.

The average values of LWC in aerosols in four seasons were 17.3±28.5, 18.8±24.9, 12.8±27.3 and 25.2±32.8 μg m$^{-3}$, respectively. The Pearson correlation coefficients between nitrate and LWC in four seasons were 0.827, 0.873, 0.854 and 0.819 ($p<0.01$), respectively, indicating that the aqueous-phase reactions in aerosols played an important role in nitrate formation in Beijing.

As listed in Table S15, the good correlations between $F_N$ and RH/ LWC also support that aqueous-phase reactions in aqueous aerosols and/or clouds could contribute to nitrate remarkably in highly humid air.

In the revision, "***Nitrate showed good correlations with RH (Pearson correlation coefficients r=0.34–0.79, p<0.01) and the LWC in aerosols (r=0.82–0.87, p<0.01). These results indicate that the aqueous-phase reactions in wet aerosols and/or clouds could substantially contribute to nitrate formation.***" was added.

For more details, please refer to the response to Comment 6 of Referee#2.

4. Page 9 and Page 10, please clarify the calculation of the particle growth rate.

**Response:** We didn't e calculate the growth rates, and it is an inference. To avoid confusing, in the revision, this sentences was **removed**.

5. Have the authors tried more factors in PMF or using ME-2 to resolve a BBOA factor in spring and winter? In Page 22 and Figure 10, it seems that in both spring and autumn, a large number of data points affected by biomass burning. Further the author mentioned that in the satellites data, they identified some days with intense biomass burning activities in spring as well. In winter, it seems that HOA and CCOA spectra also bear some BBOA features.

**Response:** We tried more factors in PMF and didn't use ME-2 to resolve a BBOA factor in spring and winter. There were some episodes during the spring and winter observations affected by biomass burning. We are applying ME-2 to new datasets to look for minor sources in urban Beijing. In this study, we would like to keep using PMF analysis for OA factorization.

In the revision, after re-evaluation the PMF result, the five-factor solution is selected for the spring observation. Two similar OOA factors are combined. Finally, four components of OA were obtained, i.e., OOA, COA, HOA, and BBOA. However, the contributions of BBOA might be relatively limited during the winter campaign in Beijing, and the BBOA factor cannot be resolved in our dataset by free PMF. The resolved factors in winter were consistent with those in other studies (Fig. 5a). The processes of evaluating the PMF solution are available in the responses to Comment 8 of Reviewer#1.

The key diagnostic plots of the PMF analysis are shown in Sect. S5 in the supplement.

6. Page 16, Line 7-9, the authors state "the peaks of OOA (or LO-OOA) coincided with the peaks of primary emitted COA (spring, summer and autumn) and HOA (winter) in diurnal patterns, probably because strong primary emissions favored the partitioning of oxidized gas precursors to particulate phase" However, on Page 17, Line 7-8, the they also say "In both autumn and winter, the fractions of OOA slightly increased around 100 μg m$^{-3}$, implying that POA probably transformed to SOA more effectively within this range." Please clarify if it was POA transformed to SOA or primary emissions favored the partitioning of oxidized gas precursors to particulate phase for the increase of OOA.

**Response:** Traditionally, SOA is formed from gas-phase oxidation, heterogeneous reactions on aerosol surfaces, and multiphase chemistry of gas-phase organic compounds (Hallquist et al., 2009). It has also been recognized that a substantial fraction (50–75%) of POA is semivolatile, evaporates when the plume becomes more dilute, and is then available in the gas phase to take part in photochemical reactions (Shrivastava et al., 2006; Robinson et al., 2007). This material has the physicochemical properties of SOA. Murphy and Pandis (2009) define that fresh POA is emitted in the particulate phase and has not undergone chemical processing, while oxidized

POA (OPOA) refers to POA compounds that evaporate and undergo oxidation in the gas phase, which allows them to reduce their volatility and re-condense back to the particulate phase. SOA produced from the oxidation of intermediate-volatility compounds (IVOCs) was also included in OPOA mainly because the IVOC emissions were calculated based on the POA emissions (Fountoukis et al., 2014). Intermediate-volatility organic compounds (IVOCs) have been proposed to be an important source of SOA (Zhao et al., 2014).

Grieshop et al. (2009a, b) demonstrate that photooxidation of diluted wood combustion emissions in a smog chamber rapidly produced substantial SOA with a chemical character quite distinct from the primary emissions, and attribute the production of the unexplained OA to the oxidation of low-volatility organic vapors. The existence of these vapors is demonstrated by wood smoke POA evaporating upon isothermal dilution. However, some studies have only observed oxidation without production of new OA mass in fire plumes (Capes et al., 2008; Hoffer et al., 2006). Presto et al. (2014) conducted a series of smog chamber experiments to investigate the transformation of POA and formation of SOA during the photooxidation of dilute exhaust from mobile combustion sources. PMF deconvolved POA and/or SOA factors when substantial POA is present in the dilute exhaust.

SOA chemistry is complex and the contribution of different pathways is not well understood. More work is needed to accurately identify the volatility and aging of primary emissions, and to quantify the contributions of SOA from different sources and formation mechanisms.

In this study, we have no strong evidence to clarify whether it was POA transformed to SOA or primary emissions favored the partitioning of oxidized gas precursors to particulate phase for the increase of OOA. We only give our hypotheses in these two sentences.

In the revision, "*the peaks of OOA (or LO-OOA) coincided with the peaks of primary emitted COA (spring, summer and autumn) and HOA (winter) in diurnal patterns, probably because strong primary emissions favored the partitioning of oxidized gas precursors to particulate phase.*" was changed to "**the peaks of OOA (or LO-OOA) coincided with the peaks of primary emitted COA (spring, summer and autumn) and HOA (autumn and winter) in diurnal patterns, probably because strong primary emissions favored SOA production. It has been found that primary emissions evaporate substantially upon dilution to ambient conditions and those vapors undergo photooxidation, which produces SOA efficiently (Robinson et al., 2007; Murphy and Pandis, 2009).**"

"*In both autumn and winter, the fractions of OOA slightly increased around 100 μg m$^{-3}$, implying that POA probably transformed to SOA more effectively within this range*" was

changed to "***In both autumn and winter, the fractions of OOA slightly increased around 100 μg m^{-3}. More work is needed to accurately clarify the cause of the OOA increase within this range.***"

7. Page 20, Line 12-19, please explain the use of OA/ΔCO ratio rather than ΔOA/ΔCO and ΔPOA/ΔCO used in the literature.

**Response:** Enhancement ratios of OA versus CO are denoted as ΔOA/ΔCO, where Δ indicates the excess amount over the background concentration. In practice, enhancement ratios are often calculated from correlation slopes between OA and CO (de Gouw et al., 2008; de Gouw, J., and Jimenez, 2009). Dzepina et al. (2009) summarized OA/ΔCO ratio from secondary formation during the several campaigns. DeCarlo et al. (2010) described the conceptual framework for the application of the OA/ΔCO method. Hu et al. (2013) studied the evolution of enhancement ratios OA/ΔCO) with the photochemical age at a regional receptor site.

In the revision, "***DeCarlo et al., 2010***" was added in Line 25, Page 21.

8. Figure 3, Please add the standard deviations in the diurnal plots.

**Response:** To avoid clutter, we added one panel of the diurnal plots for each season to show the ranges of diurnal variations (Fig. S12-S15) in the supplementary as follows. In the caption of Figure 3, "***The ranges of diurnal variations for each season are shown in Fig S12–S15.***" was added.

[Figure]

Figure S12. Diurnal patterns of chemical species of submicron particles and gaseous pollutants during the spring observation. The shaded area is between the 25% and 75% quantiles.

[Figure]

Figure S13. Diurnal patterns of chemical species of submicron particles and gaseous pollutants during the summer observation. The shaded area is between the 25% and 75% quantiles.

[Figure]

Figure S14. Diurnal patterns of chemical species of submicron particles and gaseous pollutants during the autumn observation. The shaded area is between the 25% and 75% quantiles.

[Figure]

Figure S15. Diurnal patterns of chemical species of submicron particles and gaseous pollutants during the autumn observation. The shaded area is between the 25% and 75% quantiles.

9. Figure S5, I suggest making use of the OM:OC ratio in the AMS to convert OA to OC or OC to OA in the comparison with EC/OC analyzer.

**Response:** The OA/OC ratios measured by the AMS were used to convert OC measured by EC/OC analyzer to OA. The re-illustrated figure is as follows.

[Figure]

Figure S5. Time series and scatter plots of OA detected by the AMS vs. OA converted from OC measured by a semi-continuous OC/EC analyzer (PM$_{2.5}$ cutoff) using the OA/OC ratios measured by the AMS, and BC vs. EC during the spring observation.

To clarify the possible reasons causing the difference, "*OA concentrations measured by the AMS showed tight correlation with the OC concentrations measured by a Sunset OC/EC Analyzer ($r^2 =0.61$). The linear regression slope of 2.4 is comparable to or within the range (2.0–2.7) of the previous results (Sun et al., 2011; Hu et al., 2013; Lan et al., 2011; Weimer et al., 2006), but is higher than the average OA/OC ratio of 1.81 determined via elemental analysis of the AMS. When the OM were at high concentration levels, the OA converted from OC measured by the OC/EC analyzer, using the OA/OC ratios measured by the AMS, deviated more from the OA time-series trends, consistent with the result of Lan et al. (2011). Possible reasons for this discrepancy include: (1) evaporative losses of semi-volatile organic species due to striking the balance of gas-particle partition after passing the activated-carbon denuder (Grover et al., 2008), and during the carbon analysis (Sun et al., 2011); (2) "over-calibration" of the OC data using the blank filter values (Bae et al., 2006).*" was added.

Thank you very much for your comments and suggestions. Your any further comments and suggestions are appreciated.


**S4 Variations of meteorological parameters and main compositions in PM₁**

[Figure]

Figure S6. Time series of meteorological parameters (wind speed and direction, temperature, relative humidity (RH), and atmospheric pressure), and concentrations and fractions of main chemical compositions in submicron aerosols during the spring observation.

[Figure]

Figure S7. Time series of meteorological parameters (wind speed and direction, temperature, RH, and atmospheric pressure), and concentrations and fractions of main chemical compositions in submicron aerosols during the summer observation.

[Figure]

Figure S8. Time series of meteorological parameters (wind speed and direction, temperature, RH, and atmospheric pressure), and concentrations and fractions of main chemical compositions in submicron aerosols during the autumn observation.

[Figure]

Figure S9. Time series of meteorological parameters (wind speed and direction, temperature, RH, and atmospheric pressure), and concentrations and fractions of main chemical compositions in submicron aerosols during the winter observation.

[Figure]

Figure S10. Fire points observed by satellites (https://firms.modaps.eosdis.nasa.gov/firemap) in Beijing and surrounding areas during 7–8 (a) and 26–28 (b) Apr. 2012.

[Figure]

Figure S11. Fire points observed by satellites (https://firms.modaps.eosdis.nasa.gov/firemap) in Beijing and surrounding areas during the autumn observation.

[Figure]

Figure S12. Diurnal patterns of chemical species of submicron particles and gaseous pollutants during the spring observation. The shaded area is between the 25% and 75% quantiles.

[Figure]

Figure S13. Diurnal patterns of chemical species of submicron particles and gaseous pollutants during the summer observation. The shaded area is between the 25% and 75% quantiles.

[Figure]

Figure S14. Diurnal patterns of chemical species of submicron particles and gaseous pollutants during the autumn observation. The shaded area is between the 25% and 75% quantiles.

[Figure]

Figure S15. Diurnal patterns of chemical species of submicron particles and gaseous pollutants during the winter observation. The shaded area is between the 25% and 75% quantiles.

During four seasons, NOR ($n$−NO$_3^-$/ ($n$−NO$_3^-$+ $n$−NO$_2$)) showed strong correlations with NH$_4^+$, indicating the main form of NO$_3^-$ was NH$_4$NO$_3$. NH$_4^+$ also presented in the form of (NH$_4$)$_2$SO$_4$ and NH$_4$Cl. The predicted NH$_4^+$ was calculated assuming full neutralization of particulate anions of NO$_3^-$, SO$_4^{2-}$, and Cl$^-$ in four seasons. The slopes of linear fitting of measured against predicted NH$_4^+$ in spring, summer, autumn and winter were 1.05, 1.02, 0.85, and 0.89, respectively. In autumn and winter, the relatively lower slopes implied that NH$_4^+$ was not enough to balance NO$_3^-$, SO$_4^{2-}$ and Cl$^-$.

Field and emission studies have shown that a large fraction of KCl can exist in the fresh biomass burning plumes. As biomass burning plumes get aged, more S- and N- containing species (e.g., $KNO_3$ and $K_2SO_4$) in aerosol phase have been found (Li et al., 2003; Yokelson et al., 2009). It has also been reported that NaCl and $NH_4Cl$ are important components in the aerosols directly emitted from biomass burning (Lewis et al., 2009; Levin et al., 2010). It was found that atmospheric aerosols were mostly acidic during heavy pollution episodes (Zhang et al, 2007; Sun et al, 2014.). During the heavy pollution periods in winter, $SO_4^{2-}$ may exist in the form of $NH_4HSO_4$. Chloride existing in the form of KCl and NaCl in aerosol phase for coal combustion sources has been reported (McNallan et al., 1981; Doshi et al., 2009). Therefore, $NO_3^-$, $SO_4^{2-}$, and $Cl^-$ may exist as other forms in addition to ammonium due to the influences of intense autumn biomass burning and coal combustion in autumn and winter, respectively.

Table S3. Concentrations of main chemical components in PM$_1$ during seasonal observations in Beijing in recent years. Unit: μg m$^{-3}$.

| Seasons | Periods | PM$_1$ | OA | SO$_4^{2-}$ | NO$_3^-$ | NH$_4^+$ | Cl$^-$ | BC | References |
|---|---|---|---|---|---|---|---|---|---|
| **Spring** | 10 Apr.-4 May. 2008 | 87.0 | 39.0 | | | | | | Zhang et al., 2013 |
| | **30 Mar.-7 May. 2012** | **45.1** | **14.0** | **9.3** | **10.2** | **7.3** | **1.2** | **3.1** | **This study** |
| **Summer** | 9-21 Jul. 2006 | 80.0 | 28.1 | 20.3 | 17.3 | 13.1 | 1.1 | | Sun et al., 2010 |
| | 24 Jul.-20 Sept. 2008 | 63.1 | 23.9 | 16.8 | 10.0 | 10.0 | 0.5 | 1.8 | Huang et al., 2010 |
| | 5 Jun.-3 Jul. 2008 | 94.0 | 34.0 | 24.8 | 20.1 | 13.7 | 1.4 | | Zhang et al., 2013 |
| | 26 Jun.-28 Aug. 2011 | 50.0 | 20.0 | 9.0 | 12.4 | 8.0 | 0.5 | | Sun et al., 2012 |
| | 4 Aug.-14 Sept. 2011 | 84.3 | 26.4 | 22.0 | 16.8 | 13.7 | 1.0 | 4.4 | Hu et al., 2016 |
| | **29 Jul.-29 Aug. 2012** | **37.5** | **12.5** | **9.7** | **6.4** | **5.4** | **0.4** | **3.2** | **This study** |
| **Autumn** | 4-18 Oct. 2008 | 51.0 | 24.0 | 8.1 | 12.0 | 6.2 | 0.7 | | Zhang et al., 2013 |
| | 1-30 Sept. 2012 | 40.9 | 17.1 | 6.4 | 8.1 | 5.1 | 0.5 | 3.7 | Jiang et al., 2013 |
| | **13 Oct.-13 Nov. 2012** | **41.3** | **18.2** | **5.5** | **7.9** | **4.5** | **2.0** | **3.2** | **This study** |
| **Winter** | 4 Jan.-3 Feb. 2008 | 73.0 | 43.0 | 11.4 | 9.2 | 6.4 | 3.5 | | Zhang et al., 2013 |
| | 22 Nov.-22 Dec. 2010 | 69.5 | 34.5 | 8.7 | 6.8 | 7.7 | 5.8 | 6.0 | Hu et al., 2016 |
| | 14 Dec. 2010-15 Jan. 2011 | | 20.9 | | | | | | Liu et al., 2012 |
| | 21 Nov. 2011-20 Jan. 2012 | 66.8 | 34.4 | 9.4 | 10.7 | 8.7 | 3.3 | | Sun et al., 2013 |
| | 1-17 Jan. 2013 | 83.0 | 38.3 | 14.3 | 12.5 | 9.2 | 2.6 | 6.0 | Sun et al., 2014 |
| | 1 Jan.-1 Feb. 2013 | 89.3 | 44.7 | 19.6 | 12.5 | 8.9 | 3.6 | | Zhang et al., 2014 |
| | **23 Jan.- 2 Mar. 2013** | **81.7** | **29.7** | **17.4** | **16.2** | **11.7** | **2.8** | **3.9** | **This study** |

**S5 Determination of the PMF solution**

Factor number from 1 to 10 and the different seeds (0-50) were selected to run in the PMF model. For the spring observation, diagnostic plots of the PMF analysis are shown in Fig. S16. When OA was separated into four fractions, it included more oxidized (MO-OOA) and less oxidized OOA (LO-OOA), cooking OA (COA) and hydrocarbon-like OA (HOA). The performances of spectra and time series of the four factors at different $f_{peak}$ are shown in Fig. S17. When OA was separated into five fractions, OOA was also split into two factors, but more information on the OA sources (BBOA) could be provided. When more than five factors, OOA decomposed into three or more factors. After comparing the performances of MS spectra and time series of five factors at different $f_{peak}$, the five factors, $f_{Peak}=1$ solution is chosen as the optimal solution for this PMF analysis because the signal of the characteristic ion fragment m/z is more obvious in one factor. In the five-factor solution, the mass spectra of two OOA factors are similar (r= 0.955), and the elemental ratios and OA/OC ratios (O/C: 0.99, 1.00; H/C: 1.50, 1.26; OA/OC: 2.51, 2.47) are close. It is unclear if the two OOA components represent distinct sources or chemical types. Thus, two OOA factors were combined into total OOA for further analysis (Hayes et al., 2013). Finally, four factors of OA were obtained, i.e., oxygenated OA (OOA), cooking OA (COA), hydrocarbon-like OA (HOA), and biomass burning OA(BBOA), as shown in Fig. S28. The detailed information on how to select the optimum PMF solution is available in Table S4.

[Figure]

Figure S16. Diagnostic plots of the PMF analysis on OA mass spectral matrix for the spring observation.

[Figure]

Figure S17. The spectra and time series of 4-factor solution at different $f_{peak}$ values for the spring observation.

[Figure]

Figure S18. The spectra and time series of 5-factor solution at different $f_{peak}$ values for the spring observation.

**Table S4** Descriptions of PMF solutions for the spring observation in Beijing.

| Factor number | $F_{peak}$ | Seed | Q/Q$_{exp}$ | Solution Description |
|---|---|---|---|---|
| 1 | 0 | 0 | 2.90 | Too few factors, large residuals at time periods and key *m/z*'s |
| 2 | 0 | 0 | 1.79 | Too few factors, large residuals at time periods and key *m/z*'s |
| 3 | 0 | 0 | 1.49 | Too few factors (OOA, HOA and COA). The Q/Q$_{exp}$ at different seeds (0-50) are very unstable. Factors are mixed to some extent based on the time series and spectra. |
| 4 | 0 | 0 | 1.32 | OA factors could be identified as MO-OOA, LO-OOA, COA and HOA. Time series and diurnal variations of OA factors are consistent with the external tracers. But, the signal of characteristic ion *m/z* 60 biomass burning is strong in HOA factor. |
| **5** | **1** | **0** | **1.25** | **Final choice for the PMF solution. Two OOA factors, COA, HOA and BBOA are identified. Two similar OOA factors are combined for further analysis. Time series and diurnal variations of OA factors are consistent with the external tracers.** |
| 6-10 | 0 | 0 | 1.20-1.06 | Factor split. OOA was split into three or more factors with similar spectra, however, different time series. |
| 5 | -3 to 3 | 0 | 1.25-1.39 | In *f$_{peak}$* range from −1.0 to 1.0, factor MS of OOA and COA are nearly identical, but there is a shift between HOA and BBOA for some ion fragments. The time series of OOA and HOA are nearly identical, but the other show some changes. |

For the summer observation, the 4-factor, *f$_{peak}$*=0 solution was selected as the optimum solution. Four OA factors are more oxidized (MO-OOA) and less oxidized OOA (LO-OOA), cooking OA (COA) and hydrocarbon-like OA (HOA). The performances of spectra and time series of the four factors at different *f$_{peak}$* were also investigated. The detailed information on how to select the optimum PMF solution can be found in Figure S19-S21 and Table S5.

[Figure]

Figure S19. Diagnostic plots of the PMF analysis on OA mass spectral matrix for the summer observation.

[Figure]

Figure S20. The spectra and time series of 4-factor solution at different $f_{peak}$ values for the summer observation.

[Figure]

Figure S21. Unit mass spectra and time series of OA factors for 5-factor solution. The factors are marked as OOA1, OOA2, COA, OOA3 and HOA, respectively. OOA1, OOA2 and OOA3 show similar MS features (r=0.87–0.90). It is unclear if these OOA components represent distinct sources or chemical types. The elemental ratios and OA/OC ratios of each component are added.

**Table S5** Descriptions of PMF solutions for the summer observation in Beijing.

| Factor number | Fpeak | Seed | Q/Q$_{exp}$ | Solution Description |
|---|---|---|---|---|
| 1 | 0 | 0 | 7.20 | Too few factors, large residuals at time periods and key *m/z*'s |
| 2 | 0 | 0 | 5.31 | Too few factors, large residuals at time periods and key *m/z*'s |
| 3 | 0 | 0 | 4.73 | Too few factors (OOA, HOA and COA). Factors are mixed to some extent based on the time series and spectra. |
| **4** | **0** | **0** | **4.53** | **Optimum solution for the PMF analysis (MO-OOA, LO-OOA, COA and HOA). Time series and diurnal variations of OA factors are consistent with the external tracers. The spectra of four factors are consistent with the source spectra in AMS spectra database.** |
| 5-10 | 0 | 0 | 4.30-3.74 | Factor split. Take 5 factor number solution as an example, OOA is likely split into three factors with similar mass spectra and different time series. However, it is difficult to explain if they represent distinct sources or chemical types. |
| 4 | -3 to 3 | 0 | 4.53-4.58 | In *f$_{peak}$* range from −1.0 to 1.0, factor MS and time series are nearly identical. |

The solution of the PMF analysis for the autumn observation is similar to that for the spring observation. When OA was separated into five fractions, OOA was also split into two factors, but a BBOA factor of distinct characteristics ($f_{60}$=1.3%) could be identified. When more than five factors, OOA decomposed into three or more factors. The performances of spectra and time series of the four factors at different $f_{peak}$ are nearly identical. The five factors, $f_{Peak}$=0 and seed=0 solution is chosen as the optimal solution for this PMF analysis. In the five-factor solution, two OOA factors have similar MS characteristics (r= 0.976) and the elemental ratios and OA/OC ratios (O/C: 0.85–0.91; H/C: 1.24–1.40; OA/OC: 2.24–2.37) are close. It is unclear if the two OOA components represent distinct sources or chemical types. Thus, two OOA factors were combined into total OOA for further analysis (Hayes et al., 2013). Finally, four factors of OA were obtained, i.e., oxygenated OA (OOA), cooking OA (COA), hydrocarbon-like OA (HOA), and biomass burning OA(BBOA), as shown in Fig. S30. The detailed information on how to select the optimum PMF solution are given as Fig. S22-S24 and Table S6.

[Figure]

Figure S22. Diagnostic plots of the PMF analysis on OA mass spectral matrix for the autumn observation.

[Figure]

Figure S23. The spectra and time series of 5-factor solution at different $f_{peak}$ values for the autumn observation.

[Figure]

Figure S24. Unit mass spectra and time series of OA factors for 6-factor solution. The factors are marked as OOA1, BBOA, OOA2, COA, HOA1 and HOA2, respectively. The time series of BBOA and OOA2 trend well (r=0.78). HOA1 and HOA2 have similar MS (r=0.94) and diurnal variations (r=0.93). These factors appear mixed with each other.

**Table S6** Descriptions of PMF solutions for the autumn observation in Beijing.

| Factor number | Fpeak | Seed | $Q/Q_{exp}$ | Solution Description |
|---|---|---|---|---|
| 1 | 0 | 0 | 4.55 | Too few factors, large residuals at time periods and key $m/z$'s |
| 2 | 0 | 0 | 3.09 | Too few factors, large residuals at time periods and key $m/z$'s |
| 3 | 0 | 0 | 2.25 | Too few factors (OOA, COA and HOA). The $Q/Q_{exp}$ at different seeds (0-50) are very unstable. The HOA factor contain high abundance (1.0%) of $m/z$ 60. |
| 4 | 0 | 0 | 2.07 | Four factors include two similar OOA factors, COA and HOA. The HOA factor contain high abundance (1.1%) of $m/z$ 60. |
| **5** | **0** | **0** | **1.97** | **Optimum solution for the PMF analysis (two OOA factor, COA, HOA and BBOA). Two similar OOA factors are combined for further analysis. Time series and diurnal variations of OA factors are consistent with the external tracers.** |
| 6-10 | 0 | 0 | 1.88-1.71 | Factor split. Some of the split factors have time series and MS that appear mixed. |
| 5 | -3 to 3 | 0 | 1.97-2.10 | In $f_{peak}$ range from −1.0 to 1.0, factor MS and time series are nearly identical. |

For the winter observation, a 5-factor, $f_{peak}$=0 solution was selected as the optimum solution. Five OA factors are more oxidized (MO-OOA) and less oxidized OOA (LO-OOA), cooking OA (COA), coal combustion OA (CCOA) and hydrocarbon-like OA (HOA), respectively. The performances of spectra and time series of the five factors at different $f_{peak}$ were also investigated. The detailed information on how to select the optimum PMF solution can be found in Fig. S25-S27 and Table S7.

[Figure]

Figure S25. Diagnostic plots of the PMF analysis on OA mass spectral matrix for the winter observation.

[Figure]

Figure S26. The spectra and time series of 5-factor solution at different $f_{peak}$ values for the winter observation.

[Figure]

Figure S27. Unit mass spectra and time series of OA factors for 6-factor solution. The factors are marked as OOA1, OOA2, COA, OOA3, HOA and CCOA, respectively. OOA1, OOA2 and OOA3 show similar time series or MS features (r=0.56–0.95). The characteristics of OOA3 factor is not obvious. It is unclear if these factors represent distinct sources or chemical types.

**Table S7** Descriptions of PMF solutions for the winter observation in Beijing.

| Factor number | Fpeak | Seed | Q/Q$_{exp}$ | Solution Description |
|---|---|---|---|---|
| 1 | 0 | 0 | 7.09 | Too few factors, large residuals at time periods and key *m/z*'s |
| 2 | 0 | 0 | 3.57 | Too few factors, large residuals at time periods and key *m/z*'s |
| 3 | 0 | 0 | 3.14 | Too few factors (OOA-, HOA- and COA-like). The Q/Q$_{exp}$ at different seeds (0-50) are very unstable. Factors are mixed to some extent based on the time series and spectra. |
| 4 | 0 | 0 | 2.84 | OA is split to two OOA factors, COA and HOA. It seems that HOA mixed with CCOA. |
| **5** | **0** | **0** | **2.70** | **Optimum choice for PMF factors (MO-OOA, LO-OOA, COA, HOA and CCOA). Time series and diurnal variations of OA factors are consistent with the external tracers. The spectra of four factors are consistent with the source spectra in AMS spectra database.** |
| 6-10 | 0 | 0 | 2.59-2.33 | Factor split. Take 6 factor number solution as an example, OOA was split into three factors with similar spectra and/or time series. |
| 5 | -3 to 3 | 0 | 2.70-2.78 | In *f$_{peak}$* range from −1.0 to 1, factor MS and time series are nearly identical, but there is likely a shift of the time series for LO-OOA and COA during the heavy-pollution episodes. |

[revised manuscript text omitted]
. According to the results, the ambient aerosols were generally in aqueous phase. The average values of aerosol LWC in four seasons were $17.3\pm28.5$, $18.8\pm24.9$, $12.8\pm27.3$ and $25.2\pm32.8$ $\mu g\ m^{-3}$, respectively. During the heavy-polluted episodes, the LWC was frequently higher than 100 $\mu g\ m^{-3}$.

It is well known that most of aerosol sulfate are formed from heterogeneous or aqueous-phase/cloud processes (Kulmala et al., 2016). On a global scale, about 80% of the sulfate formation occurs within clouds. Ambient aerosol populations often show two distinct submicron modes (<0.2 $\mu m$ and 0.5–1 $\mu m$) where the larger (droplet) mode is formed from the smaller (condensation) mode through volume-phase reactions in clouds and wet aerosols (Ervens et al., 2011). Based on this assumption, Guo et al. (2010) found that the gas-to-particle condensation process was important for aerosol pollution in the summer of Beijing. In urban Beijing, the formation of sulfate was mainly attributed to in-cloud or aerosol droplet process (80%) and gas condensation process (14%).

Table S14 shows the correlation coefficients between OOA and some indicators (RH, LWC, $O_3$ and $O_x$). As shown in Table S14, secondary inorganics (sulfate and nitrate) correlated well with RH and/or LWC in four seasons, indicating that the aqueous-phase reactions in aerosols played an important role in secondary inorganic formation in Beijing. The contributions of photochemical processes to the formation of sulfate and nitrate in four seasons were likely less than those of aqueous-phase reactions according to the weaker correlations between secondary inorganics and odd oxygen ($O_x=O_3+NO_2$). Especially, in summer nitrate showed no correlation with $O_x$. As shown in Fig. S40, when RH was higher than 40% (or 30% in winter), aqueous-phase processed likely played a dominant role in secondary inorganic formation.

**Table S14** Pearson correlation coefficients between secondary organic and inorganic species and some indicators (RH, LWC, $O_3$ and $O_x$). Coefficients greater than 0.5 are in bold. Correlation is significant at the 0.01 level (2-tailed) except for those marked by [#].

| | | RH | LWC | $O_3$ | $O_x$ |
|---|---|---|---|---|---|
| Spring | OOA | **.661** | **.754** | -.199 | .345 |
| | $SO_4^{2-}$ | **.764** | **.901** | -.207 | .186 |
| | $NO_3^-$ | **.705** | **.827** | -.318 | .254 |
| Summer | MO-OOA | .176 | **.751** | .131 | .264 |
| | LO-OOA | .005[#] | .469 | .360 | **.527** |
| | $SO_4^{2-}$ | .114 | **.686** | .262 | .359 |
| | $NO_3^-$ | .335 | **.873** | -.123 | .000[#] |
| Autumn | OOA | .483 | **.803** | -.433 | **.571** |
| | $SO_4^{2-}$ | **.552** | **.919** | -.340 | .338 |
| | $NO_3^-$ | .489 | **.854** | -.379 | **.548** |
| Winter | MO-OOA | **.624** | **.647** | -.504 | **.640** |
| | LO-OOA | **.692** | **.840** | -.534 | **.726** |
| | $SO_4^{2-}$ | **.801** | **.899** | -.613 | **.597** |
| | $NO_3^-$ | **.785** | **.819** | -.637 | **.655** |

[Figure]

Figure S40. Influences of RH and $O_3$ concentrations on sulfate and nitrate formation.

Conversion ratios for sulfur and nitrogen ($F_S$ and $F_N$) were calculated as follows:

$$F_S = n - SO_4^{2-}/(n - SO_4^{2-} + n - SO_2) \tag{3}$$

$$F_N = n - NO_3^-/(n - NO_3^- + n - NO_2) \tag{4}$$

where $n$ means the amount of substance of the gaseous and particulate pollutants, mol m$^{-3}$.

The good correlation between $F_N$/$F_S$ and RH/LWC also support that aqueous-phase reactions in aqueous aerosols and/or clouds could contribute to secondary inorganic formation remarkably in highly humid air.

**Table S15** Pearson correlation coefficients between $F_S$ and $F_N$ with RH, LWC, $O_3$, $O_x$ and $NH_4^+$.

|  | Spring | | Summer | | Autumn | | Winter | |
| --- | --- | --- | --- | --- | --- | --- | --- | --- |
|  | $F_S$ | $F_N$ | $F_S$ | $F_N$ | $F_S$ | $F_N$ | $F_S$ | $F_N$ |
| RH | .339 | **.722** | **.639** | .393 | .432 | **.574** | **.531** | **.744** |
| LWC | .475 | **.816** | .464 | **.816** | **.647** | **.874** | **.583** | **.676** |
| $O_3$ | .024[*] | -.146 | -.100 | .024[*] | .035 | -.321 | -.268 | -.518 |
| $O_x$ | -.166 | .277 | -.096 | .052 | -.122 | .359 | .342 | .368 |
| $NH_4^+$ | .324 | **.924** | .353 | **.822** | .495 | **.938** | .598 | **.855** |

Note: Coefficients greater than 0.5 are in bold. Correlation is significant at the 0.01 level (2-tailed) except for those marked with [*].

Based on laboratory experiments and simulations, Ervens et al. (2011) suggest that SOA formed in cloud and aerosol water (aqSOA) might contribute almost as much mass as SOA formed in the gas phase to the SOA budget, with highest contributions from biogenic emissions of VOCs in the presence of anthropogenic pollutants (i.e., $NO_x$) at high RH and cloudiness. Xu et al. (2017) show that aqueous-phase processes have a dominant impact on the formation of MO-OOA, and the contribution of MO-OOA to OA increases substantially as a function of RH or liquid water content (LWC) in aerosols. In contrast, photochemical processing plays a major role in the formation of LO-OOA, as indicated by the strong correlations between LO-OOA and $O_x$ during periods of photochemical production.

The good correlations between OOA and RH and/or LWC indicate that aqueous-phase reactions play a dominant role in OOA formation (Table S14). The slope of OOA against $O_x$ steepened with the increase of RH and LWC (Figs. S41 and S42), also implying that the aqueous-phase oxidation was an important pathway of the OOA formation. The strong correlations between $O_x$ and LO-OOA in summer, and between $O_x$ and OOA in autumn and winter, suggesting photochemical processes also contributed substantially to OOA, especially LO-OOA, in these seasons.

It is difficult to give a quantitative result for the relative contribution of photochemical vs. aqueous-phase oxidation to the secondary formation based on field observation data only. Further studies including laboratory experiments, field observations and model simulations are needed to close the gaps in the current understanding of SOA formation pathways (Ervens et al., 2011).

[Figure]

Figure S41. Scattering plots of OOA mass concentrations against $O_x$ concentrations. (a) spring; (b) summer; (c) autumn and (d) winter. Data points are color coded by RH.

[Figure]

Figure S42. The same as above. Data points are color coded by estimated LWC in aerosols.

**S11 Impacts of regional and long range transport on atmospheric aerosols**

To give an insight into the impacts of regional/long-distance transport on atmospheric aerosols in Beijing, the backward trajectories of air parcels during the observation periods were calculated with the NOAA's HYSPLIT4 trajectory model (http://www.arl.noaa.gov/hysplit.html). A new 3-day backward trajectory was traced from the observation site at an altitude of 500 m above ground level every hour. Cluster analyses of backward trajectories were applied to reveal the major pathways during different campaigns (Fig. S43).

During the seasonal observations in Beijing, the pathways of dominant air masses are different. Both long-distance transported and regional/local air masses influenced Beijing. In summer, the transport distance of long-distance transported air masses was shorter than in other seasons. In general, with the decrease of transport distance, the concentration of $PM_1$ gradually increased. When Beijing was dominated by regional/local air masses, the fractions of secondary inorganic species (SNA) increased, while the contributions of carbonaceous components (OA+BC) decreased, which is consistent with the previous results in Beijing (Sun et al., 2010; Huang et al., 2010; Zhang et al., 2014). Higher concentrations of SNA and $PM_1$ under the control of regional/local air masses reflected the great contribution of secondary formation from the gaseous precursors (e.g., $NO_x$ and $SO_2$) emitted by vehicles and coal combustion in urban areas.

During the observations in spring, summer and autumn, the contributions of OOA (MO-OOA+LO-OOA) increased when Beijing was dominated by regional or local air masses. In summer, the fractions of LO-OOA in OA were high (29–48%) regardless of the different trajectories, signifying that the secondary formation from photochemical oxidations probably made an important contribution to OA. During the winter observation, POA and OOA contributed equally to OA in most cases due to the long-lasting stable weather conditions, indicating that both primary pollutants and regional secondary formation made important contributions to OA. When Beijing was dominated by long-distance transported air masses from north polar regions in winter, OOA contributed more significantly to OA, implying that organic aerosols were fully aged during long-distance transport.

[Figure]

Figure S43. Back trajectories for each of the identified clusters and corresponding average main components of PM$_1$ and OA in PM$_1$ during the seasonal campaigns. (a) spring; (b)summer; (c) autumn and (d) winter. The filling color of main chemical species in PM$_1$ is the same with other figures.

---

## Author Response (AR2)

**Response to Co-editor Comments**

Comments to the Author:

Dear Authors, thank you for your patience. I've heard back from the reviewers and based on their readings and mine my reading of the revised manuscript we feel this study is ready before publication. However, it came up again about IEPOX-derived SOA not being resolved or discussed from this seasonal dataset. Have the authors seen the recent publication by Zhang et al. (2017, GRL, http://onlinelibrary.wiley.com/doi/10.1002/2016GL072368/abstract) on the limited formation of IEPOX-derived SOA in eastern China during summer? The reviewer and I were still curious if you tried to determine the presence of IEPOX-derived SOA from your PMF analyses? If you didn't resolve it, is that consistent with this recent paper? You might want to at least mention this point (either way) in the discussion of the results. Once you address this final point, I will gladly accept your revised manuscript.

Most sincerely, Jason Surratt

**Response:** Dear Dr. Jason Surratt, thank you very much for handling our manuscript and your decision "Reconsider after minor revisions".

About IEPOX-SOA, as we mentioned in the previous responses, we give response as follows.

IEPOX-SOA can contribute substantially to OA concentrations in forested areas under low NO conditions (Hu et al., 2016). However, this study was conducted at a typical urban site, and the concentrations of NO were 10.5, 3.5, 37.5 and 24.4 during four seasons, respectively (Table 1). The IEPOX-SOA factors were resolved by PMF because they

made a significant contribution to OA, and the influence of vegetation was important. For example, Budisulistiorini et al. (2013, 2016) found that IEPOX-SOA accounted for 27-41% of OA, and showed a seasonal difference at a downtown urban site and a rural/forested site. Hu et al. (2015) mapped the IEPOX-SOA fractions of OA at diverse sites across the world, and the resolved IEPOX-SOA accounted for 6−36%. IEPOX-SOA is only identified at the sites where the predicted average IEPOX concentration higher than ~30 ppt. In addition, no IEPOX-SOA factor (i.e., below the PMF detection limit of ~5% of OA) was found in areas strongly influenced by urban emissions where high-NO concentrations suppress the IEPOX pathway, even in the presence of substantial isoprene concentrations (Hu et al., 2015).

Even in summer (modeled data in July 2013), the gas-phase IEPOX concentrations are quite low (close to 0) in North China, so there were no IEPOX-SOA factors resolved at the sites in North China, including Beijing and Changdao Island (Hu et al., 2015). There might be relatively limited IEPOX-SOA contribution to the OA in Beijing, but the IEPOX-SOA factor cannot be resolved in our dataset by free PMF. In recently published papers (Fig. 5; Hu et al., 2016; Sun et al., 2015, 2016a, 2016b; Wang et al., 2015; Xu et al., 2017; Zhang et al., 2016), there were also no IEPOX-SOA factor resolved by PMF analysis in urban Beijing.

Recently, Zhang et al. (2017) reported that the average IEPOX-SOA concentration is $0.33\pm0.19$ μg m$^{-3}$ (3.8% of the total OA) under $NO_x$-rich environments in mid-Eastern China in summer 2013. The concentration was much smaller than those (2-4 μg m$^{-3}$) in IEPOX-rich regions in the southeastern US, i.e., the formation of IEPOX-SOA under polluted urban environments is quite limited.

In summary, it is reasonable that IEPOX-SOA factor cannot be resolved in Beijing. In this study, we would like to keep using PMF analysis for OA factorization.

In the revision,

Page 12, Line 10: "*Different from the two sites in southeastern USA, there were no isoprene-epoxydiols-derived SOA and a biogenic influenced factor characterized by distinct m/z 91 resolved in urban Beijing, the same as addressed in Hu et al. (2015).*" was changed to "**Different from the two sites in southeastern USA, there were no isoprene-epoxydiols-derived SOA (IEPOX-SOA) and a biogenic influenced factor characterized by distinct m/z 91 resolved in urban Beijing. No IEPOX-SOA factor (i.e., below the PMF detection limit of ~5% of OA) was found in Beijing because the formation of IEPOX-SOA under IEPOX-poor and polluted urban environments is quite limited (Hu et al., 2015).*"

References:

Hu, W. W., Campuzano-Jost, P., Palm, B. B., Day, D. A., Ortega, A. M., Hayes, P. L., Krechmer, J. E., Chen, Q., Kuwata, M., Liu, Y. J., de Sá, S. S., McKinney, K., Martin, S. T., Hu, M., Budisulistiorini, S. H., Riva, M., Surratt, J. D., St. Clair, J. M., Isaacman-Van Wertz, G., Yee, L. D., Goldstein, A. H., Carbone, S., Brito, J., Artaxo, P., de Gouw, J. A., Koss, A., Wisthaler, A., Mikoviny, T., Karl, T., Kaser, L., Jud, W., Hansel, A., Docherty, K. S., Alexander, M. L., Robinson, N. H., Coe, H., Allan, J. D., Canagaratna, M. R., Paulot, F., and Jimenez, J. L.: Characterization of a real-time tracer for isoprene epoxydiols-derived secondary organic aerosol (IEPOX-SOA) from aerosol mass spectrometer measurements, Atmos. Chem. Phys., 15, 11807-11833, doi:10.5194/acp-15-11807-2015, 2015.

Hu, W., Hu, M., Hu, W., Jimenez, J. L., Yuan, B., Chen, W., Wang, M., Wu, Y., Chen, C., Wang, Z., Peng, J., Zeng, L., and Shao, M.: Chemical composition, sources and aging process of sub-micron aerosols in Beijing: contrast between summer and winter, J. Geophys. Res. Atmos., 121, 1955–1977, doi:10.1002/2015JD024020,

2016.

Sun, Y., Du, W., Fu, P., Wang, Q., Li, J., Ge, X., Zhang, Q., Zhu, C., Ren, L., Xu, W., Zhao, J., Han, T., Worsnop, D. R., and Wang, Z.: Primary and secondary aerosols in Beijing in winter: sources, variations and processes, Atmos. Chem. Phys., 16, 8309-8329, doi:10.5194/acp-16-8309-2016, 2016a.

Sun, Y., Du, W., Wang, Q., Zhang, Q., Chen, C., Chen, Y., Chen, Z., Fu, P., Wang, Z., Gao, Z., and Worsnop, D. R.: Real-Time Characterization of Aerosol Particle Composition above the Urban Canopy in Beijing: Insights into the Interactions between the Atmospheric Boundary Layer and Aerosol Chemistry, Environ. Sci. Technol., doi:10.1021/acs.est.5b02373, 2015.

Sun, Y., Wang, Z., Wild, O., Xu, W., Chen, C., Fu, P., Du, W., Zhou, L., Zhang, Q., Han, T., Wang, Q., Pan, X., Zheng, H., Li, J., Guo, X., Liu, J., and Worsnop, D. R.: "APEC Blue": Secondary aerosol reductions from emission controls in Beijing, Sci. Rep., 6, 20668, doi:10.1038/srep20668, 2016b.

Wang, Q., Sun, Y., Jiang, Q., Du, W., Sun, C., Fu, P., and Wang, Z.: Chemical composition of aerosol particles and light extinction apportionment before and during the heating season in Beijing, China, J. Geophys. Res. Atmos., 120, 12,708–12,722, doi:10.1002/2015JD023871, 2015.

Xu, W., Han, T., Du, W., Wang, Q., Chen, C., Zhao, J., Zhang, Y., Li, J., Fu, P., Wang, Z., Worsnop, D. R., and Sun, Y.: Effects of aqueous-phase and photochemical processing on secondary organic aerosol formation and evolution in Beijing, China, Environ. Sci. Technol., 51, 762-770, 10.1021/acs.est.6b04498, 2017.

Zhang, J. K., Cheng, M. T., Ji, D. S., Liu, Z. R., Hu, B., Sun, Y., and Wang, Y. S.: Characterization of submicron particles during biomass burning and coal combustion periods in Beijing, China, Sci. Total Environ., 562, 812-821, 10.1016/j.scitotenv.2016.04.015, 2016.

Zhang, Y., Tang, L., Sun, Y., Favez, O., Canonaco, F., Albinet, A., Couvidat, F., Liu, D., Jayne, J.T., Wang, Z., Croteau, P.L., Canagaratna, M.R., Zhou, H.-c., Prévôt, A.S.H., Worsnop, D.R., 2017. Limited formation of isoprene epoxydiols-derived secondary organic aerosol under NOx-rich environments in Eastern China. Geophys. Res. Lett., doi: 10.1002/2016GL072368.